# SIGN-SGD VIA PARAMETER-FREE OPTIMIZATION

**Daniil Medyakov** [1,2]* **Sergey Stanko** [1,2] **Gleb Molodtsov** [1,2] **Philip Zmushko** [1,2,3,4]†
**Grigoriy Evseev**[1,2] **Egor Petrov**[1,2,3] **Aleksandr Beznosikov**[1,2,5]

[1] Basic Research of Artificial Intelligence Laboratory (BRAIn Lab)

[2] Moscow Independent Research Institute of Artificial Intelligence

[3] Yandex Research

[4] Institute of Science and Technology Austria (ISTA)

[5] Innopolis University

## ABSTRACT

Large language models have achieved major advances across domains, yet training them remains extremely resource-intensive. We revisit SIGN-SGD, which serves both as a memory-efficient optimizer for single-node training and as a gradient compression mechanism for distributed learning. This paper addresses a central limitation: the effective stepsize cannot be determined a priori because it relies on unknown, problem-specific quantities. We present a parameter-free SIGN-SGD that removes manual stepsize selection. We analyze the deterministic single-node case, and extend the method to stochastic single-node training and multi-node settings. We also incorporate the momentum technique into our algorithms and propose a memory-efficient variant that stores only gradient signs instead of full gradients. We evaluate our methods on pre-training LLaMA models (130M and 350M) and fine-tuning a Swin Transformer (28M). Across considered tasks, the proposed methods match the performance of tuned SIGN-SGD and ADAMW (grid-searched stepsizes with a cosine schedule), while avoiding tuning overhead. Employing parameter-free training yields approximately $1.5\times$ end-to-end speedup compared to runs with grid-searched stepsizes.

## 1 INTRODUCTION

Models and datasets continue to scale rapidly (Vaswani, 2017; Hoffmann et al., 2022; Alzubaidi et al., 2021). This growth drives steep increases in compute requirements, memory footprint, and wall-clock training time, consequently raising hardware costs. These pressures motivate the development of methods that accelerate training and reduce resource usage without sacrificing accuracy. A significant breakthrough arose not from designing advanced learning algorithms, but primarily from the manner in which these algorithms can be applied: distributed learning (Konečný et al., 2016; McMahan et al., 2017; Verbraeken et al., 2020). However, distributing training across $M$ nodes does not yield an $M$-fold speedup in practice, as inter-device communication remains a significant bottleneck.

The reduction of the number of transmitted packages through compression is one of the key techniques to address this issue (Seide et al., 2014; Alistarh et al., 2018). Among others, the SIGN-SGD method stands out (Bernstein et al., 2018). Solving the classic optimization problem $\min_{x \in \mathbb{R}^d} f(x)$, it utilizes

---

**Algorithm 1** SIGN-SGD

1: **Input:** Start point $x^0 \in \mathbb{R}^d$, number of iterations $T$
2: **Parameter:** Stepsize $\gamma > 0$
3: **for** $t = 0, \ldots, T-1$ **do**
4: $\quad x^{t+1} = x^t - \gamma \text{sign}(\nabla f(x^t))$
5: **end for**

---

an intuitive heuristic that takes the sign of each gradient coordinate (Algorithm 1). In the distributed setup, aggregation is performed by a majority vote on the transmitted signs of the gradients.

Additionally, SIGN-SGD is rapidly gaining popularity, even for single-node training. In contrast to methods such as ADAM (Kingma, 2014) and ADAMW (Loshchilov, 2017), which require substantial memory for storing statistics, SIGN-SGD is free from this constraint. Moreover, sign-based

---

*Email: medyakovd3@gmail.com.

†The work was completed prior to joining ISTA.

approaches offer both theoretical and practical advantages over traditional SGD (Robbins & Monro, 1951), demonstrating superior convergence (Balles & Hennig, 2018; Balles et al., 2020) and empirical performance (Kunstner et al., 2023; Zhao et al., 2024; Zmushko et al., 2024) in training large models.

Although SIGN-SGD is effectively used both for compression in distributed learning and as a memory-efficient method in a single-node regime, achieving its full potential requires selecting an appropriate stepsize. The optimal choice depends on problem-specific quantities that are unknown in practice, necessitating costly manual tuning. To address this issue, we introduce parameter-free SIGN-SGD algorithm that employ automatic stepsize selection schemes.

## 2 BRIEF LITERATURE REVIEW AND CONTRIBUTIONS

To situate the problem and motivate our algorithms, this section reviews the literature and distills the open challenges that guide our contributions.

- We begin by revisiting SIGN-SGD and identifying the theoretically desirable stepsize that would enable effective training without manual tuning.
- Next, we survey parameter-free optimization methods, highlighting their advantages and limitations.
- We conclude by stating the contributions of this work and explaining how they address the gaps.

### 2.1 RELATED WORK

• **Sign-SGD.** In the original paper on SIGN-SGD (Bernstein et al., 2018), the authors explored convergence in the paradigm of finding a near-stationary point, i.e., such $x \in \mathbb{R}^d$, that $\|\nabla f(x)\| \leqslant \varepsilon$, where $\varepsilon$ represents the accuracy of the solution. Moreover, to achieve convergence with respect to the variance term, the authors utilized mini-batches. Both this convergence criterion and the use of mini-batches are essential components of the analysis. As shown in (Karimireddy et al., 2019), SIGN-SGD may fail to converge when considered the regret minimization. Moreover, to achieve convergence with respect to the variance term, the authors of (Karimireddy et al., 2019) utilized mini-batches. Meanwhile, Safaryan & Richtárik (2021) proposed a relaxation of the SIGN-SGD method and showed that at least half of the coordinates in the sign of the stochastic gradient align with those of the exact gradient, thereby enabling convergence with respect to the variance term. A number of works have also emerged around SIGN-SGD, extending it with momentum (Sun et al., 2023), providing high-probability convergence bounds (Kornilov et al., 2025), and studying it in the context of differential privacy (Jin & Dai, 2025). *Nevertheless, the possibility of selecting a stepsize independent of problem properties while achieving optimal convergence rate has been largely overlooked.*

Let us provide the basic estimate of SIGN-SGD convergence with the exact gradient oracles. This can be simply derived from Theorem 1 in (Bernstein et al., 2018):

$$\frac{1}{T} \sum_{t=0}^{T-1} \|\nabla f(x^t)\|_1 \leqslant \frac{\Delta^*}{\gamma T} + \frac{\gamma L_\infty}{2},$$

where $L_\infty$ is the smoothness constant of the objective $f$ with respect to $l_\infty$-norm, and $\Delta^* = f(x^0) - f(x^*)$ represents the initial distance to the solution. Putting

$$\gamma = \frac{\sqrt{\Delta^*}}{\sqrt{L_\infty T}} \quad, \quad \text{we obtain optimal } \mathcal{O}\left(\frac{\sqrt{\Delta^* L_\infty}}{\sqrt{T}}\right) \quad \text{convergence rate.} \tag{1}$$

This stepsize poses challenges, as it depends on the problem's hyperparameters. To address this issue, we turn to various techniques that facilitate the provision of an adaptive stepsize.

• **Parameter-free approaches.** In the non-smooth setting, considering regret minimization, classic gradient methods (Robbins & Monro, 1951; Moulines & Bach, 2011; Stich, 2019; Lan, 2020) require

$$\gamma = \frac{\|x^0 - x^*\|_2}{M\sqrt{T}} \quad \text{to have } \mathcal{O}\left(\frac{\|x^0 - x^*\|_2 M}{\sqrt{T}}\right) \quad \text{convergence rate.} \tag{2}$$

This estimate is (worst-case) optimal in its complexity class (Nemirovskij & Yudin, 1983). We let $M$ denote the Lipschitz constant $\left(|f(x) - f(y)| \leqslant M \|x - y\|_2 \text{ for all } x, y \in \mathbb{R}^d\right)$. The parameter-free

setting aims to adapt the stepsize automatically, without prior knowledge of the initial distance $\left\|x^0 - x^*\right\|_2$ or the Lipschitz constant $M$.

For the first time, the idea of an automatic stepsize setting was proposed to achieve adaptation to constant $M$. It was embodied in methods such as ADAGRAD (Duchi et al., 2011), ADAM (Kingma, 2014), RMSPROP (Tieleman & Hinton, 2012), ADADELTA (Zeiler, 2012), and ADAPTIVE SGD (Gupta et al., 2017; Attia & Koren, 2023). In these methods, computed gradients were utilized to adjust the stepsize based on the properties of $M$. *However, these methods required additional memory and computations, and they lacked adaptivity to the initial distance.* Attempts to modify $\gamma$ in equation 2 led to approaches within the general online stochastic learning setting (Orabona, 2019), such as coin betting and reward-doubling techniques (Streeter & McMahan, 2012; Orabona, 2013; McMahan & Orabona, 2014; Orabona & Pál, 2016; Cutkosky & Orabona, 2018; Cutkosky, 2019), which can also be classified as parameter-free algorithms. *Nevertheless, these approaches assumed that the stochastic oracles have some (loose) bound.*

Further studies suggested more intricate solutions in parameter-free convex stochastic optimization. These methods achieved asymptotic convergence rates comparable to classic approaches while adapting to essential hyperparameters. The starting point was the work (Carmon & Hinder, 2022) which provided adaptivity to the initial distance $\left\|x^0 - x^*\right\|_2$ through estimators of the form $\max_{t \leqslant T} \left\|x^0 - x^t\right\|_2$. To find such estimators, the authors employed an additional grid search procedure *which increased the required number of steps only in double-logarithmic time.* The primary objective of this work was to derive high-probability convergence estimates in the stochastic convex non-smooth setup. Several studies that did not utilize the additional search procedure were built upon, including (Khaled et al., 2023), (Ivgi et al., 2023) and (Kreisler et al., 2024).

The work (Defazio & Mishchenko, 2023) provided another approach for sensitivity to the initial distance. The authors iteratively constructed a sequence upper bounded by $\left\|x^0 - x^*\right\|_2$ and approximated it accordingly. *However, they considered only exact gradient oracles, which represents a significant limitation.* Later, in (Mishchenko & Defazio, 2023), the authors introduced a damping factor in the denominator to improve convergence in the logarithmic factor's square root. *Nevertheless, theoretical analysis depended on the knowledge of the Lipschitz constant, which is not a parameter-free approach.* We note that the use of the classic ADAGRAD-NORM stepsize (Duchi et al., 2011; Streeter & McMahan, 2010; Ward et al., 2020), possibly with additional factors in the denominators, remains standard for adaptation to $M$.

The orthogonal approach was presented in the work (Mishkin et al., 2024). The authors considered a smooth setup and proposed the use of local approximations of the smoothness constant $L$ to achieve adaptivity. However, the authors employed the stepsize $\gamma^t = \frac{\left\|x^{t+1}(\gamma^t) - x^t\right\|_2}{\left\|\nabla f(x^{t+1}(\gamma^t)) - \nabla f(x^t)\right\|_2}$ at the $t$-th iteration, where $\gamma^t$ was determined by exponential search in the manner (Carmon & Hinder, 2022) or by Newton's method. *Both variants are inefficient.*

In light of the literature, we present the main directions of this study. Our goal is to provide the parameter-free SIGN-SGD method that achieves a convergence rate comparable to the optimal stepsize tuning 1.

## 2.2 CONTRIBUTIONS

We propose a novel mechanism for estimators compared to existing approaches. Instead of the classic $\left\|x^0 - x^*\right\|$ and $M$ hyperparameters in equation 2, we aim to gain the tolerance to $f(x^0) - f(x^*)$ and $L_\infty$ from equation 1. We now outline our contributions.

• **Parameter-free SIGN-SGD.** We introduce a parameter-free SIGN-SGD method. The core idea involves per-iteration step-size adaptation. Every iteration, we choose estimators of $L_\infty$ and $f(x^0) - f(x^*)$ using the current gradient information. This design is practical, as it requires no additional hyperparameter search or restarts. As a starting point, we analyze the exact gradients setup.
• **Stochastic and distributed settings.** We study our algorithm in the distributed setting and the case of stochastic gradient oracles. A lack of stochastic analysis presents a significant drawback in parameter-free optimization. Our work addresses this limitation.
• **Practical extensions.** We extend our approach in two important directions.
  • We incorporate momentum to improve practical performance.

- We provide a memory-efficient parameter-free version. It stores only the sign of the gradient from the previous step while remaining an adaptivity to the problem properties.

• **Theoretical analysis.** We provide a comprehensive theoretical analysis of the proposed methods and establish convergence guarantees. In our setup, we consider a convex and smooth objective.

• **Experimental validation.** We demonstrate that our methods are competitive in practical tasks, including LLM and ViT training. An Adam-style momentum variant further improves performance across both language and vision benchmarks. Empirically, parameter-free training matches or is slightly below tuned SIGN-SGD and AdamW with cosine schedules, while achieving appreciably better overall training time.

# 3 ALGORITHMS AND CONVERGENCE ANALYSIS

• **Notation.** We begin with the following notation: $\mathbb{E}[\cdot]$ denotes the expected value of a random variable, $\|x\|_2 = \sqrt{\langle x, x \rangle}$ represents the Euclidean norm of the vector $x \in \mathbb{R}^d$, $\|x\|_1 = \sum_{i=1}^d |x_i|$ refers to the $\ell_1$-norm of the vector $x$, and $\|x\|_\infty = \max_{i \in [d]} |x_i|$ defines the $\ell_\infty$-norm of the vector $x$.

• **Assumptions.** We present the assumptions regarding the objective function $f$

**Assumption 3.1.** The function $f$ is $L_\infty$-smooth, i.e., it satisfies $\|\nabla f(x) - \nabla f(y)\|_1 \leqslant L_\infty \|x - y\|_\infty$ for any $x, y \in \mathbb{R}^d$.

**Assumption 3.2.** The function $f$ is convex, i.e., it satisfies $f(x) \leqslant f(y) + \langle \nabla f(x), x - y \rangle$ for any $x, y \in \mathbb{R}^d$.

Although neural networks are inherently non-convex, theoretical analysis under convexity assumptions remains relevant. Recent studies suggest that deep neural networks often exhibit properties similar to convexity in certain regions, making insights from convex analysis applicable (Kleinberg et al., 2018; Zhou et al., 2019; Liu et al., 2022). Moreover, convex optimization serves as a theoretical foundation for the design of optimization algorithms. For example, momentum (Nesterov et al., 2018) and AdaGrad (Duchi et al., 2011) were initially developed and analyzed for convex problems.

**Assumption 3.3.** The function $f$ has a (maybe not unique) finite minimum, i.e., $f(x^*) = \inf_{x \in \mathbb{R}^d} f(x) > -\infty$.

Now we move to the base point of our analysis: the algorithms with exact gradient oracles.

## 3.1 EXACT GRADIENTS SETTING

We begin with an additional assumption regarding the gradient oracles.

**Assumption 3.4.** At any point $x \in \mathbb{R}^d$, we have access to the exact gradient, i.e., we can compute the full gradient value $\nabla f(x)$.

We now present the main algorithm of this paper named ALIAS (Automatic Local per-Iteration Approximation of the Stepsize, Algorithm 2). At each iteration, it utilizes the stepsize selection in a specific manner to gain adaptivity to the global parameters of the problem. Below, we provide an explanation of the algorithm and offer some intuition why the presented stepsize facilitates adaptivity. Considering the stepsize in equation 1, we need to approximate the numerator and denominator. Thus, we first analyze how to estimate $\Delta^*$, and then proceed with $L_\infty$.

We start with a positive scalar $d^0$, representing the initial approximation of $\Delta^*$. Next, we construct a new approximation based on the newly calculated gradient (Line 6) at each iteration of the algorithm. To bring these approximations closer to $\Delta^*$ over iterations, we take the maximum of the previous and newly computed values (Line 7). This approach yields an non-decreasing sequence that is upper bounded by $\Delta^*$ (see Lemma F.1). We adopt this iterative scheme as Option I in Algorithm 2 (Line 9).

We note that estimating $\Delta^*$ does not require advanced schemes such as Option I for most tasks, as adaptivity to $f(x^*)$ is typically not critical. As shown in (Boyd et al., 2003), the condition $f(x^*) = 0$ arises in problems such as finding a point in the intersection of convex sets, completing positive semi-definite matrices, or solving systems of convex inequalities.

Moreover, a lower bound $\widetilde{f}$ on $f(x^*)$ is often known or readily available. For instance, $\widetilde{f} = 0$ serves as a valid estimate in the empirical risk minimization setting. Taking this into account, we present the second option for our method, where we use $f(x^0) - \widetilde{f}$ with $\widetilde{f} \leqslant f(x^*)$ (Line 10) instead of the sequence $\{d^t\}_{t=0}^{T-1}$.

As for the denominator, at the $t$-th iteration, we approximate the local Lipschitz constant $L_\infty$ between $x^t$ and $x^{t-1}$. We accumulate it in the manner of ADAGRAD-NORM by adding it to the sum of previous approximations:

$$\eta^t = \eta^{t-1} + \frac{\left\|\nabla f(x^t) - \nabla f(x^{t-1})\right\|_1}{\|x^t - x^{t-1}\|_\infty}.$$

In the stepsize, the corresponding to the denominator coefficient appears as:

---

**Algorithm 2** ALIAS

1: **Input:** Starting point $x^0 \in \mathbb{R}^d$, initial $L_\infty$-approximation $\eta^{-1} = 0$, initial $\Delta^*$-approximation $d^0 \in \mathbb{R}_+$, lower bound $\widetilde{f}$ on $f(x^*)$, number of iterations $T$
2: **for** $t = 0, \ldots, T-1$ **do**
3:     Compute gradient $\nabla f(x^t)$
4:     $\eta^t = \eta^{t-1} + \frac{\left\|\nabla f(x^t) - \nabla f(x^{t-1})\right\|_1}{\|x^t - x^{t-1}\|_\infty}; \ \lambda^t = \frac{1}{\sqrt{\eta^t}}$
5:     **if** $t \neq 0$ **then**
6:         $\widetilde{d^t} = \sum_{i=0}^{t-1} \gamma^i \langle \nabla f(x^{i+1}), \text{sign}(\nabla f(x^i)) \rangle$
7:         $d^t = \max\left(d^{t-1}, \widetilde{d^t}\right)$
8:     **end if**
9:     Option I: $\gamma^t = \lambda^t \sqrt{d^t}$
10:    Option II: $\gamma^t = \lambda^t \sqrt{f(x^0) - \widetilde{f}}$
11:    $x^{t+1} = x^t - \gamma^t \text{sign}(\nabla f(x^t))$
12: **end for**

---

$$\lambda^t = \frac{1}{\sqrt{\eta^t}} = \frac{1}{\sqrt{\sum_{i=0}^{t-1} \frac{\|\nabla f(x^{i+1}) - \nabla f(x^i)\|_1}{\|x^{i+1} - x^i\|_\infty}}}.$$

This stepsize facilitates iterative adaptation to the objective landscape. We are now prepared to present the main theoretical results of this section.

**Theorem 3.5.** *Suppose Assumptions 3.1, 3.2, 3.3, 3.4 hold. We denote $\varepsilon \geqslant \frac{1}{T} \sum_{t=0}^{T-1} \|\nabla f(x^t)\|_1$, $L_\infty^0 = \frac{\|\nabla f(x^1) - \nabla f(x^0)\|_1}{\|x^1 - x^0\|_\infty}$. Then Algorithm 2 with $d^0 < \Delta^*$ to reach $\varepsilon$-accuracy needs*

$$\widetilde{\mathcal{O}}\left(\frac{(\Delta^*)^2 (L_\infty)^3}{d^0 (L_\infty^0)^2 \varepsilon^2}\right) \quad and \quad \widetilde{\mathcal{O}}\left(\frac{\Delta^* (L_\infty)^3}{(L_\infty^0)^2 \varepsilon^2}\right) \quad \text{iterations with Options I and II, respectively.}$$

*Remark* 3.6. Under conditions of Theorem 3.5, Algorithm 2 with $\lambda^t = \dfrac{1}{\sqrt{L_\infty + \sum\limits_{i=0}^{t-1} \frac{\|\nabla f(x^{i+1}) - \nabla f(x^i)\|_1}{\|x^{i+1} - x^i\|_\infty}}}$

to reach $\varepsilon$-accuracy, where $\varepsilon \geqslant \frac{1}{T} \sum_{t=0}^{T-1} \|\nabla f(x^t)\|_1$, needs

$$\widetilde{\mathcal{O}}\left(\frac{(\Delta^*)^2 L_\infty}{d^0 \varepsilon^2}\right) \quad and \quad \widetilde{\mathcal{O}}\left(\frac{\Delta^* L_\infty}{\varepsilon^2}\right) \quad \text{iterations with Options I and II, respectively.}$$

**Discussion of the results.** Since we provide convergence guarantees for finding near-stationary points for a convex objective, we first examine the relationship between convergence rates in convex and non-convex settings. For instance, we compare gradient descent rates using the gradient norm as the convergence criterion. While the behavior of gradient norm minimization is well understood in the non-convex setting (Arjevani et al., 2023), it is specific in the context of convex optimization. Notably, Allen-Zhu (2018) showed that vanilla gradient descent – without acceleration or additional techniques – achieves the same $\mathcal{O}\left(1/\varepsilon^2\right)$ rate for finding near-stationary points in both convex and non-convex settings. However, as previously noted, SIGN-SGD does not admit convergence guarantees beyond any criterion except the gradient norm, even in the convex case. Consequently, convergence analysis for sign-based methods must be framed in terms of finding the near-stationary point. Thus, our convex rate is not superior to that of the non-convex case. Moreover, the bound in Theorem 3.5 includes an additional factor of $\left(L_\infty/L_\infty^0\right)^2$ compared to Remark 3.6. However, the algorithm analyzed in Remark 3.6 is not parameter-free: it requires prior knowledge of $L_\infty$. In Appendix A, we present empirical results for varying values of $L_\infty$, which demonstrate that this additive factor has negligible impact on the practical convergence of Algorithm 2.

So far, we propose an algorithm and provide the theoretical analysis behind it. However, the analysis assumes access to exact gradient oracles – an unrealistic assumption in practice. We now extend the analysis to more realistic scenarios involving stochastic oracles.

## 3.2 STOCHASTIC GRADIENTS SETTING

We begin with the assumption regarding the gradient oracles.

**Assumption 3.7.** At any point $x \in \mathbb{R}^d$ we have access to the stochastic gradient, i.e., we can compute $g_\xi(x) = \nabla f(x, \xi)$ – the stochastic gradient value with respect to the randomness in the choice of samples $\xi$. Additionally, the variance of these stochastic estimators is coordinate-wise bounded, i.e., $\mathbb{E}\left([g_\xi(x)]_i - [\nabla f(x)]_i\right)^2 \leqslant \sigma_i^2$. Furthermore, this implies that $\mathbb{E}\|g_\xi(x) - \nabla f(x)\|_1 \leqslant \|\sigma\|_1$.

It is a classic assumption in stochastic optimization (Bernstein et al., 2018). Furthermore, the batch gradient $g_\xi$ typically exhibits smoothness (Liu et al., 2023). Thus, we introduce an additional assumption.

**Assumption 3.8.** The stochastic function $f_\xi$ is $L_\infty$-smooth according to the realization $\xi$, i.e., it satisfies $\|g_\xi(x) - g_\xi(y)\|_1 \leqslant L_\infty \|x - y\|_\infty$ for any $x, y \in \mathbb{R}^d$, $\xi$.

The stochastic formulation of the problem (Assumption 3.7) necessitates modifications of Algorithm 2. This algorithm assumes access to the exact gradients, and the estimation of the local smoothness constant relies on computing full gradients. Thus, our goal is to modify Line 4 in Algorithm 2. Utilizing Assumption 3.8, we can construct a local approximation of $L_\infty$ on the $t$-th iteration via stochastic gradients with respect to the stochastic realization $\xi^t$. Namely,

$$\lambda^t = \frac{1}{\sqrt{\sum_{i=0}^{t-1} \frac{\left\|g_{\xi^{i+1}}^{i+1} - g_{\xi^{i+1}}^i\right\|_1}{\|x^{i+1} - x^i\|_\infty}}},$$

where $g_{\xi^t}^t$ is the stochastic gradient computed at the $t$-th iteration based on the stochastic realization $\xi^t$. We query the oracle twice per iteration, utilizing the current and subsequent stochastic realizations. Another change in Algorithm 2 involves performing a step in Line 11 regarding $\text{sign}(g_{\xi^t}^t)$. In the subsequent theoretical analysis, we focus solely on Option II in Algorithm 2. We provide a formal description of the stochastic method, Algorithm 7, in Appendix F.2. There, we present both the practical and theoretical versions.

We now present the convergence results.

**Theorem 3.9.** *Suppose Assumptions 3.8, 3.2, 3.3, 3.7 hold. Then Algorithm 2 with Option II to reach $\varepsilon$-accuracy, where $\varepsilon \geqslant \sum_{t=0}^{T-1} \mathbb{E}\left[\frac{\gamma^t}{\sum_{t=0}^{T-1} \gamma^t} \|\nabla f(x^t)\|_1\right]$ and $L_\infty^{t,\xi^{t+1}} = \frac{\left\|g_{\xi^{t+1}}^{t+1} - g_{\xi^t}^t\right\|_1}{\|x^{t+1} - x^t\|_\infty}$, needs*

$$\widetilde{\mathcal{O}}\left(\frac{\Delta^*(L_\infty)^3}{\varepsilon^2}\left(\mathbb{E}\left(\frac{1}{L_\infty^{0,\xi^1}}\right)^2\right) + \|\sigma\|_1^2 L_\infty\left(\mathbb{E}\frac{1}{\min\limits_{0 \leqslant t \leqslant T-1} L_\infty^{t,\xi^{t+1}}}\right)\right) \quad \text{iterations.}$$

*Remark* 3.10. Under the conditions of Theorem 3.9, Algorithm 2 with $\lambda^t = \frac{1}{\sqrt{L_\infty + \sum_{i=0}^{t-1} \frac{\left\|g_{\xi^{i+1}}^{i+1} - g_{\xi^{i+1}}^i\right\|_1}{\|x^{i+1} - x^i\|_\infty}}}$,

Option II and mini-batch of the size $t + 1$ at $t$-th iteration, to reach $\varepsilon$-accuracy needs

$$\widetilde{\mathcal{O}}\left(\frac{\Delta^* L_\infty}{\varepsilon^2} + \frac{\|\sigma\|_1^2 L_\infty}{\varepsilon^2}\left(\mathbb{E}\frac{1}{\min\limits_{0 \leqslant t \leqslant T-1} L_\infty^{t,\xi^{t+1}}}\right)\right) \quad \text{iterations,}$$

where $\varepsilon \geqslant \frac{1}{T}\sum_{t=0}^{T-1} \|\nabla f(x^t)\|_1$, $L_\infty^{t,\xi^{t+1}} = \frac{\left\|g_{\xi^{t+1}}^{t+1} - g_{\xi^t}^t\right\|_1}{\|x^{t+1} - x^t\|_\infty}$.

**Discussion of the results.** With Assumption 3.8, a more stringent version of Assumption 3.1, we approximate the smoothness constant via stochastic gradients. The key point is to measure the gradient at the current point while considering the stochastic realization from the next iteration. Since $x^t$, $\xi^t$, and $\xi^{t+1}$ are independent, we can provide a theoretical analysis. Thus, we surpass works such as (Defazio & Mishchenko, 2023; Mishchenko & Defazio, 2023; Mishkin et al., 2024),

which employed a similar idea of the adaptation to the Lipschitz constant but lacked a stochastic analysis. Notably, the result of Theorem 3.9 achieves convergence only to a neighborhood, the size of which is determined by the variance. This rate fully aligns with the original SIGN-SGD convergence (Bernstein et al., 2018). To address it in theory, we introduce increasing mini-batches analogously to (Bernstein et al., 2018) in Remark 3.10. We note that mini-batching enables convergence guarantees concerning the variance term; however, the method remains parameter-free even without it. In our experiments, we do not employ mini-batching.

We develop an analysis not only for the stochastic setting, but also for the distributed one. A full description of Algorithm 2 in the distributed setup, along with theoretical statements and proofs, is presented in Appendix F.3.

Above, we present an algorithm that can be easily applied to practical tasks. It does not require multiple restarts or additional search procedures. However, Algorithm 2 lacks the main advantage of the original SIGN-SGD method. Indeed, performing a step on the $t$-th iteration requires storing the entire gradient $\nabla f(x^{t-1})$ instead of just its sign. To address this limitation, we propose a memory-efficient modification in the next section.

### 3.3 MEMORY-EFFICIENT ALIAS

In Algorithm 2, memory efficiency is sacrificed to achieve a parameter-free stepsize. Indeed,

$$\gamma^t = \lambda^t \sqrt{d^t} = \sqrt{\frac{d^t}{\eta^t}} = \sqrt{\frac{\sum_{i=0}^{t-1} \gamma^i \langle \nabla f(x^{i+1}), \text{sign}\left(\nabla f(x^i)\right)\rangle}{\sum_{i=0}^{t-1} \frac{\|\nabla f(x^{i+1}) - \nabla f(x^i)\|_1}{\|x^{i+1} - x^i\|_\infty}}}.$$

To compute $d^t$, it is sufficient to store only $\text{sign}\left(\nabla f(x^{t-1})\right)$, incurring no additional memory costs. Regarding $\lambda^t$, we calculate $\left\|\nabla f(x^t) - \nabla f(x^{t-1})\right\|_1$ and $\left\|x^{t+1} - x^t\right\|_\infty$ at each step. The last term does not present an issue since $\left\|x^t - x^{t-1}\right\|_\infty = \left\|\gamma^{t-1}\text{sign}\left(\nabla f(x^{t-1})\right)\right\|_\infty$. However, to find $\left\|\nabla f(x^t) - \nabla f(x^{t-1})\right\|_1$, it is necessary to store the entire gradient $\nabla f(x^{t-1})$.

We address this concern by modifying $\lambda^t$ in Algorithm 2:

$$\eta^t = \eta^{t-1} + \frac{\left\|\nabla f(x^t) - \nabla f(x^{t-1})\right\|_\infty}{\left\|x^t - x^{t-1}\right\|_1} \text{ followed by } \lambda^t = \frac{1}{\sqrt{\sum_{i=0}^{t-1} \frac{\|\nabla f(x^{i+1}) - \nabla f(x^i)\|_\infty}{\|x^{i+1} - x^i\|_1}}}. \quad (3)$$

To approximate the smoothness constant, we interchange the $l_\infty$-norm and $l_1$-norm in the expression, leveraging their duality relationship. Thus, we approximate the constant $L_1$, not $L_\infty$, as indicated in Algorithm 2. Theoretically, this approach still requires memorizing $\nabla f(x^{t-1})$. For this reason, we consider a practical option by the approximation $\left\|\nabla f(x^t) - \nabla f(x^{t-1})\right\|_\infty \approx \max\left(\left|\max_j[\nabla f(x^t)]_j - \min_j[\nabla f(x^{t-1})]_j\right|, \left|\max_j[\nabla f(x^{t-1})]_j - \min_j[\nabla f(x^t)]_j\right|\right)$. It necessitates storing only two additional constants: $\max_j\left[\nabla f(x^{t-1})\right]_j$ and $\min_j\left[\nabla f(x^{t-1})\right]_j$. In the theoretical analysis, we provide convergence guarantees only for the $\lambda^t$ choice, as in equation 3. However, we additionally validate the methods empirically with the approximation of the $l_\infty$-norm and provide an ablation study that shows a small deviation of the approximate solution from the exact one. More precisely, this approximation provides an upper bound on the initial $l_\infty$-norm, while remaining close to it (see Section 4 and Appendix A).

We present a theoretical analysis of a memory-efficient approach, utilizing an additional assumption on the $L_1$-smoothness.

**Assumption 3.11.** The function $f$ is $L_1$-smooth, i.e., it satisfies $\|\nabla f(x) - \nabla f(y)\|_\infty \leqslant L_1\|x - y\|_1$ for any $x, y \in \mathbb{R}^d$.

Now we present the convergence guarantees of Algorithm 2 with $\lambda^t$ as in equation 3.

**Theorem 3.12.** *Suppose Assumptions 3.11, 3.2, 3.3, 3.4 hold. We denote $\varepsilon \geqslant \frac{1}{T}\sum_{t=0}^{T-1}\|\nabla f(x^t)\|_1, L_1^0 = \frac{\|\nabla f(x^1) - \nabla f(x^0)\|_\infty}{\|x^1 - x^0\|_1}$. Then Algorithm 2 with $d^0 < \Delta^*$ and $d \cdot \lambda^t$ as*

*in equation 3, to reach $\varepsilon$-accuracy needs*

$$\widetilde{\mathcal{O}}\left(\frac{(\Delta^*)^2 (L_1)^3 d^2}{d^0 (L_1^0)^2 \varepsilon^2}\right) \quad and \quad \widetilde{\mathcal{O}}\left(\frac{\Delta^* (L_1)^3 d^2}{(L_1^0)^2 \varepsilon^2}\right) \quad iterations\ with\ Options\ I\ and\ II,\ respectively.$$

The result of Theorem 3.12 deteriorates the rate established in Theorem 3.5. Indeed, we can derive the $L_\infty \leqslant dL_1$ inequality. However, the proposed approach offers significant advantages in terms of memory efficiency.

Nevertheless, the theoretical convergence rates presented in this section are not optimal. In the stochastic case, we aim to achieve $\widetilde{\mathcal{O}}\left(\frac{\Delta^* L + \|\sigma\|_1^2}{\varepsilon^2}\right)$ rate. This issue is discussed in detail in Appendix B, where we present an algorithm that attains this rate.

### 3.4 ALIAS WITH MOMENTUM

In previous sections, we presented methods that do not utilize the momentum parameter (Polyak, 1987; Nesterov et al., 2018). However, many modern optimizers, such as ADAM (Kingma, 2014), PRODIGY (Mishchenko & Defazio, 2023), MUON (Jordan et al., 2024), and MARS (Yuan et al., 2024), employ this technique. We address this gap in the current section and present Algorithm 3, which incorporates the momentum parameter into Algorithm 2 in a manner similar to (Mishchenko & Defazio, 2023). Specifically, we include exponential moving averages

---

**Algorithm 3** ALIAS Adam version

1: **Input:** Start points $x^{-1}, x^0 \in \mathbb{R}^d$, $r^0, m^0, v^0 = 0$, $d^{-1} > 0$, number of iterations $T$
2: **Parameters:** $\gamma^t, \beta_1, \beta_2 > 0$
3: **for** $t = 0, \ldots, T-1$ **do**
4: $\quad r^{t+1} = \sqrt{\beta_2} r^t + \left(1 - \sqrt{\beta_2}\right) d^{t-1} \langle g_{\xi^t}^t, \mathrm{sign}(g_{\xi^{t-1}}^{t-1}) \rangle$
5: $\quad d^t = \max\left\{d^{t-1}, r^{t+1}\right\}$
6: $\quad m^{t+1} = \beta_1 m^t + (1 - \beta_1) d^t g_{\xi^t}^t$
7: $\quad v^{t+1} = \beta_2 v^t + (1 - \beta_2)\left(d^t\right)^2 \left(g_{\xi^t}^t\right)^2$
8: $\quad x^{t+1} = x^t - \gamma^t \sqrt{\frac{(d^t)^2}{1 + \frac{v^{t+1} - (m^{t+1})^2}{(m^{t+1})^2}}} \odot \mathrm{sign}(m^{t+1})$
9: **end for**

---

of the first and second statistics, as in ADAM to aggregate past gradients and provide coordinate-wise normalization that mitigates sharp directions and gradient noise.

## 4 EXPERIMENTS

In this section, we present empirical results for the LLM pre-training task. In Appendix A, we validate our approach on vision tasks, specifically by fine-tuning the SWIN Transformer architecture (Liu et al., 2021). Our code is open-sourced[1].

**Language model pre-training.** Following the protocol of (Lialin et al., 2023), we train a LLaMA-based architecture (Touvron et al., 2023) with 130M parameters on the C4 dataset (Raffel et al., 2020). A detailed description of the experimental setup is provided in Appendix A.1. We compare several optimization methods: SIGN-SGD with a tuned constant learning rate (lr), and three methods using a tuned learning rate with a cosine scheduler (cosine sc) – namely, SIGN-SGD, STEEPEST DESCENT, and NORMALIZED SGD. All of these methods are compared against ALIAS (Algorithm 2), which is used without any tuning. Additionally, we evaluate all methods with weight decay (wd). We provide final validation loss and perplexity in Table 1.

In Table 2, we present the results for methods incorporating momentum ($\beta$) (all methods with weight decay). ALIAS Adam version utilizes sign descent with momentum and an additional scaling factor (see Algorithm 3 for details). We consider two options for this method: with and without a cosine scheduler. We provide a comparison with ADAMW (Loshchilov, 2017) and PRODIGY (Mishchenko & Defazio, 2023). We test PRODIGY with and without a learning rate scheduler. We present the pre-training dynamic in Figure 1. These results coincides with those in Tables 1, 2.

---

[1]https://github.com/brain-lab-research/ALIAS

Table 1: SIGN methods on LLAMA pre-training.

| Algorithm | Validation Loss (↓) | Perplexity (↓) |
|---|---|---|
| SIGN-SGD (lr) | 3.041 | 20.923 |
| SIGN-SGD (lr, cosine sc) | 2.992 | 19.923 |
| STEEPEST DESCENT (lr, cosine sc) | 3.035 | 20.791 |
| NORMALIZED SGD (lr, cosine sc) | 3.135 | 22.982 |
| ALIAS (ours) | 3.017 | 20.422 |
| SIGN-SGD (wd, lr) | 3.041 | 20.923 |
| SIGN-SGD (wd, lr, cosine sc) | 2.980 | 19.693 |
| STEEPEST DESCENT (wd, lr, cosine sc) | 3.022 | 20.537 |
| NORMALIZED SGD (wd, lr, cosine sc) | 3.006 | 20.169 |
| ALIAS (wd) (ours) | 3.006 | 20.169 |

Table 2: SIGN-SGD methods with added momentum parameter ($\beta$), ADAMW and PRODIGY on LLAMA pre-training.

| Algorithm | Validation Loss (↓) | Perplexity (↓) |
|---|---|---|
| SIGN-SGD (wd, $\beta$, lr) | 2.968 | 19.459 |
| SIGN-SGD (wd, $\beta$, lr, cosine sc) | 2.923 | 18.596 |
| STEEPEST DESC. (wd, $\beta$, lr, cosine sc) | 2.932 | 18.765 |
| NORM. SGD (wd, $\beta$, lr, cosine sc) | 2.934 | 18.803 |
| ADAMW (wd, $\beta$, lr, cosine sc) | 2.929 | 18.698 |
| PRODIGY (wd, $\beta$) | 3.003 | 20.145 |
| PRODIGY (wd, $\beta$, cosine sc) | 2.930 | 18.727 |
| ALIAS Adam version (wd, $\beta$) (ours) | 2.976 | 19.609 |
| ALIAS Adam version (wd, $\beta$, cosine sc) (ours) | 2.918 | 18.504 |

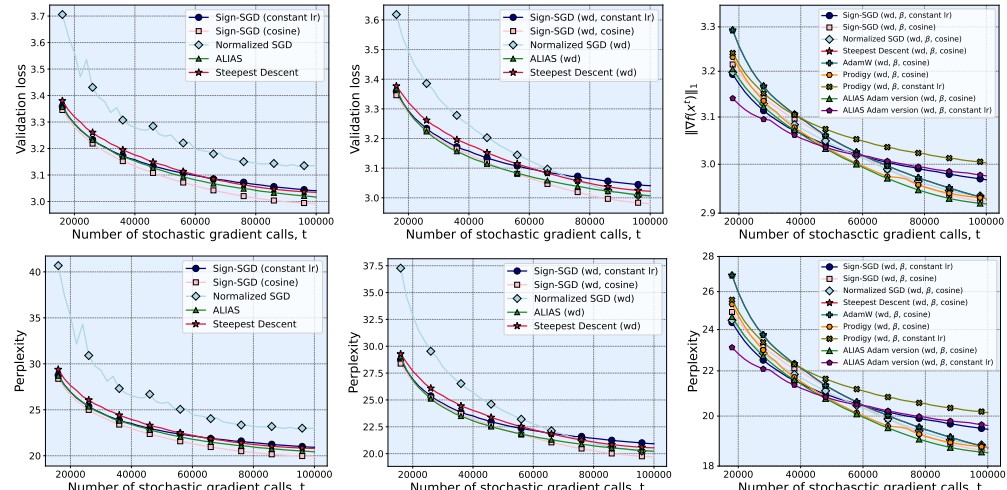

Figure 1: Comparison of SIGN-SGD methods in LLAMA pre-training. The left column shows results without weight decay, the central column presents results with weight decay (wd), and the right column displays results with weight decay (wd) and momentum parameter ($\beta$).

We highlight that our basic ALIAS achieves performance only slightly inferior to that of SIGN-SGD with a tuned cosine scheduler. The Adam-based version of ALIAS outperforms all competing methods, including tuned ADAMW and the state-of-the-art parameter-free optimizer PRODIGY with a tuned cosine scheduler. These results are particularly competitive given that our approach eliminates the need for learning rate tuning – a significant practical advantage. This feature enhances the method's usability, making it appealing for large-scale applications.

**Memory-efficient version of Algorithm 2.** We proceed with testing the memory-efficient approach, presented in Section 3.3. Recall that we approximate $\left\|\nabla f(x^t) - \nabla f(x^{t-1})\right\|_\infty \approx \max\left(\left|\max_j[\nabla f(x^t)]_j - \min_j[\nabla f(x^{t-1})]_j\right|, \left|\max_j[\nabla f(x^{t-1})]_j - \min_j[\nabla f(x^t)]_j\right|\right)$. We compare the performance of ALIAS with $\lambda^t$ as in equation 3, considering exact and approximated $l_\infty$-norm (me), SIGN-SGD with a constant (tuned) stepsize, and SIGN-SGD with a (tuned) cosine scheduler. The results of the 130M LLAMA-based model pre-training are presented in Table 3. We provide an ablation comparing exact and approximated values of $l_\infty$-norms during training in Appendix A.

The results indicate a slight performance degradation of the memory-efficient version of ALIAS compared to SIGN-SGD with a cosine scheduler baseline, as well as relative to the original ALIAS method. However, it is crucial to emphasize that this variant is a parameter-free algorithm that retains only the sign of the gradient from the previous iteration. Despite these

Table 3: SIGN-SGD methods and memory-efficient version of ALIAS on LLAMA pre-training.

| Algorithm | Validation Loss (↓) | Perplexity (↓) |
|---|---|---|
| SIGN-SGD (wd, lr) | 3.041 | 20.923 |
| SIGN-SGD (wd, lr, cosine sc) | 2.980 | 19.693 |
| ALIAS (wd, $\lambda^t$ as in equation 3) (ours) | 3.015 | 20.389 |
| ALIAS (wd, $\lambda^t$ as in equation 3, me) (ours) | 3.019 | 20.471 |

simplifications, its performance remains competitive with significantly more memory-intensive methods. We report performance metrics, memory footprint, and runtime efficiency in Appendix A, along with detailed training configurations for full reproducibility.

## 5 CONCLUSION

In this work, we present a novel parameter-free SIGN-SGD that eliminates manual stepsize selection. The method is analyzed in deterministic, stochastic, and distributed settings. Additionally, we introduce a memory-efficient variant that stores only gradient signs while maintaining adaptivity. We also explore a momentum-adapted version that demonstrates strong performance in practice.

## ACKNOWLEDGMENTS

The work was supported by the Ministry of Economic Development of the Russian Federation (agreement No. 139-15-2025-013, dated June 20, 2025, IGK 000000C313925P4B0002).

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

CONTENTS

## A  ADDITIONAL EXPERIMENTS

This section supplements our experimental validation by examining the internal mechanisms of parameter-free sign-based optimizers across LLaMA pre-training and Tiny ImageNet classification. We analyze how step-size dynamics naturally emerge without manual scheduling, investigate memory consumption and computational time compared to established optimizers, and demonstrate robustness to hyperparameter choices.

### A.1  LLaMA PRE-TRAINING

#### A.1.1  EXPERIMENTAL SETUP.

Our experiments use a LLaMA-based architecture (Touvron et al., 2023) equipped with RMSNorm and SwiGLU (Shazeer, 2020) activations, trained on the C4 dataset (Raffel et al., 2020). The training consists of 100k steps. We use batch size of 512 sequences and sequence length of 256, as in Lialin et al. (2023), and T5 tokenizer with the dictionary size of 32k since it was originally trained on C4.

For all experiments, the respective optimization method is applied to the main model parameters, while the LM Head layer is optimized with AdamW. This design follows prior work Zhao et al. (2024) which showed that the LM Head layer requires more fine-grained learning rate adjustment.

The learning rate was selected through a grid search with multiplicative step of $10^{\frac{1}{4}}$. We employ a cosine learning rate schedule with a warmup of 10% of the total steps and decay to 10% of the peak learning rate. For ALIAS Adam version (Algorithm 3), we choose stepsize $\gamma^t = 10^{-3}$.

The weight decay value was selected from [0, 0.01, 0.1] through validation. We also applied gradient clipping with threshold of 1.0 for all methods except STEEPEST DESCENT and NORMALIZED SGD. All methods with momentum utilize the Nesterov acceleration scheme with a momentum value of 0.9. For AdamW we use the standard hyperparameters: $\beta_1 = 0.9, \beta_2 = 0.999, \varepsilon = 1e - 8$.

#### A.1.2  ADDITIONAL RESULTS

In this section, we explore key aspects of our method. We analyze the stepsize derived from our approach and compare it to the effective learning rate induced by the cosine scheduler. Next, we examine the memory and computational efficiency of all considered optimizers. We present an ablation study on the approximation used in the stepsize of the memory-efficient variant, demonstrating its close alignment with exact computation. We provide empirical evidence for the robustness of ALIAS to an additional constant $L_\infty$ term (see Remark 3.6). Finally, we discuss the question regarding the performance dependence on the choice of the initial value $d^0$ and the level of gradient noise.

**Study on the stepsize.**  A question arises regarding how $\gamma^t \sqrt{\frac{(d^t)^2}{1 + \frac{v^{t+1} - (m^{t+1})^2}{(m^{t+1})^2}}}$ performs compared to the effective cosine scheduler when $\gamma^t$ remains constant. This pairing is presented in Figure 2.

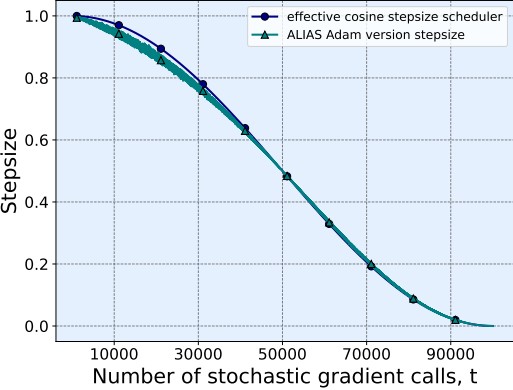

Figure 2: Comparison of ALIAS Adam version stepsize with constant $\gamma^t$ with effective cosine stepsize scheduler.

One can state that the cosine nature of the stepsize is automatically obtained. This feature highlights the distinctiveness of our parameter-free approach.

**Study on the time and memory consumption.** In Table 4, we present details of memory requirements and time consumption per-iteration.

Table 4: Comparison of memory and time consumption.

| Algorithm | Memory consumption (gb) | Time consumption per-iteration (s) |
|---|---|---|
| SIGN-SGD | 0.41 | 0.004 |
| STEEPEST DESCENT | 0.41 | 0.01 |
| NORMALIZED SGD | 0.41 | 0.01 |
| ADAMW | 1.5 | 0.007 |
| PRODIGY | 3.5 | 0.05 |
| ALIAS (**ours**) | 1.22 | 0.01 |
| ALIAS Adam version (**ours**) | 1.91 | 0.03 |
| memory-efficient ALIAS (**ours**) | 0.41 | 0.007 |

Table 4 shows a higher time per-iteration for ALIAS Adam version and PRODIGY, which we adopt from the work (Mishchenko & Defazio, 2023). We attribute this to the suboptimal implementation of these algorithms, in contrast to others that have been utilized for an extended period. Simultaneously, our algorithms are comparable to ADAMW in terms of required memory, while PRODIGY occupies more GPU resources because it stores a vector of initial model parameters. Note that the memory-efficient version of ALIAS is superior to ADAMW and comparable to the basic SIGN-SGD.

**Study on the memory-efficient ALIAS.** We now analyze the memory-efficient variant of ALIAS, focusing on the accuracy of the approximated $l_\infty$-norm used in its update rule. Figure 3 shows the dynamics of $\left\|\nabla f(x^t) - \nabla f(x^{t-1})\right\|_\infty$ across iterations, along with the deviation range of its approximation (see Section 3.3 for details on the approximation scheme). The ablation study reveals that the approximate norm deviates from the exact value by approximately 50% on average. Notably, the approximation consistently exceeds the true norm – as expected, since it constitutes an upper bound by design.

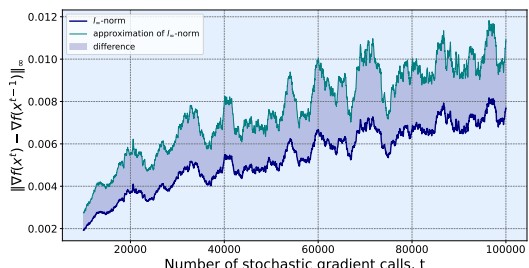

Figure 3: Ablation study on approximated $l_\infty$-norm deviation from the exact one in the memory-efficient version of ALIAS.

This leads to smaller effective stepsizes, which explains the slightly degraded performance of the memory-efficient variant compared to the basic ALIAS algorithm (Algorithm 2).

**Study on the robustness to $L_\infty$.** In Table 5, we provide empirical evidence supporting the claim made in Section 3.1 that the modification of ALIAS (Algorithm 2) is robust concerning the $L_\infty$ parameter. Hence, although the version of the algorithm considered in Remark 3.6 requires prior knowledge of $L_\infty$, this additive factor has negligible impact on the practical convergence of Algorithm 2.

Table 5: Robustness to $L_\infty$.

| $L_\infty$ value | Validation loss ($\downarrow$) |
|---|---|
| 0 | 3.006 |
| 50 | 3.006 |
| 100 | 3.007 |
| 500 | 3.005 |
| 1000 | 3.006 |

**Performance dependence on $d^0$ choice.** In this paragraph, we investigate the robustness of our ALIAS Adam version concerning the choice of the initial distance $d^0$. To this end, we compare the performance of Algorithm 3 on LLAMA pre-training using $d^0 = 1$ and $d^0 = 10^{-3}$. In both cases, we obtain the **same validation metric**: **validation loss = 2.918**. Based on these results, we conclude that our method is insensitive to the choice of $d^0$.

**Performance dependence on gradient noise.** We investigate the dependence of the performance of our ALIAS procedure (Algorithm 2) on the level of gradient noise. To simulate different noise levels, we vary the batch size. Indeed, decreasing the batch size increases the stochasticity and the variance of the gradient estimate, thereby leading to a higher level of gradient noise. While in previous experiments we used a batch size of 512 sequences, here we use 256, 128, and 64 sequences. Then we compare the validation loss on these runs. Table 6 provides a pairwise comparison of ALIAS and SIGN-SGD across these batch sizes.

Table 6: SIGN-SGD and ALIAS with different bath sizes on LLAMA pre-training.

| Batch Size (# of Sequences) | Algorithm | Validation Loss ($\downarrow$) |
|---|---|---|
| 512 | SIGN-SGD | 2.980 |
| 512 | ALIAS (**ours**) | 3.006 |
| 256 | SIGN-SGD | 2.986 |
| 256 | ALIAS (**ours**) | 3.013 |
| 128 | SIGN-SGD | 2.992 |
| 128 | ALIAS (**ours**) | 3.021 |
| 64 | SIGN-SGD | 2.999 |
| 64 | ALIAS (**ours**) | 3.029 |

The experimental results demonstrate that, when the batch size is reduced – thereby increasing the level of gradient noise – both SIGN-SGD and ALIAS exhibit a comparable decline in performance. This suggests that ALIAS is not disproportionately affected by the increased stochasticity in gradient estimates, underscoring its robustness to gradient noise.

### A.1.3 COMPARISON WITH PARAMETER-FREE APPROACHES

In this section, we present an experimental comparison of our ALIAS Adam version algorithm with competing parameter-free optimization methods. For this additional evaluation, we selected the following approaches: DoG (Ivgi et al., 2023), D-ADAPTATION (Defazio & Mishchenko, 2023), and MoMo (Schaipp et al., 2023). These methods are chosen based on their performance reported in the work (Kasimbeg et al., 2025) on the ALGOPERF benchmark (Dahl et al., 2023). Our validation results for pre-training the LLAMA-based architecture are summarized in Table 7.

Table 7: Parameter-free methods on LLAMA pre-training.

| Algorithm | Validation Loss ($\downarrow$) | Perplexity ($\downarrow$) |
|---|---|---|
| DoG | 2.939 | 18.897 |
| D-ADAPTATION (with Adam) | 2.927 | 18.672 |
| MoMo (with Adam) | 2.925 | 18.634 |
| PRODIGY | 2.930 | 18.727 |
| ALIAS Adam version (wd) (**ours**) | 2.918 | 18.504 |

These results complement our comparison against sign-based methods and ADAMW. They demonstrate that our approach achieves stronger performance than prior parameter-free methods.

### A.1.4 EXPERIMENTS ON BIG MODEL

We evaluate the methods on LLAMA with 350M parameters. The training setup remains consistent with the previous experiment. However, the number of layers in the model increases, leading to a total parameter count that rises from 130M to 350M. This experiment is essential to demonstrate the sustainability of our approaches to increasing dimensionality. We conduct experiments comparing methods with and without the momentum parameter $\beta$ along with weight decay. The results are presented Tables 8, 9, and Figure 4.

Table 8: SIGN-SGD methods on LLAMA pre-training.

| Algorithm | Validation Loss ($\downarrow$) | Perplexity ($\downarrow$) |
|---|---|---|
| SIGN-SGD (wd, lr, cosine sc) | 2.819 | 16.760 |
| STEEPEST DESCENT (wd, lr, cosine sc) | 2.828 | 16.912 |
| NORMALIZED SGD (wd, lr, cosine sc) | 3.510 | 33.448 |
| ALIAS (wd) (**ours**) | 2.821 | 16.793 |

Table 9: SIGN-SGD methods with added momentum parameter ($\beta$), ADAMW (wd) and PRODIGY on LLAMA pre-training.

| Algorithm | Validation Loss ($\downarrow$) | Perplexity ($\downarrow$) |
|---|---|---|
| SIGN-SGD (wd, $\beta$, lr, cosine sc) | 2.717 | 15.135 |
| STEEPEST DESCENT (wd, $\beta$, lr, cosine sc) | 2.711 | 15.044 |
| NORMALIZED SGD (wd, $\beta$, lr, cosine sc) | 3.460 | 31.817 |
| ADAMW (wd, $\beta$, lr, cosine sc) | 2.719 | 15.165 |
| PRODIGY (wd, $\beta$, cosine sc) | 2.715 | 15.105 |
| ALIAS Adam version (wd, $\beta$, cosine sc) (**ours**) | 2.707 | 14.984 |

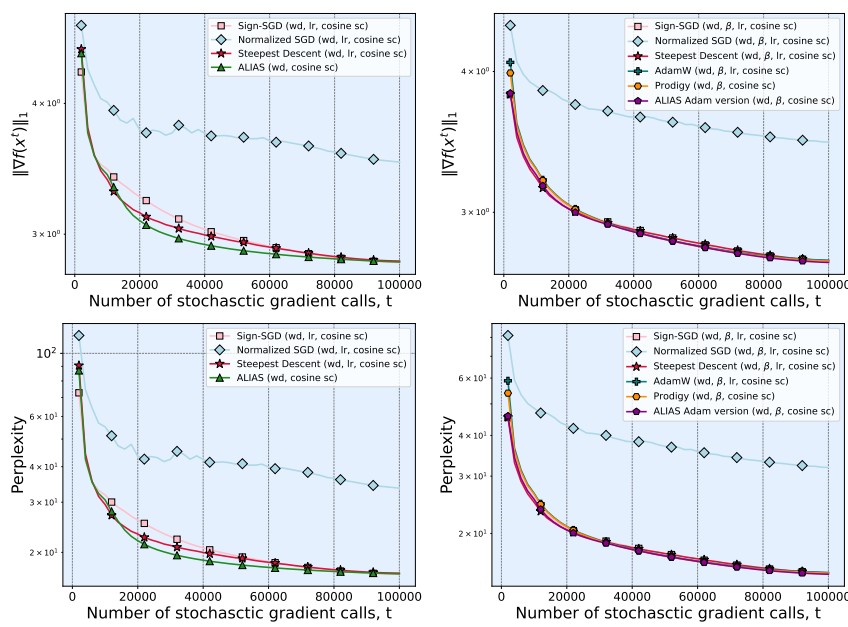

Figure 4: Comparison of SIGN-SGD methods on 350M parameters LLAMA pre-training. Left column is results for methods with weight decay and without momentum parameter $\beta$, right column – methods with momentum $\beta$.

The results are consistent with those obtained for the smaller model. Among the momentum-based methods, ALIAS Adam version demonstrates the best performance, while among the methods without momentum, ALIAS exhibits comparable performance to other solutions.

### A.1.5 COMPUTE RESOURCES.

We conducted all experiments described in Section A.1 on the cluster equipped with 4×NVIDIA A100 GPUs. A complete run of 100,000 steps took approximately 12 hours using a full node.

## A.2    TINY IMAGENET CLASSIFICATION WITH SWIN TRANSFORMER FINE-TUNING

### A.2.1    EXPERIMENTAL SETUP

Our image classification experiments on the Tiny ImageNet dataset (Le & Yang, 2015) employed the Tiny Swin Transformer architecture (Liu et al., 2021). This lightweight variant of the Swin Transformer is characterized by its hierarchical design and the use of shifted windows for efficient self-attention computation. The specific configuration utilized involved non-overlapping $4 \times 4$ input patches and a $7 \times 7$ window size for local self-attention.

We initialized the model using pretrained weights from ImageNet-1K (Deng et al., 2009), specifically the `swin_T_patch4_window7_224` checkpoint provided in the official Swin Transformer repository[2]. The model was then fine-tuned on Tiny ImageNet.

The Tiny ImageNet dataset comprises 200 classes with images of $64 \times 64$ resolution. To meet the model's input requirements, all images were upsampled to $224 \times 224$. A standard ImageNet-style data augmentation pipeline was implemented, including random resized cropping and horizontal flipping.

Training spanned 50 epochs, with a batch size of 256. The learning rate was determined via a grid search, employing a multiplicative step of $10^{\frac{1}{4}}$. A cosine learning rate schedule was adopted, featuring a linear warm-up phase for the initial 10% of total training steps, followed by decay to 10% of the peak learning rate. Weight decay was selected from $\{0, 0.01, 0.1\}$ based on validation performance. All optimization methods incorporated gradient clipping with a threshold of 1.0. When momentum was applied, Nesterov acceleration with a coefficient of 0.99 was used. For AdamW, the standard configuration of $\beta_1 = 0.9$, $\beta_2 = 0.999$, and $\varepsilon = 10^{-8}$ was maintained.

### A.2.2    PERFORMANCE ON IMAGE CLASSIFICATION

Further results and training curves for the Tiny Swin Transformer on the Tiny ImageNet classification task are presented in Figure 5 and Table 10. We provide plots for the same methods with the incorporated momentum parameter as for the LLaMA pre-training task.

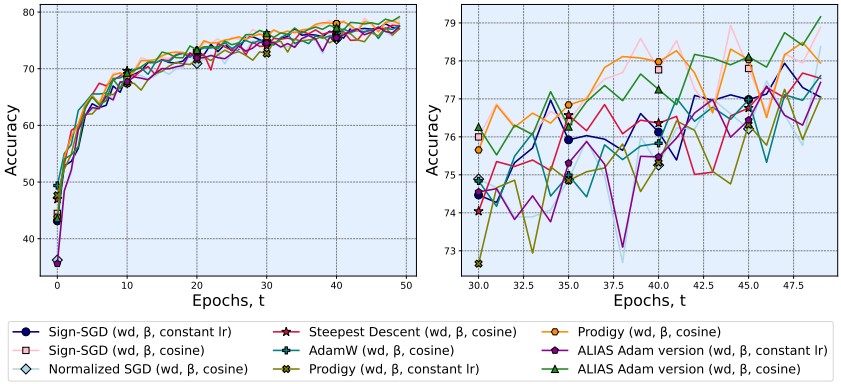

Figure 5: SIGN-SGD methods with added momentum parameter $(\beta)$, ADAMW (wd) and PRODIGY on SWIN fine-tuning. Left plot represents full process of training, right plot demonstrates accuracy on last 20 epoch.

The results demonstrate the superiority of our algorithms over both tuned sign-based methods and advanced optimizers, such as PRODIGY and ADAMW.

### A.2.3    COMPUTE RESOURCES

We conducted all experiments described in Section A.2 using a single NVIDIA A100 GPU. A complete run of 50 epochs required approximately 3 hours.

---

[2]`https://github.com/microsoft/Swin-Transformer/blob/main/MODELHUB.md`

Table 10: Final accuracy of SIGN-SGD methods with added momentum parameter ($\beta$), ADAMW (wd) and PRODIGY on SWIN fine-tuning.

| Algorithm | Final accuracy ($\uparrow$) |
|---|---|
| SIGN-SGD (wd, $\beta$, lr) | 77.045 |
| SIGN-SGD (wd, $\beta$, lr, cosine sc) | 78.885 |
| NORMALIZED SGD (wd, $\beta$, lr, cosine sc) | 78.375 |
| STEEPEST DESCNET (wd, $\beta$, lr, cosine sc) | 77.547 |
| ADAMW (wd, $\beta$, lr, cosine sc) | 77.612 |
| PRODIGY (wd, $\beta$) | 77.035 |
| PRODIGY (wd, $\beta$, cosine sc) | 77.944 |
| ALIAS Adam version (wd, $\beta$) (**ours**) | 77.433 |
| ALIAS Adam version (wd, $\beta$, cosine sc) (**ours**) | 79.161 |

### A.3 ALGOPERF BENCHMARK

In this section, we evaluate our method on some tasks from the ALGOPERF benchmark (Dahl et al., 2023). To test our approach across different modalities, we chose the MRI reconstruction and molecular property prediction (MPP) tasks. We preserve the original setups from the benchmark implementation[3]. Specifically, for the MRI reconstruction task, we use the fastMRI dataset and a U-Net model; for molecular property prediction, we utilize the OGBG dataset with a GNN model. We validate only our ALIAS Adam version algorithm. The results for the other methods are taken from Table 4 in (Kasimbeg et al., 2025), which reports comparisons between parameter-free optimizers and tuned ADAMW. For our method, we fix $\gamma^t = 10^{-3}$. The results are presented in Table 11.

Table 11: Parameter-free methods and ADAMW on MRI reconstruction and molecular property prediction tasks.

| Algorithm | MRI, SSIM ($\uparrow$) | MPP, mAP ($\uparrow$) |
|---|---|---|
| ADAMW | 0.723 | 0.254 |
| DOG | 0.714 | 0.231 |
| D-ADAPTATION (with Adam) | 0.722 | 0.221 |
| MOMO (with Adam) | 0.723 | 0.221 |
| PRODIGY | 0.723 | 0.212 |
| ALIAS Adam version (wd) (**ours**) | 0.724 | 0.242 |

The results demonstrate that our approach improves upon the metrics of prior parameter-free methods on the evaluated tasks. We also note that, for the MRI reconstruction task, the performance of our method surpasses that of the tuned ADAMW.

## B SIGN-SGD WITH ADDITIONAL STEPSIZE SEARCH PROCEDURE

In this section, we present an algorithm that achieves near-optimal convergence rates for SIGN-SGD – $\widetilde{\mathcal{O}}\left(\frac{\Delta^* L_\infty}{\varepsilon^2}\right)$ in the deterministic case, and $\mathcal{O}\left(\frac{\Delta^* L_\infty}{\varepsilon^2} + \|\sigma\|_1^2\right)$ in the stochastic case. The method does not utilize prior knowledge about the parameters of the problem and incorporates an additional automatic stepsize search procedure.

**Exact gradients.** To design the necessary algorithm, we should provide a stepsize $\gamma$ in Algorithm 1 that yields an estimate as in equation 1. Let us begin with the description of the approximation of the stepsize 1 that we utilize. We establish that the desired value is $\gamma = \frac{\mathfrak{N}_T}{\mathfrak{D}_T}$, where $\mathfrak{N}_T = \widetilde{\Delta}_T = f(x^0) - \min_{0 \leqslant t < T} f(x^t)$ is the numerator and $\mathfrak{D}_T = \sum_{t=0}^{T-1} \|\nabla f(x^{t+1}) - \nabla f(x^t)\|_1$

---

[3]https://github.com/mlcommons/algorithmic-efficiency/blob/main/docs/
GETTING_STARTED.md

is the denominator. The intuition behind this choice is that due to $L_\infty$-smoothness, we have $\mathfrak{D}_T \sim L_\infty \sum_{t=0}^{T-1} \left\| x^{t+1} - x^t \right\|_\infty = \gamma L_\infty \sum_{t=0}^{T-1} \left\| \text{sign} \left( \nabla f(x^t) \right) \right\|_\infty = \gamma L_\infty T$; then $\gamma$ has $\frac{\sqrt{\widetilde{\Delta}_T}}{\sqrt{L_\infty T}}$ magnitude. However, we face a more complex situation compared to the regret minimization paradigm: in our case, $\widetilde{\Delta}_T$ can be non-negative (in regret minimization, the analog of $\Delta_T$ is the norm of the points' difference $\left\| x^0 - x^T \right\|$ (Carmon & Hinder, 2022) which is always positive). To address this, we add an extra step to the SIGN-SGD algorithm. Define $e = \text{sign} \left( \nabla f(x^{-1}) \right)$. Let $\tau$ be a small parameter. The update is:

$$f(x^0) = \min \left\{ f(x^{-1} + \tau e), f(x^{-1} - \tau e) \right\}, \tag{4}$$

The rationale behind selecting the step is as follows. Due to the smoothness of the objective function, there exists a small neighborhood around any point within which moving in any direction decreases the objective value. The exception arises when $x^{-1}$ is the minimum itself. In this case, the sign descent algorithm would not take any steps, and we return this point as the solution. Since the neighborhood size $\tau$ depends on $L_\infty$, we iteratively decrease $\tau$ until it is sufficiently small. The choice of $\tau$ and the guarantee $f(x^0) < f(x^{-1})$ are discussed in Lemma D.4. In this manner, we ensure that $\mathfrak{N}_T = \widetilde{\Delta}_T = f(x^{-1}) - \min_{-1 \leqslant t \leqslant T} f(x^t) > 0$. To prevent the denominator from being zero, we introduce a small constant $\zeta$, which represents the minimum gradient norm encountered during learning. This leads to $\mathfrak{D}_T = \sum_{t=0}^{T-1} \| \nabla f(x^{t+1}) - \nabla f(x^t) \|_1 + \zeta$ (see Lemma D.2 for details). However, determining these values necessitates completing all $T$ iterations. To address this, we employ the BISECTION procedure from (Carmon & Hinder, 2022), which is outlined in Algorithm 4.

Our goal is to have $\gamma = \phi(\gamma) = \frac{\mathfrak{N}_T(\gamma)}{\mathfrak{D}_T(\gamma)}$. To find such $\gamma$, we take an initial interval $[\gamma_{\text{lo}}, \gamma_{\text{hi}}]$ and, iteratively narrowing it, obtain a small enough interval $[\gamma_{\text{lo}}^*, \gamma_{\text{hi}}^*]$ that contains the $\gamma - \phi(\gamma) = 0$ point. To perform this, we firstly have to make sure that the initial interval contains the desired point. For this purpose, we require $\gamma_{\text{hi}} > \phi(\gamma_{\text{hi}})$ and $\gamma_{\text{lo}} < \phi(\gamma_{\text{lo}})$. We designate the group of these two requirements as the bisection start condition (Lines 3, 5). Note that we can always satisfy the first condition, as shown in Lemma D.2. Regarding the second requirement, we can choose a sufficiently small initial $\gamma_{\text{lo}}$ value. Even if $\gamma_{\text{lo}}$ is still greater than $\phi(\gamma_{\text{lo}})$, we can select this $\gamma_{\text{lo}}$ value as the desired stepsize without performing the BISECTION procedure, thereby obtaining optimal convergence guarantees. This is

---

**Algorithm 4** BISECTION procedure

1: **Input:** Optimal stepsize value $\phi(\gamma)$, lower stepsize bound $\gamma_{\text{lo}}$, upper stepsize bound $\gamma_{\text{hi}}$, $x^{-1} \in \mathbb{R}^d$, number of iterations $T$
2: $\phi(\gamma)$ (it is always in the form $\phi(\gamma) = \frac{\mathfrak{N}_T(\gamma)}{\mathfrak{D}_T(\gamma)}$)
3: **if** $\gamma_{\text{hi}} \leqslant \phi(\gamma_{\text{hi}})$ **then return** $\infty$  *// Early infinite termination*
4: **end if**
5: **if** $\gamma_{\text{lo}} > \phi(\gamma_{\text{lo}})$ **then return** $\gamma_{\text{lo}}^* = \gamma_{\text{lo}}$  *// Early non-infinite termination*
6: **end if**
7: **while** $\gamma_{\text{hi}} > 2\gamma_{\text{lo}}$ **do**
8:     $\gamma_{\text{mid}} = \sqrt{\gamma_{\text{lo}}\gamma_{\text{hi}}}$
9:     $\mathfrak{N}_T(\gamma_{\text{mid}}), \mathfrak{D}_T(\gamma_{\text{mid}}) \leftarrow$ SIGN-SGD$(x^{-1}, T, \gamma_{\text{mid}})$  *// First step in Sign-SGD is made by equation 4*
10:     **if** $\gamma_{\text{mid}} \leqslant \phi(\gamma_{\text{mid}})$ **then**
11:         $\gamma_{\text{lo}} = \gamma_{\text{mid}}$
12:     **else**
13:         $\gamma_{\text{hi}} = \gamma_{\text{mid}}$
14:     **end if** *// Bisection invariants: $\gamma_{lo} < \phi(\gamma_{lo})$, $\gamma_{hi} > \phi(\gamma_{hi})$*
15: **end while** *// Bisection stop condition: $\gamma_{hi} \leqslant 2\gamma_{lo}$*
16: **if** $\mathfrak{N}_T(\gamma_{\text{hi}}) \leqslant \mathfrak{N}_T(\gamma_{\text{lo}}) \frac{\phi(\gamma_{hi})}{\gamma_{hi}}$ **then return** $\gamma_{\text{hi}}^* = \gamma_{\text{hi}}$  *// $\gamma_{hi}$ return condition*
17: **elsereturn** $\gamma_{\text{lo}}^* = \gamma_{\text{lo}}$  *// $\gamma_{lo}$ return condition*
18: **end if**

---

**Algorithm 5** SOS SIGN-SGD

1: **Input:** Initial stepsize bound $\gamma_s$, initial bound step $k$, start point $x^{-1} \in \mathbb{R}^d$, number of iterations $T$
2: $\gamma_0 = $ BISECTION $\left( \phi(\gamma), \gamma_s, 2^{2^k}\gamma_s, T \right)$
3: $x^T = $ SIGN-SGD$(x^{-1}, T, \gamma_0)$

---

demonstrated in **Step 2** of the proof of Theorem B.1 (Theorem E.2). This enables us to avoid early infinite termination (non-compliance with the first condition) and prevents convergence from being compromised by early non-infinite termination (non-compliance with the second condition). Additionally, we ensure that, by entering the procedure with the desired point between $\gamma_{\text{lo}}$ and $\gamma_{\text{hi}}$, it remains invariant throughout the procedure. Indeed, at each iteration we compute $\gamma_{\text{mid}}$ as the geometric average of the bounds and perform $T$ iterations of the SIGN-SGD method with this

stepsize to find $\phi(\gamma_{\mathrm{mid}})$ (Lines 8, 9). It remains for us to choose such a part of the segment ($[\gamma_{\mathrm{lo}}, \gamma_{\mathrm{mid}}]$ or $[\gamma_{\mathrm{mid}}, \gamma_{\mathrm{hi}}]$) in which $\phi(\gamma_{\mathrm{mid}})$ lies (Lines 10 - 14). We perform this bisection, until $\gamma_{\mathrm{hi}}$ exceeds $\gamma_{\mathrm{lo}}$ by more than 2 times (Line 7). In the end, by utilizing return conditions, the procedure returns $\gamma_{\mathrm{lo}}^*$ or $\gamma_{\mathrm{hi}}^*$ (Lines 16 - 18). They satisfy the specific bounds explored in Lemma D.3.

Using this procedure, we present a description of the SOS (Search of the Optimal Stepsize) SIGN-SGD (Algorithm 5). Before we pass to the convergence rate, we discuss the number of iterations required by Algorithm 4. Since we calculate the average geometric at each iteration, we need $\log\log\frac{\gamma_{\mathrm{hi}}}{\gamma_{\mathrm{lo}}}$ steps, where $\gamma_{\mathrm{lo}}$ and $\gamma_{\mathrm{hi}}$ are the boundaries of the initial segment. Thus, according to Algorithm 5, it requires $\log\log\frac{2^{2^k}\gamma_s}{\gamma_s} = k$ iterations. We establish a lower bound on $k$ by requiring that the initial $\gamma_{\mathrm{hi}}$ is greater than $\phi(\gamma_{\mathrm{hi}})$. According to Lemma D.2, $\gamma_{\mathrm{hi}}$ should be at least $\frac{\Delta^*}{\|\nabla f(x^0)\|_1}$. In this way, $k = \log\log\frac{\Delta^*}{\gamma_s\|\nabla f(x^0)\|_1}$. Therefore, allowing Algorithm 5 to perform $T$ iterations, the total number of iterations (considering Algorithm 4 performance time) is $T\log\log\frac{\Delta^*}{\gamma_s\|\nabla f(x^0)\|_1}$. We regard this additional double-logarithmic factor as negligible, as it aligns with the results in (Carmon & Hinder, 2022). We now present the main theoretical result of this section.

**Theorem B.1.** *Suppose Assumptions 3.1, 3.2, 3.3, 3.4 hold. Then for Algorithm 5 after obtaining the stepsize $\gamma_0$ the following estimate is valid:*

$$\frac{1}{T}\sum_{t=0}^{T-1}\|\nabla f(x^t)\|_1 \leqslant 6\frac{\sqrt{\Delta^* L_\infty}}{\sqrt{T}} + \frac{3\|\nabla f(x^0)\|_1}{T}.$$

*Moreover, taking into account the complexity of Algorithm 4 in relation to the initial stepsize bound $\gamma_s$, to reach $\varepsilon$-accuracy, where $\varepsilon \geqslant \frac{1}{T}\sum_{t=0}^{T-1}\|\nabla f(x^t)\|_1$, Algorithm 5 needs*

$$\widetilde{\mathcal{O}}\left(\frac{\Delta^* L_\infty}{\varepsilon^2}\right) \quad \text{iterations.}$$

**Discussion of the results.** We obtain the near-optimal convergence rate 1. Our method retains a dependency on the initial approximation. Indeed, we should take $\gamma_s$ to be less than $\frac{\Delta^*}{L_\infty T}$, according to **Step 2** in the proof of Theorem B.1 (Theorem E.2). An analogous requirement was established in the work (Carmon & Hinder, 2022) and we do not consider this to be an issue. Nevertheless, despite the theoretical optimality of the proposed approach, its practical application is not promising. Launching multiple training sessions on large models does not appear effective. In this context, Algorithm 2 remains our main contribution. While proposing Algorithm 5, we demonstrate how to obtain the near-optimal rate for SIGN-SGD in parameter-free optimization.

**Stochastic gradients and distributed settings.** The description and analysis of Algorithm 5 in stochastic and distributed setups can be found in Appendix E.2, E.3.

### B.1 SOS SIGN-SGD EXPERIMENTS

#### B.1.1 LOGISTIC REGRESSION.

We present toy experiments on logistic regression. We provide a comparison of SIGN-SGD with the theoretical stepsize $\frac{1}{\sqrt{T}}$ (Algorithm 1), SOS SIGN-SGD (Algorithm 5), ALIAS (Algorithm 2) and STEEPEST DESCENT (Algorithms 8, 9). We validate the criteria $\|\nabla f(x^t)\|_1$ on four datasets sourced from the LIBSVM library (Chang & Lin, 2011): `a9a`, `w8a`, `ijcnn1` and `skin-nonskin`. The results are presented in Figure 6.

The plots show that even on the convex problems, SOS SIGN-SGD performs worse than ALIAS. This was expected, however, testing this method on a real non-convex problem, such as training LLMs, lacks justification. Additionally, it is noteworthy that STEEPEST DESCENT performs worse compared to SIGN-SGD, highlighting the limited practical applicability of this approach. Consequently, we provide analysis for STEEPEST DESCENT only with incorporated Algorithm 4 in Appendix G. We do not focus on the analysis and development of efficient parameter-free methods based on this approach.

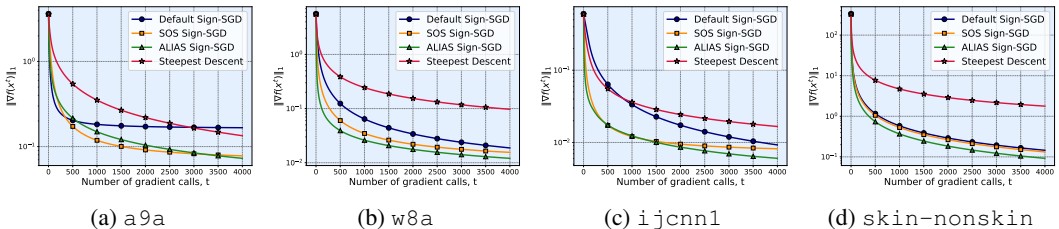

Figure 6: SIGN-SGD methods on logistic regression.

### B.1.2 NON-CONVEX PROBLEM

We provide the comparison of SIGN-SGD with theoretical stepsize $\frac{1}{\sqrt{T}}$ (Algorithm 1), SOS SIGN-SGD (Algorithm 5), ALIAS (Algorithm 2) and STEEPEST DESCENT (Algorithms 8, 9). We validate criteria $\|\nabla f(x^t)\|_1$ on four datasets, sourced from the LIBSVM library (Chang & Lin, 2011): a9a, w8a, ijcnn1 and skin-nonskin. In the main part we presented the results for the convex problem. Now we consider the non-convex objective, namely the non-linear least squares loss:

$$f(x) = \frac{1}{n} \sum_{i=1}^{n} \left( y_i - \frac{1}{1 + \exp\left(-a_i^T x\right)} \right)^2. \tag{5}$$

There we denote $a_i \in \mathbb{R}^{1 \times d}$ as the sample and $y_i \in \{0, 1\}$ as the target. The results are presented in Figure 7.

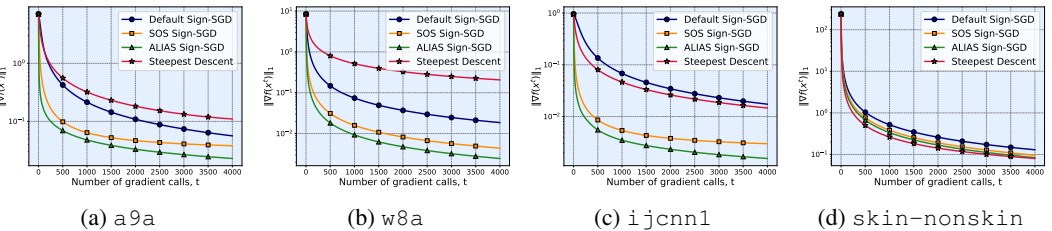

Figure 7: Comparison of SIGN-SGD methods on problem equation 5.

The plots demonstrate the superiority of our methods, SOS SIGN-SGD and ALIAS, over classical SIGN-SGD with the vanilla step size choice $\frac{1}{\sqrt{T}}$. This highlights the importance of adapting the stepsize to the problem structure and hyperparameters. However, SOS SIGN-SGD still underperforms compared to ALIAS.

## C  ADDITIONAL NOTATION AND GENERAL INEQUALITIES

**Notation.**  Here we present the full list of notation, used in our paper.
• We denote $d$ as the dimension of the problem; $T$ as the total number of iterations in the algorithms; $x^{-1}$ as the starting point in the SOS SIGN-SGD algorithm, $x^0$ as the starting point in the ALIAS algorithm; $x^t$ as the point at $t$-th iteration in the algorithms; $x^*$ as the optimal solution of the problem; $\widetilde{\Delta}_T = f(x^{-1}) - \min_{-1 \leqslant t \leqslant T} f(x^t)$; $\Delta^* = f(x^{-1}) - f(x^*)$ for the SOS SIGN-SGD method, $\Delta^* = f(x^0) - f(x^*)$ for the ALIAS method.
• We denote $\nabla f(x^t)$ as the honest full gradient of the objective function at the point $x^t$; $g^t$ (or $g_{\xi^t}^t$) as the stochastic gradient of the objective function at the point $x^t$, according to the stochastic realization $\xi^t$ (we add lower index only when we use different stochastic realizations in the method); $g_j^t$ (or $g_{j,\xi^t}^t$) as the stochastic gradient of the objective function at the point $x^t$ on $j$-th device in the distributed setup, according to the stochastic realization $\xi^t$.
• For vectors $x, y \in \mathbb{R}^d$ we denote $\text{sign}(x)$ as the vector of the dimension $d$, where the $i$-th coordinate

defines as

$$[\text{sign}(x)]_i = \text{sign}(x_i) = \begin{cases} 1, & \text{if } x_i > 0 \\ 0, & \text{if } x_i = 0 \ ; \\ -1, & \text{if } x_i < 0 \end{cases}$$

$\langle x, y \rangle = \sum_{i=1}^{d} x_i y_i$ is the scalar product; $\|x\|_1 = \sum_{i=1}^{d} |x_i|$ is $l_1$-norm; $\|x\|_2 = \sqrt{\sum_{i=1}^{d} x_i^2}$ is $l_2$-norm; $\|x\|_\infty = \max_{i \in 1,d} |x_i|$ is $l_\infty$-norm.

• For a random vector $\xi \in \mathbb{R}^d$ and fixed vector $\psi \in \mathbb{R}^d$ we denote $\mathbb{E}[\xi]$ is the expected value with respect to a random vector $\xi$ and $\mathbb{E}[\xi|\psi]$ as the expected value with the respect to a random vector $\xi$, conditioned on the fixed vector $\psi$.

**General inequalities.** Suppose $x, y \in \mathbb{R}^d$, $a, b \in \mathbb{R}$, $f(\cdot)$ is under Assumption 3.1 and $\xi, \psi \in \mathbb{R}_+$ are random variables. Then,

$$\|\nabla f(x) - \nabla f(y)\|_1 \quad \leqslant \quad L_\infty \|x - y\|_\infty \tag{Lip}$$

$$\|x + y\|_1 \quad \leqslant \quad \|x\|_1 + \|y\|_1 \ \text{ or } \ \sqrt{a+b} \leqslant \sqrt{a} + \sqrt{b} \tag{CS}$$

$$\langle x, y \rangle \quad \leqslant \quad \|x\|_1 \|y\|_\infty \tag{Conj}$$

$$\mathbb{E}[\xi\psi] \quad \leqslant \quad \left(\mathbb{E}[\xi]^p\right)^{\frac{1}{p}} \left(\mathbb{E}[\psi]^q\right)^{\frac{1}{q}}, \ \text{ where } \ \frac{1}{p} + \frac{1}{q} = 1 \tag{Höl}$$

## D   LEMMAS FOR SOS SIGN-SGD

**Lemma D.1** (Quadratic inequality). *Let $x \in \mathbb{R}_+$ be a variable and $u, v \in \mathbb{R}_+$ be constants. Then $x^2 - ux - v \leqslant 0$ implies $x \leqslant u + \sqrt{v}$. Additionally, $x^2 + ux - v \leqslant 0$ implies $x \leqslant \sqrt{v}$.*

*Proof.* Since $u, v$ are non-negative constants, the plain algebra involves $x_{\text{s.p.}} = \frac{u \pm \sqrt{u^2 + 4v}}{2}$ being stationary points of $x^2 - 2ux - v \leqslant 0$ inequality. Since $x$ is the positive variable, the boundary $x \leqslant x_{\text{s.p.}+}$ is the appropriate area of the solution. It remains for us to say that

$$x \leqslant \frac{1}{2}u + \frac{1}{2}\sqrt{u^2 + 4v} \overset{CS}{\leqslant} u + \sqrt{v},$$

which finishes the proof of the first statement. Proceeding analogically for the second part, we obtain $x \leqslant -\frac{1}{2}u + \frac{1}{2}\sqrt{u^2 + 4v} \leqslant -\frac{1}{2}u + \frac{1}{2}u + \sqrt{v} = \sqrt{v}$. $\qquad\square$

**Lemma D.2** (Bisection entry). *Let $\gamma_{max} = \frac{\Delta^*}{\|Grad(f(x^0))\|_1}$ $\left(or \ \gamma_{max} = \frac{\Delta^*}{\frac{1}{M}\sum_{j=1}^{M} \|Grad(f(x^0))\|_1} \ for \ dis-\right.$*

*$\left.tributed \ setting\right)$, where $\Delta^* = f(x^{-1}) - f(x^*)$ and the gradient oracle $Grad(f(\cdot))$ can be specified as $\nabla f(\cdot)$ or $g(\cdot)$ or $g_j(\cdot)$, that depends on the algorithm setting (exact gradient, stochastic gradient or gradient on the $i$-th node in distributed setting). Then we can always entry the bisection procedure without infinite early terminations taking $\gamma_{hi} \geqslant \gamma_{max}$.*

*Proof.* We can entry the BISECTION procedure, when $\gamma_{hi} \geqslant \phi(\gamma_{hi})$. Thus, to proof the lemma statement we can show that $\gamma_{hi} < \phi(\gamma_{hi})$ is impossible, when $\gamma_{hi} \geqslant \gamma_{max} = \frac{\Delta^*}{\|Grad(f(x^0))\|_1}$. Using $\widetilde{\Delta}_T = f(x^{-1}) - \min_{-1 \leqslant t \leqslant T} f(x^t)$ notation, we consider

$$\frac{\widetilde{\Delta}_T(\gamma_{hi})}{\mathfrak{D}_T(\gamma_{hi})} = \frac{\mathfrak{N}_T(\gamma_{hi})}{\mathfrak{D}_T(\gamma_{hi})} = \phi(\gamma_{hi}) > \gamma_{hi} \geqslant \gamma_{max} = \frac{\Delta^*}{\|Grad(f(x^0))\|_1}. \tag{6}$$

Let us look at the numerators of the fractions in the obtained inequality. According to Assumption 3.3, $f(x^*) \leqslant \min\limits_{-1 \leqslant t \leqslant T} f(x^t)$. In that way,

$$\widetilde{\Delta}_T(\gamma_{\mathsf{hi}}) \leqslant \Delta^*. \tag{7}$$

Now we consider denominators in equation 6. $\mathfrak{D}_T(\gamma_{\mathsf{hi}})$ has the following form in any setting: $\sum\limits_{t=0}^{T-1} \|\mathrm{Grad}(f(x^{t+1}(\gamma_{\mathsf{hi}})) - \mathrm{Grad}(f(x^t(\gamma_{\mathsf{hi}}))\|_1 + \zeta(\gamma_{\mathsf{hi}})$, where $\zeta(\gamma)$ is defined as the minimum of gradients norm over the training: $\zeta(\gamma) = \min\limits_{0 \leqslant t \leqslant T} \|\mathrm{Grad}(f(x^t(\gamma_{\mathsf{hi}}))\|_1$. Using equation CS, we obtain

$$
\begin{aligned}
\|\mathrm{Grad}(f(x^0))\|_1 \quad &\overset{(i)}{\leqslant} \quad \sum_{t=0}^{\bar{t}-1} \|\mathrm{Grad}(f(x^{t+1}(\gamma_{\mathsf{hi}})) - \mathrm{Grad}(f(x^t(\gamma_{\mathsf{hi}}))\|_1 + \|\mathrm{Grad}(f(x^{\bar{t}}(\gamma_{\mathsf{hi}}))\|_1 \\
&\leqslant \quad \sum_{t=0}^{T-1} \|\mathrm{Grad}(f(x^{t+1}(\gamma_{\mathsf{hi}})) - \mathrm{Grad}(f(x^t(\gamma_{\mathsf{hi}}))\|_1 + \|\mathrm{Grad}(f(x^{\bar{t}}(\gamma_{\mathsf{hi}}))\|_1 \\
&\overset{(ii)}{=} \quad \sum_{t=0}^{T-1} \|\mathrm{Grad}(f(x^{t+1}(\gamma_{\mathsf{hi}})) - \mathrm{Grad}(f(x^t(\gamma_{\mathsf{hi}}))\|_1 \\
&\quad\quad + \min_{0 \leqslant t \leqslant T} \|\mathrm{Grad}(f(x^t(\gamma_{\mathsf{hi}}))\|_1 \\
&\overset{(iii)}{=} \quad \sum_{t=0}^{T-1} \|\mathrm{Grad}(f(x^{t+1}(\gamma_{\mathsf{hi}})) - \mathrm{Grad}(f(x^t(\gamma_{\mathsf{hi}}))\|_1 + \zeta(\gamma_{\mathsf{hi}}) \\
&= \quad \mathfrak{D}_T(\gamma_{\mathsf{hi}})),
\end{aligned} \tag{8}
$$

where inequality (i) holds for any $1 \leqslant \bar{t} \leqslant T$ and in (ii) we choose $\bar{t} = \arg\min\limits_{0 \leqslant t \leqslant T} \|\mathrm{Grad}(f(x^t(\gamma_{\mathsf{hi}}))\|_1$. One can note that we omit the case when the norm of the oracle reaches its minimum at iteration $t = 0$ in $\zeta$ definition, when use it in (iii). However, it is a trivial case and it satisfies

$$\|\mathrm{Grad}(f(x^0))\|_1 \leqslant \zeta(\gamma_{\mathsf{hi}}) \leqslant \sum_{t=0}^{T-1} \|\mathrm{Grad}(f(x^{t+1}(\gamma_{\mathsf{hi}})) - \mathrm{Grad}(f(x^t(\gamma_{\mathsf{hi}}))\|_1 + \zeta(\gamma_{\mathsf{hi}}) = \mathfrak{D}_T(\gamma_{\mathsf{hi}}).$$

In that way, combining it with equation 8 and equation 7, we obtain

$$\frac{\widetilde{\Delta}_T(\gamma_{\mathsf{hi}})}{\mathfrak{D}_T(\gamma_{\mathsf{hi}})} \leqslant \frac{\Delta^*}{\|\mathrm{Grad}(f(x^0))\|_1},$$

which contradicts to equation 6. Thus, we can entry the Algorithm 4 without infinite early termination if take initial $\gamma_{\mathsf{hi}}$ at least $\frac{\Delta^*}{\|\mathrm{Grad}(f(x^0))\|_1}$. Note that for the distributed case we can obtain $\frac{1}{M} \sum\limits_{j=1}^{M} \left\|\mathrm{Grad}\left(f(x^0)\right)\right\|_1 \leqslant \mathfrak{D}_T(\gamma_{\mathsf{hi}})$ in the same way as in equation 8. $\qquad\square$

**Lemma D.3** (Bisection invariants). *If The* BISECTION *procedure (Algorithm 4) has no early termination at all, it returns $\gamma_0$ such that*

$$\frac{\mathfrak{N}_T(\gamma_0)}{2\mathfrak{D}_T(\gamma_{lo}^*)} \leqslant \gamma_0 \leqslant \frac{\mathfrak{N}_T(\gamma_{lo}^*)}{\mathfrak{D}_T(\gamma_0)}, \tag{9}$$

*where $\gamma_{lo}^*$ and $\gamma_{hi}^*$ are values, from which $\gamma_0$ is chosen in the end of Algorithm 4. Moreover,*

$$\mathfrak{N}_T(\gamma_0) \leqslant \mathfrak{N}_T(\gamma_{lo}^*), \tag{10}$$
$$\mathfrak{D}_T(\gamma_0) \leqslant \mathfrak{D}_T(\gamma_{hi}^*). \tag{11}$$

*Proof.* Consider the case procedure returns $\gamma_0 = \gamma_{\mathsf{lo}}^*$. Then

$$\frac{\mathfrak{N}_T(\gamma_{\mathsf{lo}}^*)}{2\mathfrak{D}_T(\gamma_{\mathsf{hi}}^*)} = \frac{\mathfrak{N}_T(\gamma_{\mathsf{lo}}^*)}{2\mathfrak{N}_T(\gamma_{\mathsf{hi}}^*)} \cdot \frac{\mathfrak{N}_T(\gamma_{\mathsf{hi}}^*)}{\mathfrak{D}_T(\gamma_{\mathsf{hi}}^*)} = \frac{\mathfrak{N}_T(\gamma_{\mathsf{lo}}^*)}{2\mathfrak{N}_T(\gamma_{\mathsf{hi}}^*)} \phi(\gamma_{\mathsf{hi}}^*) \quad \overset{(i)}{\leqslant} \quad \frac{1}{2}\gamma_{\mathsf{hi}}^* \overset{(ii)}{\leqslant} \gamma_{\mathsf{lo}}^*$$

$$\overset{(iii)}{\leqslant} \quad \phi(\gamma_{\text{lo}}^*) = \frac{\mathfrak{N}_T(\gamma_{\text{lo}}^*)}{\mathfrak{D}_T(\gamma_{\text{lo}}^*)}, \qquad (12)$$

where (*i*) is correct due to the $\gamma_{\text{lo}}$ return condition, (*ii*) – bisection stop condition, (*iii*) – bisection invariant. Consider the case when procedure returns $\gamma_0 = \gamma_{\text{hi}}^*$. Then

$$\frac{\mathfrak{N}_T(\gamma_{\text{hi}}^*)}{2\mathfrak{D}_T(\gamma_{\text{hi}}^*)} = \frac{1}{2}\phi(\gamma_{\text{hi}}^*) \overset{(i)}{\leqslant} \frac{1}{2}\gamma_{\text{hi}}^* \leqslant \gamma_{\text{hi}}^* \overset{(ii)}{\leqslant} \frac{\mathfrak{N}_T(\gamma_{\text{hi}}^*)}{\mathfrak{D}_T(\gamma_{\text{hi}}^*)}, \qquad (13)$$

where (*i*) is correct due to the bisection invariant and (*ii*) – $\gamma_{\text{hi}}$ the return condition. Combining equation 12 with equation 13, we obtain the first claim of the lemma whether Algorithm 4 returns $\gamma_0 = \gamma_{\text{lo}}^*$ or $\gamma_0 = \gamma_{\text{hi}}^*$. It remains to notice that equation 12 is followed by $\mathfrak{D}_T(\gamma_{\text{lo}}^*) \leqslant \mathfrak{D}_T(\gamma_{\text{hi}}^*)$ when $\gamma_0 = \gamma_{\text{lo}}^*$, and, consequently, $\mathfrak{D}_T(\gamma_0) \leqslant \mathfrak{D}_T(\gamma_{\text{hi}}^*)$ since $\mathfrak{D}_T(\gamma_{\text{hi}}^*) \leqslant \mathfrak{D}_T(\gamma_{\text{hi}}^*)$ is trivial. Analogically, equation 13 is followed by $\mathfrak{N}_T(\gamma_{\text{hi}}^*) \leqslant \mathfrak{N}_T(\gamma_{\text{lo}}^*)$ when $\gamma_0 = \gamma_{\text{hi}}^*$, and, consequently, $\mathfrak{N}_T(\gamma_0) \leqslant \mathfrak{N}_T(\gamma_{\text{lo}}^*)$. This finishes the proof. $\qquad\square$

**Lemma D.4** (Extra step). *Suppose Assumptions 3.1, 3.2, 3.3 hold. Then, considering update of the following form:*

$$f(x^0) = \min\left\{ f(x^{-1} + \tau e), f(x^{-1} - \tau e) \right\},$$

*where $e$ is the random vector from the unit basis, and we can guarantee $f(x^0) < f(x^{-1})$, when $\tau < \frac{\|\nabla f(x^{-1})\|_1}{L_\infty}$. Moreover, Algorithm 4, starting with $\tau = \tau_s$ and performing $\tau = \frac{\tau}{2}$, needs at least $\log\left( \frac{\tau_s L_\infty}{\|\nabla f(x^{-1})\|_1} \right)$ extra iterations to find efficient value of $\tau$.*

*Proof.* We choose $f(x^0) = \min\left\{ f(x^{-1} + \tau e), f(x^{-1} - \tau e) \right\}$. We use convexity to show

$$
\begin{aligned}
f(x^{-1} + \tau e) &\leqslant & f(x^{-1}) + \langle \nabla f(x^{-1} + \tau e), \tau e \rangle \\
&= & f(x^{-1}) + \tau \langle \nabla f(x^{-1}), e \rangle + \tau \langle \nabla f(x^{-1} + \tau e) - \nabla f(x^{-1}), e \rangle \\
&\overset{Conj}{\leqslant} & f(x^{-1}) + \tau \langle \nabla f(x^{-1}), e \rangle + \tau \left\| \nabla f(x^{-1} + \tau e) - \nabla f(x^{-1}) \right\|_1 \|e\|_\infty \\
&\overset{Lip}{\leqslant} & f(x^{-1}) + \tau \langle \nabla f(x^{-1}), e \rangle + \tau^2 L_\infty \|e\|_\infty^2, \\
f(x^{-1} - \tau e) &\leqslant & f(x^{-1}) - \langle \nabla f(x^{-1} - \tau e), \tau e \rangle \\
&= & f(x^{-1}) - \tau \langle \nabla f(x^{-1}), e \rangle - \tau \langle \nabla f(x^{-1} - \tau e) - \nabla f(x^{-1}), e \rangle \\
&\overset{Conj}{\leqslant} & f(x^{-1}) - \tau \langle \nabla f(x^{-1}), e \rangle + \tau \left\| \nabla f(x^{-1} - \tau e) - \nabla f(x^{-1}) \right\|_1 \|e\|_\infty \\
&\overset{Lip}{\leqslant} & f(x^{-1}) - \tau \langle \nabla f(x^{-1}), e \rangle + \tau^2 L_\infty \|e\|_\infty^2.
\end{aligned}
$$

Utilizing $e = \text{sign}\left(\nabla f(x^{-1})\right)$, we take expectation and obtain

$$
\begin{aligned}
f(x^0) &\leqslant & f(x^{-1}) - \tau \left| \langle \nabla f(x^{-1}), e \rangle \right| + \tau^2 L_\infty \|e\|_\infty^2 \\
&= & f(x^{-1}) - \tau \left| \sum_{i=1}^d \left[ |\nabla f(x^{-1})| \right]_i \right| + \tau^2 L_\infty \left\| \text{sign}\left(\nabla f(x^{-1})\right) \right\|_\infty^2 \\
&\leqslant & f(x^{-1}) - \tau \left\| \nabla f(x^{-1}) \right\|_1 + \tau^2 L_\infty \\
&= & f(x^{-1}) - \tau \left( \left\| \nabla f(x^{-1}) \right\|_1 - \tau L_\infty \right).
\end{aligned}
$$

In that way, if we have $\tau < \frac{\|\nabla f(x^{-1})\|_1}{L_\infty}$, we derive

$$f(x^0) < f(x^{-1}).$$

Since in the algorithm we start with $\tau = \tau_s$ and divide it by 2, after $l$ divisions, we have

$$\frac{\tau_s}{2^l} < \frac{\|\nabla f(x^{-1})\|_1}{L_\infty}.$$

Thus, we need at least $l = \log\left( \frac{\tau_s L_\infty}{\|\nabla f(x^{-1})\|_1} \right)$ iterations. $\qquad\square$

# E   MAIN PROOFS AND DETAILS FOR SOS SIGN-SGD

## E.1   EXACT GRADIENT SETTING

**Lemma E.1** (Descent lemma). *For Algorithm 5 under Assumptions 3.1, 3.2, 3.3, 3.4, the following estimate is valid:*

$$\sum_{t=0}^{T-1} \|\nabla f(x^t)\|_1 \;\leqslant\; \frac{f(x^{-1}) - f(x^T)}{\gamma_0} + \sum_{t=0}^{T-1} \|\nabla f(x^{t+1}) - \nabla f(x^t)\|_1.$$

*Proof.* Starting from the convexity of the objective,

$$
\begin{aligned}
f(x^{t+1}) - f(x^t) \;&\leqslant\; \langle \nabla f(x^{t+1}), x^{t+1} - x^t \rangle = -\gamma^t \left\langle \nabla f(x^{t+1}), \mathrm{sign}\left(\nabla f(x^t)\right) \right\rangle \\
&= \; -\gamma^t \left\langle \nabla f(x^t), \mathrm{sign}\left(\nabla f(x^t)\right) \right\rangle \\
&\quad -\gamma^t \left\langle \nabla f(x^{t+1}) - \nabla f(x^t), \mathrm{sign}\left(\nabla f(x^t)\right) \right\rangle \\
&\overset{Conj}{\leqslant}\; -\gamma^t \|\nabla f(x^t)\|_1 + \gamma^t \|\nabla f(x^{t+1}) - \nabla f(x^t)\|_1 \|\mathrm{sign}\left(\nabla f(x^t)\right)\|_\infty \\
&\leqslant\; -\gamma^t \|\nabla f(x^t)\|_1 + \gamma^t \|\nabla f(x^{t+1}) - \nabla f(x^t)\|_1.
\end{aligned}
$$

Now we express the gradient norm and sum over all iterations to obtain

$$
\begin{aligned}
\sum_{t=0}^{T-1} \gamma^t \|\nabla f(x^t)\|_1 \;&\leqslant\; \sum_{t=0}^{T-1} \left[f(x^t) - f(x^{t+1})\right] + \sum_{t=0}^{T-1} \gamma^t \|\nabla f(x^{t+1}) - \nabla f(x^t)\|_1 \\
&=\; f(x^0) - f(x^T) + \sum_{t=0}^{T-1} \gamma^t \|\nabla f(x^{t+1}) - \nabla f(x^t)\|_1.
\end{aligned}
$$

Using Lemma D.4 to consider the extra step, we get

$$\sum_{t=0}^{T-1} \gamma^t \|\nabla f(x^t)\|_1 \;\leqslant\; f(x^{-1}) - f(x^T) + \sum_{t=0}^{T-1} \gamma^t \|\nabla f(x^{t+1}) - \nabla f(x^t)\|_1.$$

Since Algorithm 5 performs all the steps with the constant rate $\gamma_0$ which we define later, we can rewrite the result in the following form:

$$\sum_{t=0}^{T-1} \|\nabla f(x^t)\|_1 \;\leqslant\; \frac{f(x^{-1}) - f(x^T)}{\gamma_0} + \sum_{t=0}^{T-1} \|\nabla f(x^{t+1}) - \nabla f(x^t)\|_1,$$

which ends the proof of the lemma. $\qquad\square$

**Theorem E.2** (**Theorem B.1**). *Suppose Assumptions 3.1, 3.2, 3.3, 3.4 hold. Then for Algorithm 5 after obtaining the stepsize $\gamma_0$, the following estimate is valid:*

$$\frac{1}{T}\sum_{t=0}^{T-1} \|\nabla f(x^t)\|_1 \leqslant 6\frac{\sqrt{\Delta^* L_\infty}}{\sqrt{T}} + \frac{3\left\|\nabla f(x^0)\right\|_1}{T}.$$

*Moreover, taking into account the complexity of Algorithm 4 in relation to the initial stepsize bound $\gamma_s$, to reach $\varepsilon$-accuracy, where $\varepsilon \geqslant \frac{1}{T}\sum_{t=0}^{T-1} \|\nabla f(x^t)\|_1$, Algorithm 5 needs*

$$\mathcal{O}\left(\frac{\Delta^* L_\infty}{\varepsilon^2} \log\log \frac{\Delta^*}{\gamma_s \|\nabla f(x^0)\|_1}\right) \quad \text{iterations.}$$

*Proof.* Let us start with the result of Lemma E.1:

$$\sum_{t=0}^{T-1} \|\nabla f(x^t)\|_1 \;\leqslant\; \frac{f(x^{-1}) - f(x^T)}{\gamma_0} + \sum_{t=0}^{T-1} \|\nabla f(x^{t+1}) - \nabla f(x^t)\|_1$$

$$\leqslant \quad \frac{\widetilde{\Delta}_T}{\gamma_0} + \sum_{t=0}^{T-1} \|\nabla f(x^{t+1}) - \nabla f(x^t)\|_1, \tag{14}$$

where $\widetilde{\Delta}_T = f(x^{-1}) - \min_{-1 \leqslant t \leqslant T} f(x^t)$. Now, we accurately estimate the last term in equation 14,

which is additionally denoted as $F_T = \sum_{t=0}^{T-1} \|\nabla f(x^{t+1}) - \nabla f(x^t)\|_1$. Thus,

$$
\begin{aligned}
F_T &= \sum_{t=0}^{T-1} \|\nabla f(x^{t+1}) - \nabla f(x^t)\|_1 \overset{Lip}{\leqslant} L_\infty \sum_{t=0}^{T-1} \|x^{t+1} - x^t\|_\infty \\
&= L_\infty \sum_{t=0}^{T-1} \gamma^t \|\text{sign}\left(\nabla f(x^t)\right)\|_\infty \leqslant L_\infty \sum_{t=0}^{T-1} \gamma^t.
\end{aligned} \tag{15}
$$

Now let us choose $\phi(\gamma)$, which we push into the BISECTION procedure (Algorithm 4): $\phi(\gamma) = \frac{\mathfrak{N}(\gamma)}{\mathfrak{D}(\gamma)} = \frac{\widetilde{\Delta}_T(\gamma)}{F_T(\gamma) + \zeta(\gamma)}$, where $\widetilde{\Delta}_T = f(x^{-1}) - \min_{-1 \leqslant t \leqslant T} f(x^t)$ and $\zeta = \min_{0 \leqslant t \leqslant T} \|\nabla f(x^t)\|_1$. In that way, we obtain some $\gamma_0$, which can be equal to $\gamma_{\text{lo}}^*$ or $\gamma_{\text{hi}}^*$ (see Lemma D.2, Lemma D.3) and use it as a constant stepsize for our method. Thus, equation 15 transforms into

$$F_T(\gamma_0) \leqslant \gamma_0 L_\infty T. \tag{16}$$

Mention that, according to Lemma D.2, we can always entry to the procedure without infinite early termination. In that way, we have two situations: when we have no early terminations at all and we are under the activity of Lemma D.3, and when we have early termination with initial $\gamma_{\text{lo}}^*$. We divide the following proof into two steps, where we separately show the convergence guarantees in this two situations.

**Step 1: no early terminations.**
Since we have only two cases: $\gamma_0 = \gamma_{\text{lo}}^*$ or $\gamma_0 = \gamma_{\text{hi}}^*$, let us consider them separately.

- $\gamma_0 = \gamma_{\text{hi}}^*$ : equation 16 transforms into

$$F_T(\gamma_{\text{hi}}^*) \leqslant \gamma_{\text{hi}}^* L_\infty T \overset{\text{Lemma } D.3,9}{\leqslant} \frac{\mathfrak{N}_T(\gamma_{\text{lo}}^*)}{\mathfrak{D}_T(\gamma_{\text{hi}}^*)} L_\infty T \overset{(i)}{=} \frac{\widetilde{\Delta}_T(\gamma_{\text{lo}}^*)}{F_T(\gamma_{\text{hi}}^*) + \zeta(\gamma_{\text{hi}}^*)} L_\infty T,$$

where $(i)$ is correct due to the $\phi(\gamma)$ choice. Solving this quadratic inequality with respect to $F_T(\gamma_{\text{hi}}^*)$ (Lemma D.1), we obtain

$$F_T(\gamma_{\text{hi}}^*) \leqslant \sqrt{\widetilde{\Delta}_T(\gamma_{\text{lo}}^*) L_\infty T} \leqslant \sqrt{\Delta^* L_\infty T}, \tag{17}$$

where $\Delta^* = f(x^{-1}) - f(x^*)$. Plugging it into equation 14, we obtain

$$
\begin{aligned}
\frac{1}{T} \sum_{t=0}^{T-1} \|\nabla f(x^t)\|_1 &\leqslant & \frac{1}{T} \frac{\widetilde{\Delta}_T(\gamma_{\text{hi}}^*)}{\gamma_{\text{hi}}^*} + \frac{1}{T} F_T(\gamma_{\text{hi}}^*) \\
&\overset{\text{Lemma } D.3,9}{\leqslant} & \frac{1}{T} \frac{2\mathfrak{D}_T(\gamma_{\text{hi}}^*)}{\mathfrak{N}_T(\gamma_{\text{hi}}^*)} \widetilde{\Delta}_T(\gamma_{\text{hi}}^*) + \frac{1}{T} F_T(\gamma_{\text{hi}}^*) \\
&= & \frac{2}{T} \frac{[F_T(\gamma_{\text{hi}}^*) + \zeta(\gamma_{\text{hi}}^*)] \widetilde{\Delta}_T(\gamma_{\text{hi}}^*)}{\widetilde{\Delta}_T(\gamma_{\text{hi}}^*)} + \frac{1}{T} F_T(\gamma_{\text{hi}}^*) \\
&= & \frac{3}{T} F_T(\gamma_{\text{hi}}^*) + \frac{2\zeta(\gamma_{\text{hi}}^*)}{T} \\
&\overset{17}{\leqslant} & 3 \frac{\sqrt{\Delta^* L_\infty}}{\sqrt{T}} + \frac{2 \|\nabla f(x^0)\|_1}{T}.
\end{aligned} \tag{18}
$$

In that way, equation 18 is the final estimate when BISECTION procedure returns $\gamma_{\text{hi}}^*$.

- $\gamma_0 = \gamma_{\text{lo}}^*$ : equation 16 transforms into

$$F_T(\gamma_{\text{lo}}^*) \leqslant \gamma_{\text{lo}}^* L_\infty T \overset{\text{Lemma } D.3,9}{\leqslant} \frac{\mathfrak{N}_T(\gamma_{\text{lo}}^*)}{\mathfrak{D}_T(\gamma_{\text{lo}}^*)} L_\infty T \overset{(i)}{=} \frac{\widetilde{\Delta}_T(\gamma_{\text{lo}}^*)}{F_T(\gamma_{\text{lo}}^*) + \zeta(\gamma_{\text{lo}}^*)} L_\infty T,$$

where $(i)$ is correct due to the $\phi(\gamma)$ choice. Solving this quadratic inequality with respect to $F_T(\gamma_{\mathrm{lo}}^*)$ (Lemma D.1), we obtain

$$F_T(\gamma_{\mathrm{lo}}^*) \leqslant \sqrt{\widetilde{\Delta}_T(\gamma_{\mathrm{lo}}^*)L_\infty T} \leqslant \sqrt{\Delta^* L_\infty T}. \tag{19}$$

Now we make an additional distinction and consider the estimates separately: one case when $\gamma_{\mathrm{lo}}^* > \sqrt{\frac{\Delta^*}{L_\infty T}}$, and another when $\gamma_{\mathrm{lo}}^* \leqslant \sqrt{\frac{\Delta^*}{L_\infty T}}$. We can do this without any limitations, since combining the intervals considered for $\gamma_{\mathrm{lo}}^*$ returns all possible values.

○ $\gamma_{\mathrm{lo}}^* > \sqrt{\frac{\Delta^*}{L_\infty T}}$ : we straightforwardly move to the equation 14 estimation:

$$
\begin{aligned}
\frac{1}{T}\sum_{t=0}^{T-1}\|\nabla f(x^t)\|_1 &\leqslant& \frac{1}{T}\frac{\widetilde{\Delta}_T(\gamma_{\mathrm{lo}}^*)}{\gamma_{\mathrm{lo}}^*} + \frac{1}{T}F_T(\gamma_{\mathrm{lo}}^*) \\
&\leqslant& \frac{\sqrt{L_\infty}}{\sqrt{\Delta^* T}}\widetilde{\Delta}_T(\gamma_{\mathrm{lo}}^*) + \frac{1}{T}F_T(\gamma_{\mathrm{lo}}^*) \\
&\overset{19}{\leqslant}& \frac{\sqrt{\Delta^* L_\infty}}{\sqrt{T}} + \frac{\sqrt{\Delta^* L_\infty}}{\sqrt{T}} = 2\frac{\sqrt{\Delta^* L_\infty}}{\sqrt{T}}.
\end{aligned}
\tag{20}
$$

○ $\gamma_{\mathrm{lo}}^* \leqslant \sqrt{\frac{\Delta^*}{L_\infty T}}$ : in this case, we start from the estimate that is followed by equation 16:

$$F_T(\gamma_{\mathrm{hi}}^*) \leqslant \gamma_{\mathrm{hi}}^* L_\infty T \overset{(i)}{\leqslant} 2\gamma_{\mathrm{lo}}^* L_\infty T \leqslant 2\sqrt{L_\infty \Delta^* T}, \tag{21}$$

where $(i)$ is done due to the bisection stop condition. Now we proceed with estimation of equation 14:

$$
\begin{aligned}
\frac{1}{T}\sum_{t=0}^{T-1}\|\nabla f(x^t)\|_1 &\leqslant& \frac{1}{T}\frac{\widetilde{\Delta}_T(\gamma_{\mathrm{lo}}^*)}{\gamma_{\mathrm{lo}}^*} + \frac{1}{T}F_T(\gamma_{\mathrm{lo}}^*) \\
&\overset{\text{Lemma } D.3,9}{\leqslant}& \frac{1}{T}\frac{2\mathfrak{D}_T(\gamma_{\mathrm{hi}}^*)}{\mathfrak{N}_T(\gamma_{\mathrm{lo}}^*)}\widetilde{\Delta}_T(\gamma_{\mathrm{lo}}^*) + \frac{1}{T}F_T(\gamma_{\mathrm{lo}}^*) \\
&\overset{\text{Lemma } D.3,11}{\leqslant}& \frac{2}{T}\frac{[F_T(\gamma_{\mathrm{hi}}^*) + \zeta(\gamma_{\mathrm{hi}}^*)]\widetilde{\Delta}_T(\gamma_{\mathrm{lo}}^*)}{\widetilde{\Delta}_T(\gamma_{\mathrm{lo}}^*)} + \frac{F_T(\gamma_{\mathrm{hi}}^*) + \zeta(\gamma_{\mathrm{hi}}^*)}{T} \\
&=& \frac{3F_T(\gamma_{\mathrm{hi}}^*)}{T} + \frac{3\zeta(\gamma_{\mathrm{hi}}^*)}{T} \\
&\overset{21}{\leqslant}& 6\frac{\sqrt{\Delta^* L_\infty}}{\sqrt{T}} + \frac{3\zeta(\gamma_{\mathrm{hi}}^*)}{T} \\
&\leqslant& 6\frac{\sqrt{\Delta^* L_\infty}}{\sqrt{T}} + \frac{3\|\nabla f(x^0)\|_1}{T}.
\end{aligned}
\tag{22}
$$

Combining equation 20 and equation 22, we get that equation 22 is the final estimate when BISECTION procedure returns $\gamma_{\mathrm{lo}}^*$.

In the end, equation 18 and equation 22 give us the estimate in the case when BISECTION procedure does not have early terminations at all and outputs any value of $\gamma_0$:

$$\frac{1}{T}\sum_{t=0}^{T-1}\|\nabla f(x^t)\|_1 \leqslant 6\frac{\sqrt{\Delta^* L_\infty}}{\sqrt{T}} + \frac{3\|\nabla f(x^0)\|_1}{T}. \tag{23}$$

**Step 2: early termination with $\gamma_{\mathrm{lo}}$.**

Now we consider the scenario when with initial $\gamma_{\mathrm{lo}}$, there is $\gamma_{\mathrm{lo}} \geqslant \phi(\gamma_{\mathrm{lo}})$ and algorithm early returns $\gamma_{\mathrm{lo}}^*$. To dissect this, we should choose an initial $\gamma_{\mathrm{lo}} = \gamma_{\mathrm{lo}}^* \leqslant \frac{\Delta^*}{L_\infty T}$. Thus, equation 16 transforms into

$$F_T(\gamma_{\mathrm{lo}}^*) \leqslant \gamma_{\mathrm{lo}} L_\infty T \leqslant \sqrt{L_\infty \Delta^* T}. \tag{24}$$

In that way, equation 14 turns into

$$\frac{1}{T}\sum_{t=0}^{T-1}\|\nabla f(x^t)\|_1 \leqslant \frac{1}{T}\frac{\widetilde{\Delta}_T(\gamma_{\mathrm{lo}}^*)}{\gamma_{\mathrm{lo}}^*} + \frac{1}{T}F_T(\gamma_{\mathrm{lo}}^*)$$

$$
\begin{aligned}
&\leqslant && \frac{1}{T} \frac{\widetilde{\Delta}_T(\gamma_{\text{lo}}^*)}{\phi(\gamma_{\text{lo}}^*)} + \frac{1}{T} F_T(\gamma_{\text{lo}}^*) \\
&= && \frac{1}{T} \frac{[F_T(\gamma_{\text{lo}}^*) + \zeta(\gamma_{\text{lo}}^*)] \, \widetilde{\Delta}_T(\gamma_{\text{lo}}^*)}{\widetilde{\Delta}_T(\gamma_{\text{lo}}^*)} + \frac{1}{T} F_T(\gamma_{\text{lo}}^*) \\
&= && \frac{2F_T(\gamma_{\text{lo}}^*)}{T} + \frac{\zeta(\gamma_{\text{lo}}^*)}{T} \overset{24}{\leqslant} 2\frac{\sqrt{\Delta^* L_\infty}}{\sqrt{T}} + \frac{\left\|\nabla f(x^0)\right\|_1}{T}. && (25)
\end{aligned}
$$

Hence, equation 25 delivers the estimate, when Algorithm 4 makes an early termination. Combining equation 23 with equation 25, we finally obtain the estimate for all possible cases of the BISECTION procedure return:

$$
\frac{1}{T} \sum_{t=0}^{T-1} \|\nabla f(x^t(\gamma_0))\|_1 \leqslant 6\frac{\sqrt{\Delta^* L_\infty}}{\sqrt{T}} + \frac{3\left\|\nabla f(x^0)\right\|_1}{T}.
$$

Expressing the number of iterations and using $\varepsilon \geqslant \frac{1}{T} \sum_{t=0}^{T-1} \|\nabla f(x^t)\|_1$ as a criterion, we obtain that algorithm needs $\mathcal{O}\left(\frac{\Delta^* L_\infty}{\varepsilon^2}\right)$ iterations to reach $\varepsilon$-accuracy. Note that we drop the term $\frac{\left\|\nabla f(x^0)\right\|_1}{T}$, since it is asymptotically smaller than the main one. However, we firstly need to find the step $\gamma_0$ with the bisection procedure which takes $T \log\log\left(\frac{\gamma_\varepsilon 2^{2^k}}{\gamma_\varepsilon}\right) = \mathcal{O}\left(\frac{\Delta^* L_\infty}{\varepsilon^2} k\right)$ iterations, where $2^{2^k}$ denotes the length of the initial interval for the stepsize. We have already discussed in the main part that, according to Lemma D.2, $k$ should be at least $k = \log\log \frac{\Delta^*}{\gamma_s \|\nabla f(x^0)\|_1}$. Thus, $\mathcal{O}\left(\frac{\Delta^* L_\infty}{\varepsilon^2} \log\log \frac{\Delta^*}{\gamma_s \|\nabla f(x^0)\|_1}\right)$ is the final iteration complexity. $\qquad \square$

### E.2 STOCHASTIC GRADIENT SETTING

Let us start with the description of the stepsize choice for stochastic version of Algorithm 5. The main purpose of the BISECTION procedure (Algorithm 4) is to find stepsize $\gamma$ close enough to the $\phi(\gamma)$ desired value utilizing small number of sign descent launches. Recall we establish

$$
\phi(\gamma) = \frac{\widetilde{\Delta}_T(\gamma)}{\sum_{t=0}^{T-1} \|\nabla f(x^{t+1}) - \nabla f(x^t)\|_1 + \zeta(\gamma)}
$$

for the exact gradient case. The numerator can remain unchanged. However, since we lack access to exact gradients, we cannot use the original denominator. Instead, we employ stochastic oracles: $\mathfrak{D}_T(\gamma) = \sum_{t=0}^{T-1} \|g(x^{t+1}) - g(x^t)\|_1 + \zeta(\gamma)$. Other details remain the same, and we can straightforwardly pass to the convergence results.

**Lemma E.3** (Descent lemma). *For Algorithm 5 under Assumptions 3.1, 3.2, 3.3, 3.7, the following estimate is valid:*

$$
\sum_{t=0}^{T-1} \|\nabla f(x^t)\|_1 \leqslant \frac{f(x^{-1}) - f(x^T)}{\gamma_0} + \sum_{t=0}^{T-1} \|g^{t+1} - g^t\|_1 + 3\delta^t + \delta^{t+1},
$$

*where $\delta^t = \sum_{t=0}^{T-1} \|\nabla f(x^t) - g^t\|_1$.*

*Proof.* Starting from the convexity of the objective,

$$
\begin{aligned}
f(x^{t+1}) - f(x^t) &\leqslant && \langle \nabla f(x^{t+1}), x^{t+1} - x^t \rangle = -\gamma^t \langle \nabla f(x^{t+1}), \text{sign}(g^t) \rangle \\
&= && -\gamma^t \langle g^t, \text{sign}(g^t) \rangle - \gamma^t \langle \nabla f(x^{t+1}) - g^t, \text{sign}(g^t) \rangle \\
&= && -\gamma^t \|g^t\|_1 - \gamma^t \langle \nabla f(x^t) - g^t, \text{sign}(g^t) \rangle \\
& && -\gamma^t \langle \nabla f(x^{t+1}) - \nabla f(x^t), \text{sign}(g^t) \rangle \\
&\overset{Conj}{\leqslant} && -\gamma^t \|\nabla f(x^t)\|_1
\end{aligned}
$$

$$
\begin{aligned}
&+\gamma^t\|\nabla f(x^t) - g^t\|_1 + \gamma^t\|\nabla f(x^t) - g^t\|_1\|\mathrm{sign}\left(g^t\right)\|_\infty\\
&+\gamma^t\|\nabla f(x^{t+1}) - \nabla f(x^t)\|_1\|\mathrm{sign}\left(g^t\right)\|_\infty\\
\leqslant\quad &-\gamma^t\|\nabla f(x^t)\|_1 + 3\gamma^t\|\nabla f(x^t) - g^t\|_1 + \gamma^t\|\nabla f(x^{t+1}) - g^{t+1}\|_1\\
&+\gamma^t\|g^{t+1} - g^t\|_1.
\end{aligned}
$$

Now we rearrange terms and summarize over all iterations to obtain

$$
\begin{aligned}
\sum_{t=0}^{T-1}\gamma^t\|\nabla f(x^t)\|_1 \leqslant\quad & \sum_{t=0}^{T-1}\left[f(x^t) - f(x^{t+1})\right] + \sum_{t=0}^{T-1}\gamma^t\|g^{t+1} - g^t\|_1\\
&+3\sum_{t=0}^{T-1}\gamma^t\|\nabla f(x^t) - g^t\|_1 + \sum_{t=0}^{T-1}\gamma^t\|\nabla f(x^{t+1}) - g^{t+1}\|_1.
\end{aligned}
$$

Since Algorithm 5 performs all the steps with the constant rate $\gamma_0$, which we define later, we can rewrite the result in the following form:

$$
\begin{aligned}
\sum_{t=0}^{T-1}\|\nabla f(x^t)\|_1 \leqslant\quad & \sum_{t=0}^{T-1}\frac{\left[f(x^t) - f(x^{t+1})\right]}{\gamma_0} + \sum_{t=0}^{T-1}\|g^{t+1} - g^t\|_1\\
&+3\sum_{t=0}^{T-1}\|\nabla f(x^t) - g^t\|_1 + \sum_{t=0}^{T-1}\|\nabla f(x^{t+1}) - g^{t+1}\|_1.
\end{aligned}
$$

In the obtained estimate the last two terms consist from differences between the honest and stochastic gradient at the $t$-th and $(t+1)$-th moments. One of our goals is to estimate them, however, we want perform analogically to Theorem E.2 and continue the proof with the $\sum_{t=0}^{T-1}\|g^{t+1} - g^t\|_1$ term estimate. In order to simplify our following writing we give additional notation and denote $\delta^t = \sum_{t=0}^{T-1}\|\nabla f(x^t) - g^t\|_1$. In that way, additionally considering the extra step (Lemma D.4), we derive

$$
\sum_{t=0}^{T-1}\|\nabla f(x^t)\|_1 \leqslant \frac{f(x^{-1}) - f(x^T)}{\gamma_0} + \sum_{t=0}^{T-1}\|g^{t+1} - g^t\|_1 + 3\delta^t + \delta^{t+1},
$$

which ends the proof of the lemma. □

**Theorem E.4.** *Suppose Assumptions 3.1, 3.2, 3.3, 3.7 hold. Then for Algorithm 5 using at $t$-th iteration mini-batches of sizes $t+1$, after obtaining the stepsize $\gamma_0$, the following estimate is valid:*

$$
\frac{1}{T}\sum_{t=0}^{T-1}\mathbb{E}\|\nabla f(x^t)\|_1 \leqslant 6\frac{\sqrt{\Delta^* L_\infty}}{\sqrt{T}} + 10\|\sigma\|_1 + \frac{3\mathbb{E}\left\|g^0\right\|_1}{T}.
$$

*Moreover, taking into account the complexity of Algorithm 4 in relation to the initial stepsize bound $\gamma_s$, to reach $\varepsilon$-accuracy, where $\varepsilon \geqslant \frac{1}{T}\sum_{t=0}^{T-1}\|\nabla f(x^t)\|_1$, Algorithm 5 needs*

$$
\mathcal{O}\left(\left(\frac{\Delta^* L_\infty}{\varepsilon^2} + \|\sigma\|_1^2\right)\log\log\frac{\Delta^*}{\gamma_s\|g^0\|_1}\right) \quad \text{iterations.}
$$

*Proof.* Let us start with the result of Lemma E.3. We transform it due to the fact that Algorithm 5 performs all the steps with the constant rate $\gamma_0$, which we define later:

$$
\begin{aligned}
\sum_{t=0}^{T-1}\|\nabla f(x^t)\|_1 \leqslant\quad & \frac{f(x^{-1}) - f(x^T)}{\gamma_0} + \sum_{t=0}^{T-1}\|g^{t+1} - g^t\|_1 + 3\delta^t + \delta^{t+1}\\
\leqslant\quad & \frac{\widetilde{\Delta}_T}{\gamma_0} + \sum_{t=0}^{T-1}\|g^{t+1} - g^t\|_1 + 3\delta^t + \delta^{t+1}, \quad\quad (26)
\end{aligned}
$$

where $\widetilde{\Delta}_T = f(x^{-1}) - \min\limits_{-1 \leqslant t \leqslant T} f(x^t)$. Now, we focus on estimating $G_T = \sum\limits_{t=0}^{T-1} \|g^{t+1} - g^t\|_1$ term in equation 26. Thus,

$$
\begin{aligned}
G_T = \sum_{t=0}^{T-1} \|g^{t+1} - g^t\|_1 \quad &\leqslant \quad \sum_{t=0}^{T-1} \|\nabla f(x^{t+1}) - g^{t+1}\|_1 + \sum_{t=0}^{T-1} \|\nabla f(x^t) - g^t\|_1 \\
&\quad + \sum_{t=0}^{T-1} \|\nabla f(x^{t+1}) - \nabla f(x^t)\|_1 \\
&\overset{Lip}{\leqslant} \quad \delta^t + \delta^{t+1} + L_\infty \sum_{t=0}^{T-1} \|x^{t+1} - x^t\|_\infty \\
&= \quad \delta^t + \delta^{t+1} + L_\infty \sum_{t=0}^{T-1} \gamma^t \|\text{sign}\left(\nabla f(x^t)\right)\|_\infty \\
&\leqslant \quad \delta^t + \delta^{t+1} + L_\infty \sum_{t=0}^{T-1} \gamma^t.
\end{aligned}
\tag{27}
$$

Now let us choose $\phi(\gamma)$, which we push to the BISECTION procedure (Algorithm 4): $\phi(\gamma) = \frac{\mathfrak{N}(\gamma)}{\mathfrak{D}(\gamma)} = \frac{\widetilde{\Delta}_T(\gamma)}{G_T(\gamma)+\zeta(\gamma)}$, where $\widetilde{\Delta}_T = f(x^{-1}) - \min\limits_{-1 \leqslant t \leqslant T} f(x^t)$ and $\zeta = \min\limits_{0 \leqslant t \leqslant T} \|g^t\|_1$. In that way, we obtain some $\gamma_0$, which can be equal to $\gamma_{\text{lo}}^*$ or $\gamma_{\text{hi}}^*$ (see Lemma D.2, Lemma D.3) and use it as a constant stepsize for our method. Thus, equation 27 transforms into

$$
G_T(\gamma_0) \leqslant \delta^t + \delta^{t+1} + \gamma_0 L_\infty T.
\tag{28}
$$

Mention that, according to Lemma D.2, we can always entry to the procedure without infinite early termination. In that way we have two situations: when we have no early terminations at all and we are under the activity of Lemma D.3, and when we have an early termination with the initial $\gamma_{\text{lo}}^*$. We divide the following proof into two steps, where we separately show the convergence guarantees in these two situations.

**Step 1: no early terminations.**

Since we have only two cases: $\gamma_0 = \gamma_{\text{lo}}^*$ or $\gamma_0 = \gamma_{\text{hi}}^*$, let us consider them separately.

- $\gamma_0 = \gamma_{\text{hi}}^*$ : equation 28 transforms into

$$
\begin{aligned}
G_T(\gamma_{\text{hi}}^*) \quad &\leqslant \quad \delta^t + \delta^{t+1} + \gamma_{\text{hi}}^* L_\infty T \overset{\text{Lemma } D.3,9}{\leqslant} \delta^t + \delta^{t+1} + \frac{\mathfrak{N}_T(\gamma_{\text{lo}}^*)}{\mathfrak{D}_T(\gamma_{\text{hi}}^*)} L_\infty T \\
&\overset{(i)}{=} \quad \delta^t + \delta^{t+1} + \frac{\widetilde{\Delta}_T(\gamma_{\text{lo}}^*)}{G_T(\gamma_{\text{hi}}^*) + \zeta(\gamma_{\text{hi}}^*)} L_\infty T \leqslant \delta^t + \delta^{t+1} + \frac{\widetilde{\Delta}_T(\gamma_{\text{lo}}^*)}{G_T(\gamma_{\text{hi}}^*)} L_\infty T,
\end{aligned}
$$

where $(i)$ is correct due to the $\phi(\gamma)$ choice. Solving this quadratic inequality with respect to $G_T(\gamma_{\text{hi}}^*)$ (Lemma D.1), we obtain

$$
G_T(\gamma_{\text{hi}}^*) \leqslant \delta^t + \delta^{t+1} + \sqrt{\widetilde{\Delta}_T(\gamma_{\text{lo}}^*) L_\infty T} \leqslant \delta^t + \delta^{t+1} + \sqrt{\Delta^* L_\infty T},
\tag{29}
$$

where $\Delta^* = f(x^{-1}) - f(x^*)$. Plugging it into equation 26, we obtain

$$
\begin{aligned}
\frac{1}{T} \sum_{t=0}^{T-1} \|\nabla f(x^t)\|_1 \quad &\leqslant \quad \frac{1}{T} \frac{\widetilde{\Delta}_T(\gamma_{\text{hi}}^*)}{\gamma_{\text{hi}}^*} + \frac{1}{T} G_T(\gamma_{\text{hi}}^*) + \frac{1}{T}(3\delta^t + \delta^{t+1}) \\
&\overset{\text{Lemma } D.3,9}{\leqslant} \frac{1}{T} \frac{2\mathfrak{D}_T(\gamma_{\text{hi}}^*)}{\mathfrak{N}_T(\gamma_{\text{hi}}^*)} \widetilde{\Delta}_T(\gamma_{\text{hi}}^*) + \frac{1}{T} G_T(\gamma_{\text{hi}}^*) + \frac{1}{T}(3\delta^t + \delta^{t+1}) \\
&= \quad \frac{2}{T} \frac{[G_T(\gamma_{\text{hi}}^*) + \zeta(\gamma_{\text{hi}}^*)] \widetilde{\Delta}_T(\gamma_{\text{hi}}^*)}{\widetilde{\Delta}_T(\gamma_{\text{hi}}^*)} + \frac{1}{T} G_T(\gamma_{\text{hi}}^*) + \frac{1}{T}(3\delta^t + \delta^{t+1}) \\
&= \quad \frac{3}{T} G_T(\gamma_{\text{hi}}^*) + \frac{1}{T}(3\delta^t + \delta^{t+1}) + \frac{2\zeta(\gamma_{\text{hi}}^*)}{T}
\end{aligned}
$$

$$\overset{29}{\leqslant} \quad 3\frac{\sqrt{\Delta^* L_\infty}}{\sqrt{T}} + \frac{1}{T}(6\delta^t + 4\delta^{t+1}) + \frac{2\|g^0\|_1}{T}. \tag{30}$$

In that way, equation 30 is the final estimate when BISECTION procedure returns $\gamma_{\text{hi}}^*$.

- $\gamma_0 = \gamma_{\text{lo}}^*$ : equation 28 transforms into

$$
\begin{aligned}
G_T(\gamma_{\text{lo}}^*) \quad &\leqslant \quad \delta^t + \delta^{t+1} + \gamma_{\text{lo}}^* L_\infty T \overset{\text{Lemma } D.3,9}{\leqslant} \delta^t + \delta^{t+1} + \frac{\mathfrak{N}_T(\gamma_{\text{lo}}^*)}{\mathfrak{D}_T(\gamma_{\text{lo}}^*)} L_\infty T \\
&\overset{(i)}{=} \quad \delta^t + \delta^{t+1} + \frac{\widetilde{\Delta}_T(\gamma_{\text{lo}}^*)}{G_T(\gamma_{\text{lo}}^*) + \zeta(\gamma_{\text{lo}}^*)} L_\infty T \leqslant \delta^t + \delta^{t+1} + \frac{\widetilde{\Delta}_T(\gamma_{\text{lo}}^*)}{G_T(\gamma_{\text{lo}}^*)} L_\infty T,
\end{aligned}
$$

where $(i)$ is correct due to $\phi(\gamma)$ choice. Solving this quadratic inequality with respect to $G_T(\gamma_{\text{lo}}^*)$ (Lemma D.1), we obtain

$$G_T(\gamma_{\text{lo}}^*) \leqslant \delta^t + \delta^{t+1} + \sqrt{\widetilde{\Delta}_T(\gamma_{\text{lo}}^*)L_\infty T} \leqslant \delta^t + \delta^{t+1} + \sqrt{\Delta^* L_\infty T}. \tag{31}$$

Now we make an additional distinction and consider the estimates separately: one case when $\gamma_{\text{lo}}^* > \sqrt{\frac{\Delta^*}{L_\infty T}}$ and another when $\gamma_{\text{lo}}^* \leqslant \sqrt{\frac{\Delta^*}{L_\infty T}}$. We can do this without any limitations, since combining the intervals considered for $\gamma_{\text{lo}}^*$ returns all possible values.

○ $\gamma_{\text{lo}}^* > \sqrt{\frac{\Delta^*}{L_\infty T}}$ : we straightforwardly move to the equation 26 estimation:

$$
\begin{aligned}
\frac{1}{T}\sum_{t=0}^{T-1}\|\nabla f(x^t)\|_1 \quad &\leqslant \quad \frac{1}{T}\frac{\widetilde{\Delta}_T(\gamma_{\text{lo}}^*)}{\gamma_{\text{lo}}^*} + \frac{1}{T}G_T(\gamma_{\text{lo}}^*) + \frac{1}{T}(3\delta^t + \delta^{t+1}) \\
&\leqslant \quad \frac{\sqrt{L_\infty}}{\sqrt{\Delta^* T}}\widetilde{\Delta}_T(\gamma_{\text{lo}}^*) + \frac{1}{T}G_T(\gamma_{\text{lo}}^*) + \frac{1}{T}(3\delta^t + \delta^{t+1}) \\
&\overset{31}{\leqslant} \quad \frac{\sqrt{\Delta^* L_\infty}}{\sqrt{T}} + \frac{\sqrt{\Delta^* L_\infty}}{\sqrt{T}} + \frac{1}{T}(4\delta^t + 2\delta^{t+1}) \\
&= \quad 2\frac{\sqrt{\Delta^* L_\infty}}{\sqrt{T}} + \frac{1}{T}(4\delta^t + 2\delta^{t+1}).
\end{aligned} \tag{32}
$$

○ $\gamma_{\text{lo}}^* \leqslant \sqrt{\frac{\Delta^*}{L_\infty T}}$ : in this case we start from the estimate that is followed by equation 28:

$$G_T(\gamma_{\text{hi}}^*) \leqslant \delta^t + \delta^{t+1} + \gamma_{\text{hi}}^* L_\infty T \overset{(i)}{\leqslant} \delta^t + \delta^{t+1} + 2\gamma_{\text{lo}}^* L_\infty T \leqslant \delta^t + \delta^{t+1} + 2\sqrt{\Delta^* L_\infty T} \tag{33}$$

where $(i)$ is done due to bisection stop condition. Now we proceed to the equation 26 estimation:

$$
\begin{aligned}
\frac{1}{T}\sum_{t=0}^{T-1}\|\nabla f(x^t)\|_1 \quad &\leqslant \quad \frac{1}{T}\frac{\widetilde{\Delta}_T(\gamma_{\text{lo}}^*)}{\gamma_{\text{lo}}^*} + \frac{1}{T}G_T(\gamma_{\text{lo}}^*) + \frac{1}{T}(3\delta^t + \delta^{t+1}) \\
&\overset{\text{Lemma } D.3,9}{\leqslant} \quad \frac{1}{T}\frac{2\mathfrak{D}_T(\gamma_{\text{hi}}^*)}{\mathfrak{N}_T(\gamma_{\text{lo}}^*)}\widetilde{\Delta}_T(\gamma_{\text{lo}}^*) + \frac{1}{T}G_T(\gamma_{\text{lo}}^*) + \frac{1}{T}(3\delta^t + \delta^{t+1}) \\
&\overset{\text{Lemma } D.3,11}{\leqslant} \quad \frac{2}{T}\frac{[G_T(\gamma_{\text{hi}}^*) + \zeta(\gamma_{\text{hi}}^*)]\,\widetilde{\Delta}_T(\gamma_{\text{lo}}^*)}{\widetilde{\Delta}_T(\gamma_{\text{lo}}^*)} + \frac{G_T(\gamma_{\text{hi}}^*) + \zeta(\gamma_{\text{hi}}^*)}{T} \\
&\qquad\qquad + \frac{1}{T}(3\delta^t + \delta^{t+1}) \\
&= \quad \frac{3G_T(\gamma_{\text{hi}}^*)}{T} + \frac{1}{T}(3\delta^t + \delta^{t+1}) + \frac{3\zeta(\gamma_{\text{hi}}^*)}{T} \\
&\overset{33}{\leqslant} \quad 6\frac{\sqrt{\Delta^* L_\infty}}{\sqrt{T}} + \frac{1}{T}(6\delta^t + 4\delta^{t+1}) + \frac{3\zeta(\gamma_{\text{hi}}^*)}{T} \\
&\leqslant \quad 6\frac{\sqrt{\Delta^* L_\infty}}{\sqrt{T}} + \frac{1}{T}(6\delta^t + 4\delta^{t+1}) + \frac{3\|g^0\|_1}{T}.
\end{aligned} \tag{34}
$$

Combining equation 32 and equation 34, we get that equation 34 is the final estimate when BISECTION procedure returns $\gamma_{\text{lo}}^*$.

In the end, equation 30 and equation 34 give us the estimate in the case when BISECTION procedure does not have early terminations at all and outputs any value of $\gamma_0$:

$$\frac{1}{T}\sum_{t=0}^{T-1}\|\nabla f(x^t)\|_1 \leqslant 6\frac{\sqrt{\Delta^* L_\infty}}{\sqrt{T}} + \frac{1}{T}(6\delta^t + 4\delta^{t+1}) + \frac{3\left\|g^0\right\|_1}{T}. \tag{35}$$

**Step 2: early termination with $\gamma_{\text{lo}}$.**

Now we consider the scenario when, with the initial $\gamma_{\text{lo}}$, there is $\gamma_{\text{lo}} \geqslant \phi(\gamma_{\text{lo}})$ and algorithm early returns $\gamma_{\text{lo}}^*$. To consider this, we should choose the initial $\gamma_{\text{lo}} = \gamma_{\text{lo}}^* \leqslant \frac{\Delta^*}{L_\infty T}$. Thus, equation 28 transforms into

$$G_T(\gamma_{\text{lo}}^*) \leqslant \delta^t + \delta^{t+1} + \gamma_{\text{lo}} L_\infty T \leqslant \delta^t + \delta^{t+1} + \sqrt{L_\infty \Delta^* T}. \tag{36}$$

In that way, equation 26 transforms into

$$
\begin{aligned}
\frac{1}{T}\sum_{t=0}^{T-1}\|\nabla f(x^t)\|_1 &\leqslant \frac{1}{T}\frac{\widetilde{\Delta}_T(\gamma_{\text{lo}}^*)}{\gamma_{\text{lo}}^*} + \frac{1}{T}G_T(\gamma_{\text{lo}}^*) + \frac{1}{T}(3\delta^t + \delta^{t+1}) \\
&\leqslant \frac{1}{T}\frac{\widetilde{\Delta}_T(\gamma_{\text{lo}}^*)}{\phi(\gamma_{\text{lo}}^*)} + \frac{1}{T}G_T(\gamma_{\text{lo}}^*) + \frac{1}{T}(3\delta^t + \delta^{t+1}) \\
&= \frac{1}{T}\frac{[G_T(\gamma_{\text{lo}}^*) + \zeta(\gamma_{\text{lo}}^*)]\,\widetilde{\Delta}_T(\gamma_{\text{lo}}^*)}{\widetilde{\Delta}_T(\gamma_{\text{lo}}^*)} + \frac{1}{T}G_T(\gamma_{\text{lo}}^*) + \frac{1}{T}(3\delta^t + \delta^{t+1}) \\
&= \frac{2G_T(\gamma_{\text{lo}}^*)}{T} + \frac{1}{T}(3\delta^t + \delta^{t+1}) + \frac{\zeta(\gamma_{\text{lo}}^*)}{T} \\
&\overset{36}{\leqslant} 2\frac{\sqrt{\Delta^* L_\infty}}{\sqrt{T}} + \frac{1}{T}(5\delta^t + 3\delta^{t+1}) + \frac{\|g^0\|_1}{T}.
\end{aligned}
\tag{37}
$$

In that way, equation 37 delivers the estimate, when Algorithm 4 makes an early termination. Combining equation 35 with equation 37, we finally obtain the estimate for all possible cases of the BISECTION procedure return:

$$\frac{1}{T}\sum_{t=0}^{T-1}\|\nabla f(x^t(\gamma_0))\|_1 \leqslant 6\frac{\sqrt{\Delta^* L_\infty}}{\sqrt{T}} + \frac{1}{T}(6\delta^t + 4\delta^{t+1}) + \frac{3\left\|g^0\right\|_1}{T}. \tag{38}$$

Now it is time to take expectation and give estimate to $\delta^t$. One can note, using the law of total expectation ($\mathbb{E}\left[\xi\right] = \mathbb{E}\left[\mathbb{E}\left[\xi|\psi\right]\right]$),

$$
\begin{aligned}
\mathbb{E}\|\nabla f(x^t) - g^t\|_1 &= \sum_{i=1}^{d}\mathbb{E}\left|\left[\nabla f(x^t)\right]_i - \left[g^t\right]_i\right| \overset{(Jen)}{\leqslant} \sum_{i=1}^{d}\sqrt{\mathbb{E}\left(\left[\nabla f(x^t)\right]_i - \left[g^t\right]_i\right)^2} \\
&= \sum_{i=1}^{d}\sqrt{\mathbb{E}\left[\left(\left[\nabla f(x^t)\right]_i - \left[g^t\right]_i\right)^2 | x^t\right]} \leqslant \sum_{i=1}^{d}\sigma_i^t.
\end{aligned}
$$

In that way, we obtain important bound:

$$\mathbb{E}\|\nabla f(x^t) - g^t\|_1 \leqslant \|\sigma\|_1. \tag{39}$$

Then,

$$
\begin{aligned}
\mathbb{E}\delta^t &= \sum_{t=0}^{T-1}\mathbb{E}\|\nabla f(x^t) - g^t\|_1 \leqslant \sum_{t=0}^{T-1}\|\sigma\|_1 \leqslant \|\sigma\|_1 T, \\
\mathbb{E}\delta^{t+1} &= \sum_{t=0}^{T-1}\mathbb{E}\|\nabla f(x^{t+1}) - g^{t+1}\|_1 \leqslant \sum_{t=0}^{T-1}\|\sigma\|_1 = \|\sigma\|_1 T.
\end{aligned}
$$

Substituting it to equation 38, we have

$$\frac{1}{T}\sum_{t=0}^{T-1}\mathbb{E}\|\nabla f(x^t)\|_1 \leqslant 6\frac{\sqrt{\Delta^* L_\infty}}{\sqrt{T}} + 10\|\sigma\|_1 + \frac{3\mathbb{E}\left\|g^0\right\|_1}{T}.$$

Expressing the number of iterations and using $\varepsilon \geqslant \frac{1}{T}\sum_{t=0}^{T-1}\|\nabla f(x^t)\|_1$ as a criterion, we obtain that algorithm needs $\mathcal{O}\left(\frac{\Delta^* L_\infty}{\varepsilon^2} + \|\sigma\|_1^2\right)$ iterations to reach $\varepsilon$-accuracy. Note the we drop the term $\frac{3\mathbb{E}\|g^0\|_1}{T}$, since it is asymptotically smaller than the main one. However we firstly need to find step $\gamma_0$ with bisection procedure that takes $T \log\log\left(\frac{\gamma_\varepsilon 2^{2^k}}{\gamma_\varepsilon}\right) = \mathcal{O}\left(\left(\frac{\Delta^* L_\infty}{\varepsilon^2} + \|\sigma\|_1^2\right)k\right)$ iterations, where $2^{2^k}$ denotes the length of the initial interval for the stepsize. We have already discussed in the main part that, according to Lemma D.2, $k$ should be at least $k = \log\log\frac{\Delta^*}{\gamma_s\|g^0\|_1}$. Thus, $\mathcal{O}\left(\left(\frac{\Delta^* L_\infty}{\varepsilon^2} + \|\sigma\|_1^2\right)\log\log\frac{\Delta^*}{\gamma_s\|g^0\|_1}\right)$ is the final iteration complexity. $\qquad\square$

*Remark* E.5. Under conditions of Theorem E.4 Algorithm 5 with mini-batch of the size $t+1$ at $t$-th iteration to reach $\varepsilon$-accuracy needs

$$\mathcal{O}\left(\frac{\Delta^* L_\infty + \|\sigma\|_1^2}{\varepsilon^2}\log\log\frac{\Delta^*}{\gamma_s\|g^0\|_1}\right) \quad \text{iterations.}$$

*Proof.* The proof of the remark repeats the proof of Theorem 3.9 except for the estimate on $\mathbb{E}\|\nabla f(x^t) - g^t\|_1^2$ term. Since we now use mini-batches, we can bound

$$\mathbb{E}\left\|\nabla f(x^t) - g^t\right\|_1^2 \leqslant \frac{\|\sigma\|_1}{\sqrt{t+1}},$$

instead of equation 39. In that way,

$$\frac{1}{T}\mathbb{E}\delta^t = \frac{1}{T}\sum_{t=0}^{T-1}\mathbb{E}\|\nabla f(x^t) - g^t\|_1 \leqslant \frac{1}{T}\sum_{t=0}^{T-1}\frac{\|\sigma\|_1}{t+1} \leqslant 2\frac{\|\sigma\|_1}{\sqrt{T}},$$

which ends the proof of the remark. $\qquad\square$

### E.3 DISTRIBUTED SETTING

To begin with, we present the modification of the classic SIGN-SGD algorithm (Algorithm 1) that aligns with the distributed learning. We consider SIGN-SGD with majority vote (Algorithm 6), similarly to (Bernstein et al., 2018). We present the assumption that we utilize in distributed regime.

**Assumption E.6.** In the multi-node regime of learning each node $j = \overline{1, M}$ at any point $x \in \mathbb{R}^d$ has an access to the stochastic gradient, i.e., it can compute $g_j(x) = \nabla f(x, \xi_j)$ – the stochastic gradient value with respect to the randomness in the choice fo samples $\xi_j$. Additionally, this stochastic estimators is unbiased, i.e., $\mathbb{E}[g_j(x)] = \nabla f(x)$, and its variance is coordinate-wise bounded, i.e., $\mathbb{E}\left([g_j(x)]_i - [\nabla f(x)]_i\right) \leqslant \sigma_i^2$.

---

**Algorithm 6** SIGN-SGD with majority vote

---

1: **Input:** Start point $x^0 \in \mathbb{R}^d$, number of iterations $T$
2: **Parameter:** Stepsize $\gamma > 0$
3: **for** $t = 0, \dots, T-1$ **do**
4:     **for** all nodes $j = 1, \dots, M$ in parallel **do**
5:         Compute stochastic gradient $g_j(x^t) = \nabla f(x^t, \xi_j)$
6:         Send $\text{sign}(g_j(x^t))$ to the server
7:     **end for**
8:     $x^{t+1} = x^t - \gamma\text{sign}\left(\sum_{j=1}^M \text{sign}(g_j(x^t))\right)$
9: **end for**

---

Proceeding analogically to the stochastic one-node regime, we establish $\mathfrak{N}_T(\gamma)$ and $\mathfrak{D}_T(\gamma)$ that we use in $\phi(\gamma)$ in the BISECTION procedure: $\mathfrak{N}_T(\gamma) = \widetilde{\Delta}_T(\gamma), \mathfrak{D}_T(\gamma) =$

$\sum_{t=0}^{T-1} \frac{1}{M} \sum_{j=1}^{M} \left( \|g_j(x^{t+1}) - g_j(x^t)\|_1 + \zeta(\gamma) \right)$. Let us emphasize how this affects Algorithms 4, 5. Firstly, we now need to call the SIGN-SGD with majority vote method (Algorithm 6) instead of SIGN-SGD (Algorithm 1). Secondly, to obtain $\mathfrak{D}_T(\gamma)$ in the bisection procedure, each node $j$ counts $\sum_{t=0}^{T-1} \|g_j(x^{t+1}) - g_j(x^t)\|_1$ using locally stored gradients, and sends the complete sum to the server in the end. Note that this requirement has no effect on extra memory and communication complexity, since each device requires only $\mathcal{O}(d)$ extra memory and performs only one extra communication during the whole learning. Now we present the theoretical result for the distributed setting.

**Lemma E.7** (Theorem 2 (a) from (Bernstein et al., 2018)). *Suppose Assumption E.6 holds. Then, at any point $x \in \mathbb{R}^d$, the following estimate is valid:*

$$
|[\nabla f(x)]_i| \, \mathbb{P} \left( sign \left( \sum_{j=1}^{M} sign \left( [g_j(x)]_i \right) \right) \neq sign \left( [\nabla f(x)]_i \right) \right) \leqslant \sigma_i.
$$

**Lemma E.8** (Descent lemma). *For Algorithm 5 under Assumptions 3.1, 3.2, 3.3, E.6, the following estimate is valid:*

$$
\sum_{t=0}^{T-1} \|\nabla f(x^t)\|_1 \quad \leqslant \quad \frac{f(x^{-1}) - f(x^T)}{\gamma_0} + \sum_{t=0}^{T-1} \frac{1}{M} \sum_{j=1}^{M} \|g_j^{t+1} - g_j^t\|_1 + 2\widetilde{\delta}^T + \delta^t + \delta^{t+1},
$$

*where $\delta^t = \sum\limits_{t=0}^{T-1} \frac{1}{M} \sum\limits_{j=1}^{M} \|\nabla f(x^t) - g_j^t\|_1$*

*and $\widetilde{\delta}^T = \sum\limits_{t=0}^{T-1} \sum\limits_{i=1}^{d} |[\nabla f(x^t)]_i| \, \mathbb{I} \left( sign \left( \sum\limits_{j=1}^{M} sign \left( [g_j^t]_i \right) \right) \neq sign \left( [\nabla f(x^t)]_i \right) \right).$*

*Proof.* Starting from the convexity of the objective,

$$
f(x^{t+1}) - f(x^t) \quad \leqslant \quad \langle \nabla f(x^{t+1}), x^{t+1} - x^t \rangle = -\gamma^t \left\langle \nabla f(x^{t+1}), sign \left( \sum_{j=1}^{M} sign(g_j^t) \right) \right\rangle
$$

$$
= \quad -\gamma^t \left\langle \nabla f(x^t), sign \left( \sum_{j=1}^{M} sign(g_j^t) \right) \right\rangle
$$

$$
-\gamma^t \left\langle \nabla f(x^{t+1}) - \nabla f(x^t), sign \left( \sum_{j=1}^{M} sign(g_j^t) \right) \right\rangle
$$

$$
= \quad -\gamma^t \|\nabla f(x^t)\|_1 + 2\gamma^t \sum_{i=1}^{d} \left| [\nabla f(x^t)]_i \right|
$$

$$
\cdot \mathbb{I} \left( sign \left( \sum_{j=1}^{M} sign \left( [g_j^t]_i \right) \right) \neq sign \left( [\nabla f(x^t)]_i \right) \right)
$$

$$
-\gamma^t \left\langle \nabla f(x^{t+1}) - \nabla f(x^t), sign \left( \sum_{j=1}^{M} sign(g_j^t) \right) \right\rangle
$$

$$
\overset{Conj.,(i)}{\leqslant} \quad -\gamma^t \|\nabla f(x^t)\|_1 + 2\gamma^t \widetilde{\delta}^t
$$

$$
+\gamma^t \|\nabla f(x^{t+1}) - \nabla f(x^t)\|_1 \left\| sign \left( \sum_{j=1}^{M} sign \left( g_j^t \right) \right) \right\|_\infty
$$

$$
\leqslant \quad -\gamma^t \|\nabla f(x^t)\|_1 + 2\gamma^t \widetilde{\delta}^t + \gamma^t \|\nabla f(x^{t+1}) - \nabla f(x^t)\|_1
$$

$$
= \quad -\gamma^t \|\nabla f(x^t)\|_1 + 2\gamma^t \widetilde{\delta}^t + \gamma^t \frac{1}{M} \sum_{j=1}^{M} \|\nabla f(x^{t+1}) - \nabla f(x^t)\|_1
$$

$$\overset{CS}{\leqslant} \quad -\gamma^t \|\nabla f(x^t)\|_1 + 2\gamma^t \widetilde{\delta}^t + \gamma^t \frac{1}{M} \sum_{j=1}^M \|g_j^{t+1} - g_j^t\|_1$$

$$+ \gamma^t \frac{1}{M} \sum_{j=1}^M \|\nabla f(x^{t+1}) - g_j^{t+1}\|_1 + \gamma^t \frac{1}{M} \sum_{j=1}^M \|\nabla f(x^t) - g_j^t\|_1, \quad (40)$$

where in ($i$) we denote $\widetilde{\delta}^t = \sum_{i=1}^d |[\nabla f(x^t)]_i| \, \mathbb{I}\left( \text{sign}\left( \sum_{j=1}^M \text{sign}\left( [g_j^t]_i \right) \right) \neq \text{sign}\left( [\nabla f(x^t)]_i \right) \right)$. Now we rearrange terms and summarize over all iterations to obtain

$$\sum_{t=0}^{T-1} \gamma^t \|\nabla f(x^t)\|_1 \quad \leqslant \quad \sum_{t=0}^{T-1} \left[ f(x^t) - f(x^{t+1}) \right] + 2 \sum_{t=0}^{T-1} \gamma^t \widetilde{\delta}^t + \sum_{t=0}^{T-1} \frac{1}{M} \sum_{j=1}^M \gamma^t \|g_j^{t+1} - g_j^t\|_1$$

$$+ \sum_{t=0}^{T-1} \frac{1}{M} \sum_{j=1}^M \gamma^t \|\nabla f(x^t) - g_j^t\|_1 + \sum_{t=0}^{T-1} \frac{1}{M} \sum_{j=1}^M \gamma^t \|\nabla f(x^{t+1}) - g_j^{t+1}\|_1.$$

Since Algorithm 5 performs all the steps with the constant rate $\gamma_0$, which we define later, denoting $\widetilde{\delta}^T = \sum_{t=0}^{T-1} \widetilde{\delta}^t$, we can rewrite the result in the following form:

$$\sum_{t=0}^{T-1} \|\nabla f(x^t)\|_1 \quad \leqslant \quad \sum_{t=0}^{T-1} \frac{\left[ f(x^t) - f(x^{t+1}) \right]}{\gamma_0} + 2\widetilde{\delta}^T + \sum_{t=0}^{T-1} \frac{1}{M} \sum_{j=1}^M \|g_j^{t+1} - g_j^t\|_1$$

$$+ \sum_{t=0}^{T-1} \frac{1}{M} \sum_{j=1}^M \|\nabla f(x^t) - g_j^t\|_1 + \sum_{t=0}^{T-1} \frac{1}{M} \sum_{j=1}^M \|\nabla f(x^{t+1}) - g_j^{t+1}\|_1.$$

In the obtained estimate the last two terms consist from differences between the honest and stochastic gradient at the $t$-th and $(t+1)$-th moments. One of our goals is to estimate them, however, we want to perform analogically to Theorem E.4 and continue the proof with the $\sum_{t=0}^{T-1} \frac{1}{M} \sum_{j=1}^M \|g_j^{t+1} - g_j^t\|_1$ term estimate. To simplify the subsequent notation, we introduce the following definition: let $\delta^t = \sum_{t=0}^{T-1} \frac{1}{M} \sum_{j=1}^M \|\nabla f(x^t) - g_j^t\|_1$. In that way, the following inequality finishes the proof of the lemma:

$$\sum_{t=0}^{T-1} \|\nabla f(x^t)\|_1 \quad \leqslant \quad \frac{f(x^{-1}) - f(x^T)}{\gamma_0} + \sum_{t=0}^{T-1} \frac{1}{M} \sum_{j=1}^M \|g_j^{t+1} - g_j^t\|_1 + 2\widetilde{\delta}^T + \delta^t + \delta^{t+1}.$$

$\square$

**Theorem E.9.** *Suppose Assumptions 3.1, 3.2, 3.3, E.6 hold. Then for Algorithm 5 using at $t$-th iteration mini-batches of sizes $t+1$, after obtaining the stepsize $\gamma_0$, the following estimate is valid:*

$$\frac{1}{T} \sum_{t=0}^{T-1} \mathbb{E}\|\nabla f(x^t)\|_1 \quad \leqslant \quad 6\frac{\sqrt{\Delta^* L_\infty}}{\sqrt{T}} + 10\|\sigma\|_+ + \frac{\frac{3}{M} \sum_{j=1}^M \mathbb{E}\left\|g_j^0\right\|_1}{T}.$$

*Moreover, taking into account the complexity of Algorithm 4 in relation to the initial stepsize bound $\gamma_s$, to reach $\varepsilon$-accuracy, where $\varepsilon \geqslant \frac{1}{T} \sum_{t=0}^{T-1} \|\nabla f(x^t)\|_1$, Algorithm 5 needs*

$$\mathcal{O}\left( \left( \frac{\Delta^* L_\infty}{\varepsilon^2} + \|\sigma\|_1^2 \right) \log\log \frac{\Delta^*}{\gamma_s \sum_{j=1}^M \left\|g_j^0\right\|_1} \right) \quad \text{iterations.}$$

*Proof.* Let us mention that the result of Lemma E.8 almost matches the starting point of Theorem E.4 equation 26. If we now denote $G_T = \sum_{t=0}^{T-1} \frac{1}{M} \sum_{j=1}^{M} \|g_j^{t+1} - g_j^t\|_1$, the only difference is that there we have an additional $2\widetilde{\delta}^T$ term. However, we do not estimate it yet and it does not require any transformations. Thus, we can proceed in a manner completely analogous to the proof of Theorem E.4 and obtain an analog of the estimate in equation 38:

$$\frac{1}{T} \sum_{t=0}^{T-1} \|\nabla f(x^t(\gamma_0))\|_1 \leqslant 6\frac{\sqrt{\Delta^* L_\infty}}{\sqrt{T}} + \frac{1}{T}(2\widetilde{\delta}^T + 4\delta^t + 4\delta^{t+1}) + \frac{\frac{3}{M} \sum_{j=1}^{M} \|g_j^0\|_1}{T}, \qquad (41)$$

where $\Delta^* = f(x^{-1}) - f(x^*)$. Now we take expectation and use Lemma E.7 to obtain

$$\mathbb{E}\widetilde{\delta}^t = \sum_{i=1}^{d} \left| \left[\nabla f(x^t)\right]_i \right| \mathbb{P}\left( \text{sign}\left( \sum_{j=1}^{M} \text{sign}\left( \left[g_j^t\right]_i \right) \right) \neq \text{sign}\left( \left[\nabla f(x^t)\right]_i \right) \right)$$

$$\leqslant \sum_{i=1}^{d} \sigma_i^t = \|\sigma\|_1. \qquad (42)$$

For $\mathbb{E}\delta^t$, under Assumption E.6, we have the estimate as equation 39:

$$\mathbb{E}\|\nabla f(x^t) - g_j^t\|_1 \quad \leqslant \quad \|\sigma\|_1.$$

Thus, substituting both of these estimates to equation 41, we obtain the final convergence result:

$$\frac{1}{T} \sum_{t=0}^{T-1} \mathbb{E}\|\nabla f(x^t)\|_1 \quad \leqslant \quad 6\frac{\sqrt{\Delta^* L_\infty}}{\sqrt{T}} + \frac{1}{M} \sum_{j=1}^{M} \frac{1}{T} \sum_{t=0}^{T-1} 10\|\sigma\|_1 + \frac{\frac{3}{M} \sum_{j=1}^{M} \mathbb{E}\left\|g_j^0\right\|_1}{T}$$

$$= \quad 6\frac{\sqrt{\Delta^* L_\infty}}{\sqrt{T}} + 10\|\sigma\|_1 + \frac{\frac{3}{M} \sum_{j=1}^{M} \mathbb{E}\left\|g_j^0\right\|_1}{T}.$$

Since we obtain the same convergence estimate as in Theorem E.4, we can analogically establish the

$$\mathcal{O}\left( \left(\frac{\Delta^* L_\infty}{\varepsilon^2} + \|\sigma\|_1^2\right) \log\log \frac{\Delta^*}{\gamma_s \frac{1}{M} \sum_{j=1}^{M} \|g_j^0\|_1} \right) \text{ iteration complexity.} \qquad \square$$

*Remark* E.10. Under conditions of Theorem E.9 Algorithm 5 with mini-batches of the size $t+1$ at $t$-th iteration to reach $\varepsilon$-accuracy needs

$$\mathcal{O}\left( \frac{\Delta^* L_\infty + \|\sigma\|_1^2}{\varepsilon^2} \log\log \frac{\Delta^*}{\gamma_s \frac{1}{M} \sum_{j=1}^{M} \|g_j^0\|_1} \right) \text{ iterations.}$$

*Proof.* Proof repeats the proofs of Remark E.5. $\qquad \square$

# F PROOFS FOR ALIAS

## F.1 EXACT GRADIENT SETTING

**Lemma F.1** (Approximating sequence). *Let the initial $\Delta^*$-approximation $d^0$ be $0 < d^0 < \Delta^*$, where $\Delta^* = f(x^0) - f(x^*)$. Then for Algorithm 2 under Assumptions 3.1, 3.2, 3.3, 3.4, the following estimate is valid:*

$$\Delta^* \geqslant d^n \ \forall n \in [0, T-1].$$

*Proof.* Starting from the convexity of the objective,

$$f(x^{t+1}) - f(x^t) \leqslant \langle \nabla f(x^{t+1}), x^{t+1} - x^t \rangle = -\gamma^t \langle \nabla f(x^{t+1}), \text{sign}\left(\nabla f(x^t)\right) \rangle. \qquad (43)$$

Now we summarize both sides over the first $n$ iterations:

$$-\Delta^* = f(x^*) - f(x^0) \overset{(i)}{\leqslant} f(x^n) - f(x^0) = \sum_{t=0}^{n-1} f(x^{t+1}) - f(x^t)$$

$$\overset{43}{\leqslant} -\sum_{t=0}^{n-1} \gamma^t \langle \nabla f(x^{t+1}), \text{sign}\left(\nabla f(x^t)\right) \rangle,$$

where (*i*) is correct due to Assumption 3.3. Changing the sign of the inequality,

$$\widetilde{d}^n = \sum_{t=0}^{n-1} \gamma^t \langle \nabla f(x^{t+1}), \text{sign}\left(\nabla f(x^t)\right) \rangle \leqslant \Delta^*.$$

Since our algorithm performs $d^n = \max\left(d^{n-1}, \widetilde{d}^n\right)$ and we initialize our sequence with $d^0 < \Delta^*$, we obtain the required statement. $\qquad \square$

**Lemma F.2** (Descent lemma). *For Algorithm 2 under Assumptions 3.1, 3.2, 3.3, 3.4, the following estimate is valid:*

$$\sum_{t=0}^{T-1} \gamma^t \left\| \nabla f(x^t) \right\|_1 \leqslant \Delta^* + \sum_{t=0}^{T-1} (\gamma^t)^2 L_\infty^t,$$

*where $L_\infty^t = \frac{\left\| \nabla f(x^{t+1}) - \nabla f(x^t) \right\|_1}{\left\| x^{t+1} - x^t \right\|_\infty}$.*

*Proof.*

$$
\begin{aligned}
f(x^{t+1}) \quad &\leqslant \quad f(x^t) + \langle \nabla f(x^{t+1}), x^{t+1} - x^t \rangle = f(x^t) - \gamma^t \langle \nabla f(x^{t+1}), \text{sign}\left(\nabla f(x^t)\right) \rangle \\
&= \quad f(x^t) - \gamma^t \left\| \nabla f(x^t) \right\|_1 - \gamma^t \langle \nabla f(x^{t+1}) - \nabla f(x^t), \text{sign}\left(\nabla f(x^t)\right) \rangle \\
&\overset{Conj}{\leqslant} \quad f(x^t) - \gamma^t \left\| \nabla f(x^t) \right\|_1 + \gamma^t \left\| \nabla f(x^{t+1}) - \nabla f(x^t) \right\|_1 \left\| \text{sign}\left(\nabla f(x^t)\right) \right\|_\infty \\
&\leqslant \quad f(x^t) - \gamma^t \left\| \nabla f(x^t) \right\|_1 + \gamma^t \left\| \nabla f(x^{t+1}) - \nabla f(x^t) \right\|_1 \\
&\overset{(i)}{=} \quad f(x^t) - \gamma^t \left\| \nabla f(x^t) \right\|_1 + \gamma^t \frac{\left\| \nabla f(x^{t+1}) - \nabla f(x^t) \right\|_1}{\left\| x^{t+1} - x^t \right\|_\infty} \left\| x^{t+1} - x^t \right\|_\infty \\
&= \quad f(x^t) - \gamma^t \left\| \nabla f(x^t) \right\|_1 + (\gamma^t)^2 \frac{\left\| \nabla f(x^{t+1}) - \nabla f(x^t) \right\|_1}{\left\| x^{t+1} - x^t \right\|_\infty},
\end{aligned}
$$

where in (*i*) we assume $\left\| x^{t+1} - x^t \right\|_\infty \neq 0$. Indeed, $\left\| x^{t+1} - x^t \right\|_\infty = 0$ follows from the equality sign $\left(\nabla f(x^t)\right) = 0$, which means that we find the optimum and do need to find another point $x^{t+1}$. Now we denote $L_\infty^t = \frac{\left\| \nabla f(x^{t+1}) - \nabla f(x^t) \right\|_1}{\left\| x^{t+1} - x^t \right\|_\infty}$. Summing over all iterations, we obtain

$$
\begin{aligned}
\sum_{t=0}^{T-1} \gamma^t \left\| f(x^t) \right\|_1 \quad &\leqslant \quad \sum_{t=0}^{T-1} \left[ f(x^t) - f(x^{t+1}) \right] + \sum_{t=0}^{T-1} (\gamma^t)^2 L_\infty^t \\
&= \quad f(x^0) - f(x^*) + \sum_{t=0}^{T-1} (\gamma^t)^2 L_\infty^t \leqslant \Delta^* + \sum_{t=0}^{T-1} (\gamma^t)^2 L_\infty^t,
\end{aligned}
$$

which ends the proof of the lemma. $\qquad \square$

**Theorem F.3** (**Theorem 3.5**). *Suppose Assumptions 3.1, 3.2, 3.3, 3.4 hold. We denote $\varepsilon \geqslant \frac{1}{T} \sum_{t=0}^{T-1} \left\| \nabla f(x^t) \right\|_1, L_\infty^t = \frac{\left\| \nabla f(x^{t+1}) - \nabla f(x^t) \right\|_1}{\left\| x^{t+1} - x^t \right\|_\infty}$. Then Algorithm 2 with Option I, $d^0 < \Delta^*$ to reach $\varepsilon$-accuracy needs*

$$\widetilde{\mathcal{O}}\left( \frac{(\Delta^*)^2 (L_\infty)^3}{d^0 (L_\infty^0)^2 \varepsilon^2} \right) \quad \textit{iterations.}$$

*Algorithm 2 with Option II to reach $\varepsilon$-accuracy needs*

$$\widetilde{\mathcal{O}}\left(\frac{\Delta^* (L_\infty)^3}{(L_\infty^0)^2 \varepsilon^2}\right) \quad iterations.$$

*Proof.* Let us start with the result of Lemma F.2:

$$\sum_{t=0}^{T-1} \gamma^t \|\nabla f(x^t)\|_1 \leqslant \Delta^* + \sum_{t=0}^{T-1} (\gamma^t)^2 L_\infty^t. \tag{44}$$

Now we use our $\gamma^t$ choice. Let us firstly estimate the denominator that is exactly $\lambda^t = \frac{1}{\sqrt{\sum_{i=0}^{t-1} \frac{\|\nabla f(x^{i+1}) - \nabla f(x^i)\|_1}{\|x^{i+1} - x^i\|_\infty}}} = \frac{1}{\sqrt{\sum_{i=0}^{t-1} L_\infty^i}}$ and is the same for both Options I and II. Let us estimate the following term.

$$\sum_{t=0}^{T-1} (\lambda^t)^2 L_\infty^t = \sum_{t=0}^{T-1} \frac{L_\infty^t}{\sum_{i=0}^{t-1} L_\infty^i}.$$

We mention, that each $L_\infty^i$ is bounded from the definition of smoothness (see Assumption 3.1), i.e., $L_\infty^i \leqslant L_\infty$. We consider the sequence $\{L_\infty^i\}_{i=0}^{T-1}$. Since each term in this sequence is bounded, there exists $r$ such that $\sum_{i=0}^{r-2} L_\infty^i \leqslant L_\infty^{r-1}$ and for each $t \geqslant r-1$ such that $\sum_{i=0}^{t} L_\infty^i \geqslant L_\infty^{t+1}$. In that way, we divide the sum into two parts:

$$\sum_{t=0}^{T-1} \frac{L_\infty^t}{\sum_{i=0}^{t-1} L_\infty^i} = \sum_{t=0}^{r-1} \frac{L_\infty^t}{\sum_{i=0}^{t-1} L_\infty^i} + \sum_{t=r}^{T-1} \frac{L_\infty^t}{\sum_{i=0}^{t-1} L_\infty^i}. \tag{45}$$

Considering the first sum in equation 45, we mention, that we can estimate the denominator as $\sum_{i=0}^{t-1} L_\infty^i \geqslant L_\infty^0$. As for the numerator. Thus,

$$\sum_{t=0}^{r-1} \frac{L_\infty^t}{\sum_{i=0}^{t-1} L_\infty^i} \leqslant \frac{1}{L_\infty^0}\left(\sum_{t=0}^{r-2} L_\infty^t + L_\infty^{r-1}\right) \leqslant \frac{2L_\infty^{r-1}}{L_\infty^0} \leqslant \frac{2L_\infty}{L_\infty^0}. \tag{46}$$

Considering the second sum in equation 45, we have

$$\sum_{t=r}^{T-1} \frac{L_\infty^t}{\sum_{i=0}^{t-1} L_\infty^i} = \sum_{t=r}^{T-1} \frac{L_\infty^t}{\frac{1}{2}\sum_{i=0}^{t-1} L_\infty^i + \frac{1}{2}\sum_{i=0}^{t-1} L_\infty^i}.$$

Estimating any of the sums in the denominator, we claim, that $\sum_{i=0}^{t-1} L_\infty^i \geqslant L_\infty^t$, since $t-1 \geqslant r-1$. In that way,

$$\sum_{t=r}^{T-1} \frac{L_\infty^t}{\sum_{i=0}^{t-1} L_\infty^i} \leqslant \sum_{t=r}^{T-1} \frac{2L_\infty^t}{\sum_{i=0}^{t} L_\infty^i} \leqslant 2\sum_{t=0}^{T-1} \frac{L_\infty^t}{\sum_{i=0}^{t} L_\infty^i}. \tag{47}$$

Next we denote $s^t = \sum\limits_{i=0}^{t} L_\infty^t$ and have

$$L_\infty^t \frac{1}{\sum\limits_{i=0}^{t} L_\infty^i} = (s^t - s^{t-1})\frac{1}{\sum\limits_{i=0}^{t} L_\infty^i} = \int\limits_{s^{t-1}}^{s^t} \frac{1}{\sum\limits_{i=0}^{t} L_\infty^i} dx \overset{(i)}{\leqslant} \int\limits_{s^{t-1}}^{s^t} \frac{1}{x} dx, \tag{48}$$

where $(i)$ was done due to $\frac{1}{x}$ is a non-increasing function on $(0, +\infty)$. Summing over $t$, we obtain

$$2\sum_{t=1}^{T} \frac{L_\infty^t}{\sum\limits_{i=0}^{t} L_\infty^i} \leqslant 2\int\limits_{s^0}^{s^T} \frac{1}{x} dx = 2\log(s^T) - 2\log(s^0) = 2\log\left(\frac{\sum\limits_{t=1}^{T} L_\infty^t}{L_\infty^0}\right) \leqslant 2\log\left(\frac{L_\infty T}{L_\infty^0}\right).$$

Combining this estimate with equation 47,

$$\sum_{t=r}^{T-1} \frac{L_\infty^t}{\sum\limits_{i=0}^{t-1} L_\infty^i} \leqslant 2\sum_{t=1}^{T} \frac{L_\infty^t}{\sum\limits_{i=0}^{t} L_\infty^i} + 2 \leqslant 2\left(\log\left(\frac{L_\infty T}{L_\infty^0}\right) + 1\right) \leqslant 4\log\left(\frac{L_\infty T}{L_\infty^0}\right). \tag{49}$$

Substituting equation 46 and equation 49 into equation 45, we obtain

$$\sum_{t=0}^{T-1} (\lambda^t)^2 L_\infty^t \leqslant 2\frac{L_\infty}{L_\infty^0} + 4\log\left(\frac{L_\infty T}{L_\infty^0}\right). \tag{50}$$

We additionally note, that if $r > T - 1$, only first term remains in this estimate, consequently our bound equation 50 is correct.

In this way, utilizing Option I from Algorithm 2, equation 44 together with equation 50 yields

$$\sqrt{d^0}\lambda^{T-1}\sum_{t=0}^{T-1}\|\nabla f(x^t)\|_1 \overset{(i)}{\leqslant} \sum_{t=0}^{T-1}\sqrt{d^t}\lambda^t\|\nabla f(x^t)\|_1 \leqslant \Delta^* + \sum_{t=0}^{T-1} d^t(\lambda^t)^2 L_\infty^t$$

$$\overset{\text{Lemma } F.1}{\leqslant} \Delta^* + \Delta^*\sum_{t=0}^{T-1}(\lambda^t)^2 L_\infty^t,$$

$$\sum_{t=0}^{T-1}\|\nabla f(x^t)\|_1 \leqslant \frac{\Delta^*}{\sqrt{d^0}\lambda^{T-1}} + \frac{\Delta^*}{\sqrt{d^0}\lambda^{T-1}}\sum_{t=0}^{T-1}(\lambda^t)^2 L_\infty^t$$

$$\overset{50}{\leqslant} \frac{\Delta^*}{\sqrt{d^0}\lambda^{T-1}} + 4\frac{\Delta^*}{\sqrt{d^0}\lambda^{T-1}}\log\left(\frac{L_\infty T}{L_\infty^0}\right) + 2\frac{\Delta^* L_\infty}{\sqrt{d^0}\lambda^{T-1}L_\infty^0}$$

$$\leqslant 7\frac{\Delta^* L_\infty}{\sqrt{d^0}\lambda^{T-1}L_\infty^0}\log\left(\frac{L_\infty T}{L_\infty^0}\right), \tag{51}$$

where $(i)$ was done due to the fact that $d^0$ is minimal from all $\{d^t\}_{t=0}^{T-1}$ (Line 7 from Algorithm 2) and the definition of $\lambda^t$. Utilizing $\frac{1}{\lambda^{T-1}} = \sqrt{\sum\limits_{t=0}^{T-2} L_\infty^t} \leqslant \sqrt{L_\infty T}$, we obtain the final estimate:

$$\frac{1}{T}\sum_{t=0}^{T-1}\|\nabla f(x^t)\|_1 \leqslant \frac{7\Delta^*(L_\infty)^{\frac{3}{2}}}{\sqrt{d^0 T}L_\infty^0}\log\left(\frac{L_\infty T}{L_\infty^0}\right).$$

Expressing the number of iterations and using $\varepsilon \geqslant \frac{1}{T}\sum\limits_{t=0}^{T-1}\|\nabla f(x^t)\|_1$ as a criterion, we obtain that the algorithm needs $\widetilde{\mathcal{O}}\left(\frac{(\Delta^*)^2(L_\infty)^3}{d^0(L_\infty^0)^2\varepsilon^2}\right)$ iterations to reach $\varepsilon$-accuracy.

Considering Option II from Algorithm 2, we can proceed absolutely analogical, however, using $f(x^0) - \widetilde{f} \geqslant \Delta^*$ instead of Lemma F.1. In that way,

$$
\begin{aligned}
\frac{1}{T} \sum_{t=0}^{T-1} \|\nabla f(x^t)\|_1 \quad &\leqslant \quad \frac{\Delta^* \sqrt{L_\infty}}{\sqrt{(f(x^0) - \widetilde{f})T}} + \frac{4(f(x^0) - \widetilde{f})\sqrt{L_\infty}}{\sqrt{(f(x^0) - \widetilde{f})T}} \log\left(\frac{L_\infty T}{L_\infty^0}\right) \\
&\quad + \frac{2(f(x^0) - \widetilde{f})(L_\infty)^{\frac{3}{2}}}{\sqrt{(f(x^0) - \widetilde{f})T L_\infty^0}} \\
&\leqslant \quad \frac{7\sqrt{(f(x^0) - \widetilde{f})}(L_\infty)^{\frac{3}{2}}}{\sqrt{T} L_\infty^0} \log\left(\frac{L_\infty T}{L_\infty^0}\right).
\end{aligned}
$$

Expressing the number of iterations, using $\varepsilon \geqslant \frac{1}{T} \sum_{t=0}^{T-1} \|\nabla f(x^t)\|_1$ as a criterion, and utilizing $\widetilde{f}$ is an approximation of $f(x^*)$, we obtain that the algorithm needs $\widetilde{\mathcal{O}}\left(\frac{\Delta^*(L_\infty)^3}{(L_\infty^0)^2 \varepsilon^2}\right)$ iterations to reach $\varepsilon$-accuracy. $\qquad\square$

*Remark* F.4 (Remark 3.6). Under conditions of Theorem 3.5 Algorithm 2 with $\lambda^t = \frac{1}{\sqrt{L_\infty + \sum_{i=0}^{t-1} \frac{\|\nabla f(x^{i+1}) - \nabla f(x^i)\|_1}{\|x^{i+1} - x^i\|_\infty}}}$ and Option II to reach $\varepsilon$-accuracy needs

$$
\widetilde{\mathcal{O}}\left(\frac{\Delta^* L_\infty}{\varepsilon^2}\right) \quad \text{iterations,}
$$

where $\varepsilon \geqslant \frac{1}{T} \sum_{t=0}^{T-1} \|\nabla f(x^t)\|_1$.

*Proof.* The proof of the remark repeats the proof of Theorem 3.5 except for the estimate on $\sum_{t=0}^{T-1} (\lambda^t)^2 L_\infty^t$ term. Let us derive it. We use definition $L_\infty^t = \frac{\|\nabla f(x^{t+1}) - \nabla f(x^t)\|_1}{\|x^{t+1} - x^t\|_\infty}$.

$$
\sum_{t=0}^{T-1} (\lambda^t)^2 L_\infty^t = \sum_{t=0}^{T-1} \frac{L_\infty^t}{L_\infty + \sum_{i=0}^{t-1} L_\infty^i} \leqslant \sum_{t=0}^{T-1} \frac{L_\infty^t}{\sum_{i=0}^{t} L_\infty^i}.
$$

Continuing analogically to equation 48 - equation 49, we get

$$
\sum_{t=0}^{T-1} (\lambda^t)^2 L_\infty^t \leqslant 2 \log\left(\frac{L_\infty T}{L_\infty^0}\right).
$$

Substituting this bound into equation 51 instead of equation 50, we ends the proof of the remark. $\quad\square$

### F.2 STOCHASTIC GRADIENT SETTING

In this section we denote $g_{\xi^t}^t$ the stochastic gradient at the $t$-th iteration (point $x^t$), according to the stochastic realization $\xi^t$ at the $t$-th iteration.

Before proceeding to the theoretical analysis of the algorithm, we present its formal description, Algorithm 7, specifying which option for the sequence $d^t$ we use in practice and which one we analyze theoretically.

---

**Algorithm 7** ALIAS stochastic version

---

1: **Input:** Starting point $x^0 \in \mathbb{R}^d$, initial $L_\infty$-approximation $\eta^{-1} = 0$, initial $\Delta^*$-approximation $d^0$
 $\in \mathbb{R}_+$, lower bound $\widetilde{f}$ on $f(x^*)$, number of iterations $T$
2: **for** $t = 0, \ldots, T-1$ **do**
3: $\quad$ Compute gradients $g_{\xi^t}^t, g_{\xi^t}^{t-1}$
4: $\quad \eta^t = \eta^{t-1} + \frac{\left\| g_{\xi^t}^t - g_{\xi^t}^{t-1} \right\|_1}{\left\| x^t - x^{t-1} \right\|_\infty}; \ \lambda^t = \frac{1}{\sqrt{\eta^t}}$
5: $\quad$ **if** $t \neq 0$ **then**
6: $\quad\quad \widetilde{d}^t = \sum_{i=0}^{t-1} \gamma^i \langle g_{\xi^{i+1}}^{i+1}, \text{sign}(g_{\xi^{i+1}}^i) \rangle$
7: $\quad\quad d^t = \max \left( d^{t-1}, \widetilde{d}^t \right)$
8: $\quad$ **end if**
9: $\quad$ Option I (Practical): $\gamma^t = \lambda^t \sqrt{d^t}$
10: $\quad$ Option II (Theoretical): $\gamma^t = \lambda^t \sqrt{f(x^0) - \widetilde{f}}$
11: $\quad x^{t+1} = x^t - \gamma^t \text{sign}(g_{\xi^t}^t)$
12: **end for**

---

In the practical version of the algorithm, we use the stochastic gradient at the previous point with the current stochastic realization to update $d^t$. We use the same stochastic samples, similar to the update of the smoothness constant approximation, to reduce noise from the stochastic gradients.

We now proceed to the convergence analysis.

**Lemma F.5** (Descent lemma). *For Algorithm 2 under Assumptions 3.8, 3.2, 3.3, 3.7, the following estimate is valid:*

$$
\sum_{t=0}^{T-1} \mathbb{E} \left[ \frac{\gamma^t}{\sum_{t=0}^{T-1} \gamma^t} \left\| \nabla f(x^t) \right\|_1 \right] \leqslant \Delta^* \mathbb{E} \left[ \frac{1}{\sum_{t=0}^{T-1} \gamma^t} \right] + 2 \sum_{t=0}^{T-1} \mathbb{E} \left[ \frac{\gamma^t \left\| \nabla f(x^t) - g_{\xi^t}^t \right\|_1}{\sum_{t=0}^{T-1} \gamma^t} \right]
$$

$$
+ \sum_{t=0}^{T-1} \mathbb{E} \left[ \frac{\gamma^t \left\| \nabla f(x^{t+1}) - g_{\xi^{t+1}}^{t+1} \right\|_1}{\sum_{t=0}^{T-1} \gamma^t} \right]
$$

$$
+ \sum_{t=0}^{T-1} \mathbb{E} \left[ \frac{\gamma^t \left\| \nabla f(x^t) - g_{\xi^{t+1}}^t \right\|_1}{\sum_{t=0}^{T-1} \gamma^t} \right] + \mathbb{E} \left[ \frac{\sum_{t=0}^{T-1} (\gamma^t)^2 L_\infty^{t;\xi^{t+1}}}{\sum_{t=0}^{T-1} \gamma^t} \right],
$$

*where $L_\infty^{t,\xi^t} = \frac{\left\| g_{\xi^t}^{t+1} - g_{\xi^t}^t \right\|_1}{\left\| x^{t+1} - x^t \right\|_\infty}$.*

*Proof.*

$$
f(x^{t+1}) \leqslant f(x^t) + \langle \nabla f(x^{t+1}), x^{t+1} - x^t \rangle = f(x^t) - \gamma^t \langle \nabla f(x^{t+1}), \text{sign}(g_{\xi^t}^t) \rangle
$$

$$
= f(x^t) - \gamma^t \left\| g_{\xi^t}^t \right\|_1 - \gamma^t \langle \nabla f(x^{t+1}) - g_{\xi^t}^t, \text{sign}(g_{\xi^t}^t) \rangle
$$

$$
\overset{Conj}{\leqslant} f(x^t) - \gamma^t \left\| g_{\xi^t}^t \right\|_1 + \gamma^t \left\| \nabla f(x^{t+1}) - g_{\xi^t}^t \right\|_1 \left\| \text{sign}(g_{\xi^t}^t) \right\|_\infty
$$

$$
\overset{CS}{\leqslant} f(x^t) - \gamma^t \left\| \nabla f(x^t) \right\|_1 + 2\gamma^t \left\| \nabla f(x^t) - g_{\xi^t}^t \right\|_1
$$

$$
+ \gamma^t \left\| \nabla f(x^{t+1}) - \nabla f(x^t) \right\|_1 \left\| \text{sign}(g_{\xi^t}^t) \right\|_\infty
$$

$$
\overset{CS}{\leqslant} f(x^t) - \gamma^t \left\| \nabla f(x^t) \right\|_1 + 2\gamma^t \left\| \nabla f(x^t) - g_{\xi^t}^t \right\|_1 + \gamma^t \left\| \nabla f(x^{t+1}) - g_{\xi^{t+1}}^{t+1} \right\|_1
$$

$$
+ \gamma^t \left\| \nabla f(x^t) - g_{\xi^{t+1}}^t \right\|_1 + \gamma^t \left\| g_{\xi^{t+1}}^{t+1} - g_{\xi^{t+1}}^t \right\|_1 \left\| \text{sign}(g_{\xi^t}^t) \right\|_\infty
$$

$$\overset{(i)}{=} \quad f(x^t) - \gamma^t \left\| \nabla f(x^t) \right\|_1 + 2\gamma^t \left\| \nabla f(x^t) - g_{\xi^t}^t \right\|_1 + \gamma^t \left\| \nabla f(x^{t+1}) - g_{\xi^{t+1}}^{t+1} \right\|_1$$

$$+ \gamma^t \left\| \nabla f(x^t) - g_{\xi^{t+1}}^t \right\|_1 + \gamma^t \frac{\left\| g_{\xi^{t+1}}^{t+1} - g_{\xi^{t+1}}^t \right\|_1}{\left\| x^{t+1} - x^t \right\|_\infty} \left\| x^{t+1} - x^t \right\|_\infty$$

$$= \quad f(x^t) - \gamma^t \left\| \nabla f(x^t) \right\|_1 + 2\gamma^t \left\| \nabla f(x^t) - g_{\xi^t}^t \right\|_1 + \gamma^t \left\| \nabla f(x^{t+1}) - g_{\xi^{t+1}}^{t+1} \right\|_1$$

$$+ \gamma^t \left\| \nabla f(x^t) - g_{\xi^{t+1}}^t \right\|_1 + (\gamma^t)^2 \frac{\left\| g_{\xi^{t+1}}^{t+1} - g_{\xi^{t+1}}^t \right\|_1}{\left\| x^{t+1} - x^t \right\|_\infty},$$

where in (*i*) we assume $\left\| x^{t+1} - x^t \right\|_\infty \neq 0$. Indeed, $\left\| x^{t+1} - x^t \right\|_\infty = 0$ follows from the equality sign $\left( g_{\xi^t}^t \right) = 0$, which means $\left\| \text{sign} \left( g_{\xi^t}^t \right) \right\|_\infty = 0$ and at the $t$-th iteration this term equals zero. Thus, we can omit these iterations and consider this term only when it is non-zero, without any limitations. Now we denote $L_\infty^{t,\xi^t} = \frac{\left\| g_{\xi^t}^{t+1} - g_{\xi^t}^t \right\|_1}{\left\| x^{t+1} - x^t \right\|_\infty}$. Summing over all iterations, we obtain

$$\sum_{t=0}^{T-1} \gamma^t \left\| \nabla f(x^t) \right\|_1 \leqslant \sum_{t=0}^{T-1} f(x^t) - f(x^{t+1}) + 2\sum_{t=0}^{T-1} \gamma^t \left\| \nabla f(x^t) - g_{\xi^t}^t \right\|_1$$

$$+ \sum_{t=0}^{T-1} \gamma^t \left\| \nabla f(x^{t+1}) - g_{\xi^{t+1}}^{t+1} \right\|_1 + \sum_{t=0}^{T-1} \gamma^t \left\| \nabla f(x^t) - g_{\xi^{t+1}}^t \right\|_1$$

$$+ \sum_{t=0}^{T-1} (\gamma^t)^2 L_\infty^{t,\xi^{t+1}}$$

$$= \quad f(x^0) - f(x^T) + 2\sum_{t=0}^{T-1} \gamma^t \left\| \nabla f(x^t) - g_{\xi^t}^t \right\|_1$$

$$+ \sum_{t=0}^{T-1} \gamma^t \left\| \nabla f(x^{t+1}) - g_{\xi^{t+1}}^{t+1} \right\|_1 + \sum_{t=0}^{T-1} \gamma^t \left\| \nabla f(x^t) - g_{\xi^{t+1}}^t \right\|_1$$

$$+ \sum_{t=0}^{T-1} (\gamma^t)^2 L_\infty^{t,\xi^{t+1}}$$

$$\leqslant \quad \Delta^* + 2\sum_{t=0}^{T-1} \gamma^t \left\| \nabla f(x^t) - g_{\xi^t}^t \right\|_1 + \sum_{t=0}^{T-1} \gamma^t \left\| \nabla f(x^{t+1}) - g_{\xi^{t+1}}^{t+1} \right\|_1$$

$$+ \sum_{t=0}^{T-1} \gamma^t \left\| \nabla f(x^t) - g_{\xi^{t+1}}^t \right\|_1 + \sum_{t=0}^{T-1} (\gamma^t)^2 L_\infty^{t,\xi^{t+1}}.$$

We divide both sides of inequality on $\sum_{t=0}^{T-1} \gamma^t$.

$$\sum_{t=0}^{T-1} \frac{\gamma^t}{\sum_{t=0}^{T-1} \gamma^t} \left\| \nabla f(x^t) \right\|_1 \leqslant \frac{\Delta^*}{\sum_{t=0}^{T-1} \gamma^t} + 2\sum_{t=0}^{T-1} \frac{\gamma^t \left\| \nabla f(x^t) - g_{\xi^t}^t \right\|_1}{\sum_{t=0}^{T-1} \gamma^t}$$

$$+ \sum_{t=0}^{T-1} \frac{\gamma^t \left\| \nabla f(x^{t+1}) - g_{\xi^{t+1}}^{t+1} \right\|_1}{\sum_{t=0}^{T-1} \gamma^t} + \sum_{t=0}^{T-1} \frac{\gamma^t \left\| \nabla f(x^t) - g_{\xi^{t+1}}^t \right\|_1}{\sum_{t=0}^{T-1} \gamma^t}$$

$$+ \sum_{t=0}^{T-1} \frac{(\gamma^t)^2 L_\infty^{t,\xi^{t+1}}}{\sum_{t=0}^{T-1} \gamma^t}.$$

Taking expectation, we obtain the final result of the lemma:

$$
\begin{aligned}
\sum_{t=0}^{T-1} \mathbb{E}\left[\frac{\gamma^t}{\sum_{t=0}^{T-1}\gamma^t}\left\|\nabla f(x^t)\right\|_1\right] \;\leqslant\; & \mathbb{E}\left[\frac{\Delta^*}{\sum_{t=0}^{T-1}\gamma^t}\right] + 2\sum_{t=0}^{T-1}\mathbb{E}\left[\frac{\gamma^t\left\|\nabla f(x^t)-g_{\xi^t}^t\right\|_1}{\sum_{t=0}^{T-1}\gamma^t}\right] \\
& + \sum_{t=0}^{T-1}\mathbb{E}\left[\frac{\gamma^t\left\|\nabla f(x^{t+1})-g_{\xi^{t+1}}^{t+1}\right\|_1}{\sum_{t=0}^{T-1}\gamma^t}\right] \\
& + \sum_{t=0}^{T-1}\mathbb{E}\left[\frac{\gamma^t\left\|\nabla f(x^t)-g_{\xi^{t+1}}^t\right\|_1}{\sum_{t=0}^{T-1}\gamma^t}\right] + \sum_{t=0}^{T-1}\mathbb{E}\left[\frac{(\gamma^t)^2 L_\infty^{t,\xi^{t+1}}}{\sum_{t=0}^{T-1}\gamma^t}\right] \\
\;=\; & \Delta^*\mathbb{E}\left[\frac{1}{\sum_{t=0}^{T-1}\gamma^t}\right] + 2\sum_{t=0}^{T-1}\mathbb{E}\left[\frac{\gamma^t\left\|\nabla f(x^t)-g_{\xi^t}^t\right\|_1}{\sum_{t=0}^{T-1}\gamma^t}\right] \\
& + \sum_{t=0}^{T-1}\mathbb{E}\left[\frac{\gamma^t\left\|\nabla f(x^{t+1})-g_{\xi^{t+1}}^{t+1}\right\|_1}{\sum_{t=0}^{T-1}\gamma^t}\right] \\
& + \sum_{t=0}^{T-1}\mathbb{E}\left[\frac{\gamma^t\left\|\nabla f(x^t)-g_{\xi^{t+1}}^t\right\|_1}{\sum_{t=0}^{T-1}\gamma^t}\right] + \mathbb{E}\left[\frac{\sum_{t=0}^{T-1}(\gamma^t)^2 L_\infty^{t,\xi^{t+1}}}{\sum_{t=0}^{T-1}\gamma^t}\right].
\end{aligned}
$$

$\square$

**Theorem F.6** (**Theorem 3.9**). *Suppose Assumptions 3.8, 3.2, 3.3, 3.7 hold. Then Algorithm 2 with Option II to reach $\varepsilon$-accuracy, where $\varepsilon \geqslant \sum_{t=0}^{T-1}\mathbb{E}\left[\frac{\gamma^t}{\sum_{t=0}^{T-1}\gamma^t}\left\|\nabla f(x^t)\right\|_1\right]$ needs*

$$
\widetilde{\mathcal{O}}\left(\frac{\Delta^*\left(L_\infty\right)^3}{\varepsilon^2}\left(\mathbb{E}\left(\frac{1}{L_\infty^{0,\xi^1}}\right)^2\right) + \|\sigma\|_1^2\, L_\infty\left(\mathbb{E}\frac{1}{\min\limits_{0\leqslant t\leqslant T-1} L_\infty^{t,\xi^{t+1}}}\right)\right) \quad iterations,
$$

*where $L_\infty^{t,\xi^{t+1}} = \frac{\left\|g_{\xi^{t+1}}^{t+1}-g_{\xi^t}^t\right\|_1}{\left\|x^{t+1}-x^t\right\|_\infty}$.*

*Proof.* Let us start with the result of Lemma F.5:

$$
\begin{aligned}
\sum_{t=0}^{T-1} \mathbb{E}\left[\frac{\gamma^t}{\sum_{t=0}^{T-1}\gamma^t}\left\|\nabla f(x^t)\right\|_1\right] \;\leqslant\; & \Delta^*\mathbb{E}\left[\frac{1}{\sum_{t=0}^{T-1}\gamma^t}\right] + 2\sum_{t=0}^{T-1}\mathbb{E}\left[\frac{\gamma^t\left\|\nabla f(x^t)-g_{\xi^t}^t\right\|_1}{\sum_{t=0}^{T-1}\gamma^t}\right] \\
& + \sum_{t=0}^{T-1}\mathbb{E}\left[\frac{\gamma^t\left\|\nabla f(x^{t+1})-g_{\xi^{t+1}}^{t+1}\right\|_1}{\sum_{t=0}^{T-1}\gamma^t}\right] \\
& + \sum_{t=0}^{T-1}\mathbb{E}\left[\frac{\gamma^t\left\|\nabla f(x^t)-g_{\xi^{t+1}}^t\right\|_1}{\sum_{t=0}^{T-1}\gamma^t}\right] + \mathbb{E}\left[\frac{\sum_{t=0}^{T-1}(\gamma^t)^2 L_\infty^{t,\xi^{t+1}}}{\sum_{t=0}^{T-1}\gamma^t}\right].
\end{aligned}
$$

Using equation Höl with $p = q = 2$, we rewrite it in the following form:

$$\sum_{t=0}^{T-1} \mathbb{E}\left[\frac{\gamma^t}{\sum\limits_{t=0}^{T-1} \gamma^t}\left\|\nabla f(x^t)\right\|_1\right] \leqslant \Delta^* \mathbb{E}\left[\frac{1}{\sum\limits_{t=0}^{T-1} \gamma^t}\right]$$

$$+2\sum_{t=0}^{T-1}\left(\mathbb{E}\left\|\nabla f(x^t) - g_{\xi^t}^t\right\|_1^2\right)^{\frac{1}{2}}\left(\mathbb{E}\left[\frac{\gamma^t}{\sum\limits_{t=0}^{T-1} \gamma^t}\right]^2\right)^{\frac{1}{2}}$$

$$+\sum_{t=0}^{T-1}\left(\mathbb{E}\left\|\nabla f(x^{t+1}) - g_{\xi^{t+1}}^{t+1}\right\|_1^2\right)^{\frac{1}{2}}\left(\mathbb{E}\left[\frac{\gamma^t}{\sum\limits_{t=0}^{T-1} \gamma^t}\right]^2\right)^{\frac{1}{2}}$$

$$+\sum_{t=0}^{T-1}\left(\mathbb{E}\left\|\nabla f(x^t) - g_{\xi^{t+1}}^t\right\|_1^2\right)^{\frac{1}{2}}\left(\mathbb{E}\left[\frac{\gamma^t}{\sum\limits_{t=0}^{T-1} \gamma^t}\right]^2\right)^{\frac{1}{2}}$$

$$+\left(\mathbb{E}\left[\sum_{t=0}^{T-1}(\gamma^t)^2 L_\infty^{t,\xi^{t+1}}\right]^2\right)^{\frac{1}{2}}\left(\mathbb{E}\left[\frac{1}{\sum\limits_{t=0}^{T-1} \gamma^t}\right]^2\right)^{\frac{1}{2}}. \quad (52)$$

Now we use our choice of $\gamma^t$. Let us firstly estimate the denominator that is exactly $\lambda^t = \frac{1}{\sqrt{\sum\limits_{i=0}^{t-1}\frac{\left\|g_{\xi^{i+1}}^{i+1} - g_{\xi^{i+1}}^i\right\|_1}{\left\|x^{i+1} - x^i\right\|_\infty}}} = \frac{1}{\sqrt{\sum\limits_{i=0}^{t-1} L_\infty^{i,\xi^{i+1}}}}$. Let us estimate the following term.

$$\sum_{t=0}^{T-1}(\lambda^t)^2 L_\infty^{t,\xi^{t+1}} = \sum_{t=0}^{T-1}\frac{L_\infty^{t,\xi^{t+1}}}{\sum\limits_{i=0}^{t-1} L_\infty^{i,\xi^{i+1}}}.$$

We mention, that each $L_\infty^{i,\xi^{i+1}}$ is bounded from the definition of smoothness (see Assumption 3.8), i.e., $L_\infty^{i,\xi^{i+1}} \leqslant L_\infty$. We consider the sequence $\left\{L_\infty^{i,\xi^{i+1}}\right\}_{i=0}^{T-1}$. Since each term in this sequence is bounded, there exists $r$ such that $\sum\limits_{i=0}^{r-2} L_\infty^{i,\xi^{i+1}} \leqslant L_\infty^{r-1,\xi^r}$ and for each $t \geqslant r - 1$ such that $\sum\limits_{i=0}^t L_\infty^{i,\xi^{i+1}} \geqslant L_\infty^{t+1,\xi^{t+2}}$. In that way, we divide the sum into two parts:

$$\sum_{t=0}^{T-1}\frac{L_\infty^{t,\xi^{t+1}}}{\sum\limits_{i=0}^{t-1} L_\infty^{i,\xi^{i+1}}} = \sum_{t=0}^{r-1}\frac{L_\infty^{t,\xi^{t+1}}}{\sum\limits_{i=0}^{t-1} L_\infty^{i,\xi^{i+1}}} + \sum_{t=r}^{T-1}\frac{L_\infty^{t,\xi^{t+1}}}{\sum\limits_{i=0}^{t-1} L_\infty^{i,\xi^{i+1}}}. \quad (53)$$

Considering the first sum in equation 53, we mention, that we can estimate the denominator as $\sum_{i=0}^{t-1} L_\infty^{i,\xi^{i+1}} \geqslant L_\infty^{0,\xi^1}$. As for the numerator. Thus,

$$\sum_{t=0}^{r-1}\frac{L_\infty^{t,\xi^{t+1}}}{\sum\limits_{i=0}^{t-1} L_\infty^{i,\xi^{i+1}}} \leqslant \frac{1}{L_\infty^{0,\xi^1}}\left(\sum_{t=0}^{r-2} L_\infty^{t,\xi^{t+1}} + L_\infty^{r-1,\xi^r}\right) \leqslant \frac{2L_\infty^{r-1,\xi^r}}{L_\infty^{0,\xi^1}} \leqslant \frac{2L_\infty}{L_\infty^{0,\xi^1}}. \quad (54)$$

Considering the second sum in equation 53, we have

$$\sum_{t=r}^{T-1} \frac{L_\infty^{t,\xi^{t+1}}}{\sum_{i=0}^{t-1} L_\infty^{i,\xi^{i+1}}} = \sum_{t=r}^{T-1} \frac{L_\infty^{t,\xi^{t+1}}}{\frac{1}{2}\sum_{i=0}^{t-1} L_\infty^{i,\xi^{i+1}} + \frac{1}{2}\sum_{i=0}^{t-1} L_\infty^{i,\xi^{i+1}}}.$$

Estimating any of the sums in the denominator, we claim, that $\sum_{i=0}^{t-1} L_\infty^{i,\xi^{i+1}} \geqslant L_\infty^{t,\xi^{t+1}}$, since $t-1 \geqslant r-1$. In that way,

$$\sum_{t=r}^{T-1} \frac{L_\infty^{t,\xi^{t+1}}}{\sum_{i=0}^{t-1} L_\infty^{i,\xi^{i+1}}} \leqslant \sum_{t=r}^{T-1} \frac{2L_\infty^{t,\xi^{t+1}}}{\sum_{i=0}^{t} L_\infty^{i,\xi^{i+1}}} \leqslant 2\sum_{t=0}^{T-1} \frac{L_\infty^{t,\xi^{t+1}}}{\sum_{i=0}^{t} L_\infty^{i,\xi^{i+1}}}. \tag{55}$$

Next we denote $s^t = \sum_{i=0}^{t} L_\infty^{t,\xi^{t+1}}$ and have

$$L_\infty^{t,\xi^{t+1}} \frac{1}{\sum_{i=0}^{t} L_\infty^{i,\xi^{i+1}}} = (s^t - s^{t-1}) \frac{1}{\sum_{i=0}^{t} L_\infty^{i,\xi^{i+1}}} = \int_{s^{t-1}}^{s^t} \frac{1}{\sum_{i=0}^{t} L_\infty^{i,\xi^{i+1}}} dx \overset{(i)}{\leqslant} \int_{s^{t-1}}^{s^t} \frac{1}{x} dx, \tag{56}$$

where $(i)$ was done due to $\frac{1}{x}$ is a non-increasing function on $(0, +\infty)$. Summing over $t$, we obtain

$$2\sum_{t=1}^{T} \frac{L_\infty^{t,\xi^{t+1}}}{\sum_{i=0}^{t} L_\infty^{i,\xi^{i+1}}} \leqslant 2\int_{s^0}^{s^T} \frac{1}{x} dx = 2\log(s^T) - 2\log(s^0) = 2\log\left(\frac{\sum_{t=1}^{T} L_\infty^{t,\xi^{t+1}}}{L_\infty^{0,\xi^1}}\right) \leqslant 2\log\left(\frac{L_\infty T}{L_\infty^{0,\xi^1}}\right).$$

Combining this estimate with equation 55,

$$\sum_{t=r}^{T-1} \frac{L_\infty^{t,\xi^{t+1}}}{\sum_{i=0}^{t-1} L_\infty^{i,\xi^{i+1}}} \leqslant 2\sum_{t=1}^{T} \frac{L_\infty^{t,\xi^{t+1}}}{\sum_{i=0}^{t} L_\infty^{i,\xi^{i+1}}} + 2 \leqslant 2\left(\log\left(\frac{L_\infty T}{L_\infty^{0,\xi^1}}\right) + 1\right) \leqslant 4\log\left(\frac{L_\infty T}{L_\infty^{0,\xi^1}}\right). \tag{57}$$

Substituting equation 54 and equation 57 into equation 53, we obtain

$$\sum_{t=0}^{T-1} (\lambda^t)^2 L_\infty^{t,\xi^{t+1}} \leqslant 2\frac{L_\infty}{L_\infty^{0,\xi^1}} + 4\log\left(\frac{L_\infty T}{L_\infty^{0,\xi^1}}\right). \tag{58}$$

We additionally note, that if $r > T-1$, only first term remains in this estimate, consequently our bound equation 58 is correct. Next, we estimate

$$\frac{1}{\sum_{t=0}^{T-1} \lambda^t} = \frac{1}{\sum_{t=0}^{T-1} \frac{1}{\sqrt{L_\infty + \sum_{i=0}^{t-1} L_\infty^{i,\xi^{i+1}}}}} \leqslant \frac{\sqrt{L_\infty}}{\sum_{t=0}^{T-1} \frac{1}{\sqrt{t+1}}} \leqslant \frac{\sqrt{L_\infty}}{\sqrt{T}}. \tag{59}$$

Now we estimate the second, third and forth terms in equation 52. In the same manner, as in equation 39, we can estimate

$$\mathbb{E}\left\|\nabla f(x^t) - g_{\xi^t}^t\right\|_1^2 \leqslant \|\sigma\|_1^2,$$

$$\mathbb{E}\left\|\nabla f(x^{t+1}) - g_{\xi^{t+1}}^{t+1}\right\|_1^2 \leqslant \|\sigma\|_1^2, \tag{60}$$

$$\mathbb{E}\left\|\nabla f(x^t) - g_{\xi^{t+1}}^t\right\|_1^2 \leqslant \|\sigma\|_1^2,$$

where the last inequality is correct due to the fact that that stochastic realization $\xi^{t+1}$ is independent from the point $x^t$. Thus, using equation 59,

$$\sum_{t=0}^{T-1}\left(\mathbb{E}\left\|\nabla f(x^t) - g_{\xi^t}^t\right\|_1^2\right)^{\frac{1}{2}} \cdot \left(\mathbb{E}\left[\frac{\gamma^t}{\sum_{t=0}^{T-1}\gamma^t}\right]^2\right)^{\frac{1}{2}}$$

$$\leqslant \frac{\sqrt{L_\infty}\|\sigma\|_1}{\sqrt{T}}\sum_{t=0}^{T-1}\left(\mathbb{E}\frac{1}{\sum_{i=0}^{t-1}L_\infty^{i,\xi^{i+1}}}\right)^{\frac{1}{2}}$$

$$\leqslant \frac{\sqrt{L_\infty}\|\sigma\|_1}{\sqrt{T}}\left(\mathbb{E}\frac{1}{\min_{0\leqslant t\leqslant T-1}L_\infty^{t,\xi^{t+1}}}\right)^{\frac{1}{2}}\sum_{t=0}^{T-1}\frac{1}{\sqrt{t+1}}$$

$$\leqslant 2\sqrt{L_\infty}\|\sigma\|_1\left(\mathbb{E}\frac{1}{\min_{0\leqslant t\leqslant T-1}L_\infty^{t,\xi^{t+1}}}\right)^{\frac{1}{2}}.$$

It is clear that we can bound the rest two terms in the same manner. Now, substituting this estimate along with equation 58 and equation 59 into equation 52, we obtain

$$\sum_{t=0}^{T-1}\mathbb{E}\left[\frac{\gamma^t}{\sum_{t=0}^{T-1}\gamma^t}\left\|\nabla f(x^t)\right\|_1\right] \leqslant \frac{\Delta^*\sqrt{L_\infty}}{\sqrt{(f(x^0) - \widetilde{f})T}}$$

$$+ 8\sqrt{L_\infty}\|\sigma\|_1\left(\mathbb{E}\frac{1}{\min_{0\leqslant t\leqslant T-1}L_\infty^{t,\xi^{t+1}}}\right)^{\frac{1}{2}}$$

$$+ 8\frac{(f(x^0) - \widetilde{f})\sqrt{L_\infty}}{\sqrt{(f(x^0) - \widetilde{f})T}}\left(\mathbb{E}\log^2\left(\frac{L_\infty T}{L_\infty^{0,\xi^1}}\right)\right)^{\frac{1}{2}}$$

$$+ 4\frac{(f(x^0) - \widetilde{f})\sqrt{L_\infty}}{\sqrt{(f(x^0) - \widetilde{f})T}}\left(\mathbb{E}\left(\frac{L_\infty}{L_\infty^{0,\xi^1}}\right)^2\right)^{\frac{1}{2}}. \tag{61}$$

Now we use $\Delta^* \leqslant f(x^0) - \widetilde{f}$ to obtain the final estimate:

$$\sum_{t=0}^{T-1}\mathbb{E}\left[\frac{\gamma^t}{\sum_{t=0}^{T-1}\gamma^t}\left\|\nabla f(x^t)\right\|_1\right] \leqslant 13\frac{\sqrt{(f(x^0) - \widetilde{f})}\,(L_\infty)^{\frac{3}{2}}}{T}\left(\mathbb{E}\left(\frac{1}{L_\infty^{0,\xi^1}}\right)^2\right)^{\frac{1}{2}}$$

$$\cdot \left(\mathbb{E}\log^2\left(\frac{L_\infty T}{L_\infty^{0,\xi^1}}\right)\right)^{\frac{1}{2}}$$

$$+ 8\|\sigma\|_1\left(\sqrt{L_\infty}\left(\mathbb{E}\frac{1}{\min_{0\leqslant t\leqslant T-1}L_\infty^{t,\xi^{t+1}}}\right)^{\frac{1}{2}}\right).$$

Expressing the number of iterations and using $\varepsilon \geqslant \sum_{t=0}^{T-1} \mathbb{E}\left[\frac{\gamma^t}{\sum_{t=0}^{T-1} \gamma^t} \|\nabla f(x^t)\|_1\right]$ as a criterion, we

obtain that the algorithm needs $\widetilde{\mathcal{O}}\left(\frac{\Delta^*(L_\infty)^3}{\varepsilon^2}\left(\mathbb{E}\left(\frac{1}{L_\infty^{0,\xi^1}}\right)^2\right) + \|\sigma\|_1^2 L_\infty \left(\mathbb{E}\frac{1}{\min\limits_{0\leqslant t\leqslant T-1} L_\infty^{t,\xi^{t+1}}}\right)\right)$ it-

erations to reach $\varepsilon$-accuracy. $\qquad\square$

*Remark* F.7 (Remark 3.10). Under conditions of Theorem 3.9 Algorithm 2 with $\lambda^t = \frac{1}{\sqrt{L_\infty + \sum_{i=0}^{t-1} \frac{\left\|g_{\xi^{i+1}}^{i+1} - g_{\xi^i}^i\right\|_1}{\|x^{i+1}-x^i\|_\infty}}}$, Option II and mini-batch of the size $t+1$ at $t$-th iteration to reach $\varepsilon$-accuracy

needs

$$\widetilde{\mathcal{O}}\left(\frac{\Delta^* L_\infty}{\varepsilon^2} + \frac{\|\sigma\|_1^2 L_\infty}{\varepsilon^2}\left(\mathbb{E}\frac{1}{\min\limits_{0\leqslant t\leqslant T-1} L_\infty^{t,\xi^{t+1}}}\right)\right) \quad \text{iterations,}$$

where $\varepsilon \geqslant \frac{1}{T}\sum_{t=0}^{T-1}\|\nabla f(x^t)\|_1$, $L_\infty^{t,\xi^{t+1}} = \frac{\left\|g_{\xi^{t+1}}^{t+1} - g_{\xi^t}^t\right\|_1}{\|x^{t+1}-x^t\|_\infty}$.

*Proof.* The proof of the remark repeats the proof of Theorem 3.9 except for the estimate on $\sum_{t=0}^{T-1}(\lambda^t)^2 L_\infty^{t,\xi^{t+1}}$ term and $\mathbb{E}\left\|\nabla f(x^t) - g_{\xi^t}^t\right\|_1^2$ term. Let us derive them. We use definition $L_\infty^{t,\xi^{t+1}} = \frac{\left\|g_{\xi^{t+1}}^{t+1} - g_{\xi^t}^t\right\|_1}{\|x^{t+1}-x^t\|_\infty}$.

$$\sum_{t=0}^{T-1}(\lambda^t)^2 L_\infty^{t,\xi^{t+1}} = \sum_{t=0}^{T-1}\frac{L_\infty^{t,\xi^{t+1}}}{L_\infty + \sum\limits_{i=0}^{t-1} L_\infty^{i,\xi^{i+1}}} \leqslant \sum_{t=0}^{T-1}\frac{L_\infty^{t,\xi^{t+1}}}{\sum\limits_{i=0}^{t} L_\infty^{i,\xi^{i+1}}}.$$

Continuing analogically to equation 56 - equation 57, we get

$$\sum_{t=0}^{T-1}(\lambda^t)^2 L_\infty^t \leqslant 2\log\left(\frac{L_\infty T}{L_\infty^{0,\xi^1}}\right).$$

We substitute this bound into equation 61 instead of equation 58. Next, since we now use mini-batches, we can bound

$$\mathbb{E}\left\|\nabla f(x^t) - g_{\xi^t}^t\right\|_1^2 \leqslant \frac{\|\sigma\|_1^2}{t+1},$$

$$\mathbb{E}\left\|\nabla f(x^{t+1}) - g_{\xi^{t+1}}^{t+1}\right\|_1^2 \leqslant \frac{\|\sigma\|_1^2}{t+2},$$

$$\mathbb{E}\left\|\nabla f(x^t) - g_{\xi^{t+1}}^t\right\|_1^2 \leqslant \frac{\|\sigma\|_1^2}{t+1},$$

instead of equation 60. In that way,

$$\sum_{t=0}^{T-1}\left(\mathbb{E}\left\|\nabla f(x^t) - g_{\xi^t}^t\right\|_1^2\right)^{\frac{1}{2}} \cdot \left(\mathbb{E}\left[\frac{\gamma^t}{\sum_{t=0}^{T-1}\gamma^t}\right]^2\right)^{\frac{1}{2}}$$

$$\leqslant \frac{\sqrt{L_\infty}\|\sigma\|_1}{\sqrt{T}} \sum_{t=0}^{T-1} \frac{1}{\sqrt{t+1}} \left( \mathbb{E} \frac{1}{\sum_{i=0}^{t-1} L_\infty^{i,\xi^{i+1}}} \right)^{\frac{1}{2}}$$

$$\leqslant \frac{\sqrt{L_\infty}\|\sigma\|_1}{\sqrt{T}} \left( \mathbb{E} \frac{1}{\min_{0\leqslant t\leqslant T-1} L_\infty^{t,\xi^{t+1}}} \right)^{\frac{1}{2}} \sum_{t=0}^{T-1} \frac{1}{t+1}$$

$$\leqslant 2\frac{\sqrt{L_\infty}\|\sigma\|_1}{\sqrt{T}} \left( \mathbb{E} \frac{1}{\min_{0\leqslant t\leqslant T-1} L_\infty^{t,\xi^{t+1}}} \right)^{\frac{1}{2}} \log(T),$$

which ends the proof of the remark.

$\square$

### F.3 DISTRIBUTED SETTING

We remind, that in distributed setting we consider Assumption E.6. We present the theoretical result with the following approximation of $L_\infty$ in Algorithm 2:

$$\lambda^t = \frac{1}{\sqrt{\sum_{i=0}^{t-1} \frac{1}{M} \sum_{j=1}^{M} \frac{\left\|g_{j,\xi^{i+1}}^{i+1} - g_{j,\xi^{i+1}}^{i}\right\|_1}{\|x^{i+1}-x^i\|_\infty}}}.$$

In this section, we denote $g_{j,\xi^t}^t$ the stochastic gradient from the $j$-th device, computed at the $t$-th iteration, according to the stochastic realization $\xi^t$.

**Lemma F.8** (Descent lemma). *For Algorithm 2 under Assumptions 3.8, 3.2, 3.3, E.6, the following estimate is valid:*

$$\sum_{t=0}^{T-1} \mathbb{E}\left[\gamma^t \left\|\nabla f(x^t)\right\|_1\right] \leqslant \Delta^* \mathbb{E}\left[\frac{1}{\sum_{t=0}^{T-1} \gamma^t}\right] + 2\sum_{t=0}^{T-1} \mathbb{E}\left[\frac{\gamma^t \widetilde{\delta}^t}{\sum_{t=0}^{T-1} \gamma^t}\right]$$

$$+ \sum_{t=0}^{T-1} \mathbb{E}\left[\frac{\gamma^t \frac{1}{M}\sum_{j=1}^{M}\|\nabla f(x^t) - g_{j,\xi^{t+1}}^t\|_1}{\sum_{t=0}^{T-1} \gamma^t}\right]$$

$$+ \sum_{t=0}^{T-1} \mathbb{E}\left[\frac{\gamma^t \frac{1}{M}\sum_{j=1}^{M}\|\nabla f(x^{t+1}) - g_{j,\xi^{t+1}}^{t+1}\|_1}{\sum_{t=0}^{T-1} \gamma^t}\right]$$

$$+ \mathbb{E}\left[\frac{\sum_{t=0}^{T-1}(\gamma^t)^2 L_\infty^{t,\xi^{t+1}}}{\sum_{t=0}^{T-1} \gamma^t}\right],$$

*where* $\widetilde{\delta}^t = \sum_{i=1}^{d} |[\nabla f(x^t)]_i| \, \mathbb{I}\left(\text{sign}\left(\sum_{j=1}^{M} \text{sign}\left(\left[g_{j,\xi^t}^t\right]_i\right)\right) \neq \text{sign}\left([\nabla f(x^t)]_i\right)\right)$

*and* $L_\infty^{t,\xi^t} = \frac{1}{M}\sum_{j=1}^{M} \frac{\left\|g_{j,\xi^t}^{t+1} - g_{j,\xi^t}^t\right\|_1}{\|x^{t+1}-x^t\|_\infty}.$

*Proof.*

$$
\begin{aligned}
f(x^{t+1}) - f(x^t) \quad \leqslant \quad & \langle \nabla f(x^{t+1}), x^{t+1} - x^t \rangle \\[4pt]
= \quad & -\gamma^t \left\langle \nabla f(x^{t+1}), \operatorname{sign}\left( \sum_{j=1}^{M} \operatorname{sign}\left(g_{j,\xi^t}^t\right) \right) \right\rangle \\[4pt]
= \quad & -\gamma^t \left\langle \nabla f(x^t), \operatorname{sign}\left( \sum_{j=1}^{M} \operatorname{sign}\left(g_{j,\xi^t}^t\right) \right) \right\rangle \\[4pt]
& -\gamma^t \left\langle \nabla f(x^{t+1}) - \nabla f(x^t), \operatorname{sign}\left( \sum_{j=1}^{M} \operatorname{sign}\left(g_{j,\xi^t}^t\right) \right) \right\rangle \\[4pt]
= \quad & -\gamma^t \|\nabla f(x^t)\|_1 + 2\gamma^t \sum_{i=1}^{d} \left|\left[\nabla f(x^t)\right]_i\right| \\[4pt]
& \cdot \mathbb{I}\left( \operatorname{sign}\left( \sum_{j=1}^{M} \operatorname{sign}\left(\left[g_{j,\xi^t}^t\right]_i\right) \right) \neq \operatorname{sign}\left(\left[\nabla f(x^t)\right]_i\right) \right) \\[4pt]
& -\gamma^t \left\langle \nabla f(x^{t+1}) - \nabla f(x^t), \operatorname{sign}\left( \sum_{j=1}^{M} \operatorname{sign}\left(g_{j,\xi^t}^t\right) \right) \right\rangle \\[4pt]
\overset{Conj,(i)}{\leqslant} \quad & -\gamma^t \|\nabla f(x^t)\|_1 + 2\gamma^t \widetilde{\delta}^t \\[4pt]
& + \gamma^t \|\nabla f(x^{t+1}) - \nabla f(x^t)\|_1 \left\| \operatorname{sign}\left( \sum_{j=1}^{M} \operatorname{sign}\left(g_{j,\xi^t}^t\right) \right) \right\|_\infty \\[4pt]
= \quad & -\gamma^t \|\nabla f(x^t)\|_1 + 2\gamma^t \widetilde{\delta}^t \\[4pt]
& + \gamma^t \frac{1}{M} \sum_{j=1}^{M} \|\nabla f(x^{t+1}) - \nabla f(x^t)\|_1 \left\| \operatorname{sign}\left( \sum_{j=1}^{M} \operatorname{sign}\left(g_{j,\xi^t}^t\right) \right) \right\|_\infty \\[4pt]
\overset{CS}{\leqslant} \quad & -\gamma^t \|\nabla f(x^t)\|_1 + 2\gamma^t \widetilde{\delta}^t + \gamma^t \frac{1}{M} \sum_{j=1}^{M} \|\nabla f(x^t) - g_{j,\xi^{t+1}}^t\|_1 \\[4pt]
& + \gamma^t \frac{1}{M} \sum_{j=1}^{M} \|\nabla f(x^{t+1}) - g_{j,\xi^{t+1}}^{t+1}\|_1 \\[4pt]
& + \gamma^t \frac{1}{M} \sum_{j=1}^{M} \|g_{j,\xi^{t+1}}^{t+1} - g_{j,\xi^{t+1}}^t\|_1 \left\| \operatorname{sign}\left( \sum_{j=1}^{M} \operatorname{sign}\left(g_{j,\xi^t}^t\right) \right) \right\|_\infty \\[4pt]
\overset{(ii)}{=} \quad & -\gamma^t \|\nabla f(x^t)\|_1 + 2\gamma^t \widetilde{\delta}^t + \gamma^t \frac{1}{M} \sum_{j=1}^{M} \|\nabla f(x^t) - g_{j,\xi^{t+1}}^t\|_1 \\[4pt]
& + \gamma^t \frac{1}{M} \sum_{j=1}^{M} \|\nabla f(x^{t+1}) - g_{j,\xi^{t+1}}^{t+1}\|_1 \\[4pt]
& + \gamma^t \frac{1}{M} \sum_{j=1}^{M} \frac{\|g_{j,\xi^{t+1}}^{t+1} - g_{j,\xi^{t+1}}^t\|_1}{\|x^{t+1} - x^t\|_\infty} \|x^{t+1} - x^t\|_\infty \\[4pt]
= \quad & -\gamma^t \|\nabla f(x^t)\|_1 + 2\gamma^t \widetilde{\delta}^t + \gamma^t \frac{1}{M} \sum_{j=1}^{M} \|\nabla f(x^t) - g_{j,\xi^{t+1}}^t\|_1
\end{aligned}
$$

$$+\gamma^t \frac{1}{M} \sum_{j=1}^{M} \|\nabla f(x^{t+1}) - g_{j,\xi^{t+1}}^{t+1}\|_1$$

$$+(\gamma^t)^2 \frac{1}{M} \sum_{j=1}^{M} \frac{\|g_{j,\xi^{t+1}}^{t+1} - g_{j,\xi^{t+1}}^{t}\|_1}{\|x^{t+1} - x^t\|_\infty},$$

where in (*i*) we denote $\widetilde{\delta}^t = \sum_{i=1}^{d} |[\nabla f(x^t)]_i| \, \mathbb{I}\left(\text{sign}\left(\sum_{j=1}^{M} \text{sign}\left(\left[g_{j,\xi^t}^t\right]_i\right)\right) \neq \text{sign}\left([\nabla f(x^t)]_i\right)\right)$ and in (*ii*) we assume $\|x^{t+1} - x^t\|_\infty \neq 0$ (analogous to Lemma F.5). Defining $L_\infty^{t,\xi^{t+1}} = \frac{1}{M} \sum_{j=1}^{M} \frac{\|g_{j,\xi^{t+1}}^{t+1} - g_{j,\xi^{t+1}}^{t}\|_1}{\|x^{t+1} - x^t\|_\infty}$ and summing over all iterations gives us

$$\sum_{t=0}^{T-1} \gamma^t \|\nabla f(x^t)\|_1 \leqslant \Delta^* + 2\sum_{t=0}^{T-1} \gamma^t \widetilde{\delta}^t + \sum_{t=0}^{T-1} \gamma^t \frac{1}{M} \sum_{j=1}^{M} \|\nabla f(x^t) - g_{j,\xi^{t+1}}^t\|_1$$

$$+ \sum_{t=0}^{T-1} \gamma^t \frac{1}{M} \sum_{j=1}^{M} \|\nabla f(x^{t+1}) - g_{j,\xi^{t+1}}^{t+1}\|_1 + \sum_{t=0}^{T-1} (\gamma^t)^2 L_\infty^{t,\xi^t},$$

$$\sum_{t=0}^{T-1} \frac{\gamma^t}{\sum_{t=0}^{T-1} \gamma^t} \|\nabla f(x^t)\|_1 \leqslant \frac{\Delta^*}{\sum_{t=0}^{T-1} \gamma^t} + 2\sum_{t=0}^{T-1} \frac{\gamma^t \widetilde{\delta}^t}{\sum_{t=0}^{T-1} \gamma^t} + \sum_{t=0}^{T-1} \frac{\gamma^t \frac{1}{M} \sum_{j=1}^{M} \|\nabla f(x^t) - g_{j,\xi^{t+1}}^t\|_1}{\sum_{t=0}^{T-1} \gamma^t}$$

$$+ \sum_{t=0}^{T-1} \frac{\gamma^t \frac{1}{M} \sum_{j=1}^{M} \|\nabla f(x^{t+1}) - g_{j,\xi^{t+1}}^{t+1}\|_1}{\sum_{t=0}^{T-1} \gamma^t} + \sum_{t=0}^{T-1} \frac{(\gamma^t)^2 L_\infty^{t,\xi^{t+1}}}{\sum_{t=0}^{T-1} \gamma^t}.$$

Taking expectation, we derive the result of the lemma:

$$\sum_{t=0}^{T-1} \mathbb{E}\left[\gamma^t \|\nabla f(x^t)\|_1\right] \leqslant \Delta^* \mathbb{E}\left[\frac{1}{\sum_{t=0}^{T-1} \gamma^t}\right] + 2\sum_{t=0}^{T-1} \mathbb{E}\left[\frac{\gamma^t \widetilde{\delta}^t}{\sum_{t=0}^{T-1} \gamma^t}\right]$$

$$+ \sum_{t=0}^{T-1} \mathbb{E}\left[\frac{\gamma^t \frac{1}{M} \sum_{j=1}^{M} \|\nabla f(x^t) - g_{j,\xi^{t+1}}^t\|_1}{\sum_{t=0}^{T-1} \gamma^t}\right]$$

$$+ \sum_{t=0}^{T-1} \mathbb{E}\left[\frac{\gamma^t \frac{1}{M} \sum_{j=1}^{M} \|\nabla f(x^{t+1}) - g_{j,\xi^{t+1}}^{t+1}\|_1}{\sum_{t=0}^{T-1} \gamma^t}\right]$$

$$+ \mathbb{E}\left[\frac{\sum_{t=0}^{T-1} (\gamma^t)^2 L_\infty^{t,\xi^{t+1}}}{\sum_{t=0}^{T-1} \gamma^t}\right].$$

$\square$

**Theorem F.9.** *Suppose Assumptions 3.8, 3.2, 3.3, E.6 hold. Then Algorithm 2 with Option II to reach ε-accuracy, where* $\varepsilon \geqslant \sum_{t=0}^{T-1} \mathbb{E}\left[\frac{\gamma^t}{\sum_{t=0}^{T-1} \gamma^t} \|\nabla f(x^t)\|_1\right]$ *needs*

$$\widetilde{\mathcal{O}}\left(\frac{\Delta^* (L_\infty)^3}{\varepsilon^2}\left(\mathbb{E}\left(\frac{1}{L_\infty^{0,\xi^1}}\right)^2\right) + \|\sigma\|_1^2 L_\infty \left(\mathbb{E}\frac{1}{\min\limits_{0\leqslant t\leqslant T-1} L_\infty^{t,\xi^{t+1}}}\right)\right) \quad \text{iterations,}$$

*where* $L_\infty^{t,\xi^{t+1}} = \frac{1}{M}\sum_{j=1}^{M}\frac{\left\|g_{j,\xi^{t+1}}^{t+1} - g_{j,\xi^{t+1}}^{t}\right\|_1}{\|x^{t+1}-x^t\|_\infty}$.

*Proof.* Let us start with the result of Lemma F.8:

$$\sum_{t=0}^{T-1}\mathbb{E}\left[\gamma^t \|\nabla f(x^t)\|_1\right] \leqslant \Delta^*\mathbb{E}\left[\frac{1}{\sum_{t=0}^{T-1}\gamma^t}\right] + 2\sum_{t=0}^{T-1}\mathbb{E}\left[\frac{\gamma^t\widetilde{\delta}^t}{\sum_{t=0}^{T-1}\gamma^t}\right]$$

$$+ \sum_{t=0}^{T-1}\mathbb{E}\left[\frac{\gamma^t\frac{1}{M}\sum_{j=1}^{M}\|\nabla f(x^t) - g_{j,\xi^{t+1}}^{t}\|_1}{\sum_{t=0}^{T-1}\gamma^t}\right]$$

$$+ \sum_{t=0}^{T-1}\mathbb{E}\left[\frac{\gamma^t\frac{1}{M}\sum_{j=1}^{M}\|\nabla f(x^{t+1}) - g_{j,\xi^{t+1}}^{t+1}\|_1}{\sum_{t=0}^{T-1}\gamma^t}\right]$$

$$+ \mathbb{E}\left[\frac{\sum_{t=0}^{T-1}(\gamma^t)^2 L_\infty^{t,\xi^{t+1}}}{\sum_{t=0}^{T-1}\gamma^t}\right].$$

Note that we have already estimated all terms in Theorem F.6 except $\sum_{t=0}^{T-1}\mathbb{E}\left[\frac{\gamma^t\widetilde{\delta}^t}{\sum_{t=0}^{T-1}\gamma^t}\right]$. However, using Lemma E.7 together with equation Höl, we can do the same thing and obtain

$$\sum_{t=0}^{T-1}\mathbb{E}\left[\frac{\gamma^t\widetilde{\delta}^t}{\sum_{t=0}^{T-1}\gamma^t}\right] \leqslant \sum_{t=0}^{T-1}\left(\mathbb{E}\left[\widetilde{\delta}\right]^2\right)^{\frac{1}{2}}\left(\mathbb{E}\left[\frac{\gamma^t}{\sum_{t=0}^{T-1}\gamma^t}\right]^2\right)^{\frac{1}{2}}$$

$$\leqslant 2\sqrt{L_\infty}\|\sigma\|_1\left(\mathbb{E}\frac{1}{\min\limits_{0\leqslant t\leqslant T-1} L_\infty^{t,\xi^{t+1}}}\right)^{\frac{1}{2}}.$$

In that way, we get the same estimate as in Theorem F.6:

$$\sum_{t=0}^{T-1}\mathbb{E}\left[\frac{\gamma^t}{\sum_{t=0}^{T-1}\gamma^t}\|\nabla f(x^t)\|_1\right] \leqslant 13\frac{\sqrt{(f(x^0) - \widetilde{f})}(L_\infty)^{\frac{3}{2}}}{T}\left(\mathbb{E}\left(\frac{1}{L_\infty^{0,\xi^1}}\right)^2\right)^{\frac{1}{2}}$$

$$\cdot\left(\mathbb{E}\log^2\left(\frac{L_\infty T}{L_\infty^{0,\xi^1}}\right)\right)^{\frac{1}{2}}$$

$$+8\|\sigma\|_1\left(\sqrt{L_\infty}\left(\mathbb{E}\frac{1}{\min\limits_{0\leqslant t\leqslant T-1}L_\infty^{t,\xi^{t+1}}}\right)^{\frac{1}{2}}\right).$$

Expressing the number of iterations and using $\varepsilon\geqslant\sum\limits_{t=0}^{T-1}\mathbb{E}\left[\frac{\gamma^t}{\sum\limits_{t=0}^{T-1}\gamma^t}\left\|\nabla f(x^t)\right\|_1\right]$ as a criterion, we

obtain that the algorithm needs $\widetilde{\mathcal{O}}\left(\frac{\Delta^*(L_\infty)^3}{\varepsilon^2}\left(\mathbb{E}\left(\frac{1}{L_\infty^{0,\xi^1}}\right)^2\right)+\|\sigma\|_1^2 L_\infty\left(\mathbb{E}\frac{1}{\min\limits_{0\leqslant t\leqslant T-1}L_\infty^{t,\xi^{t+1}}}\right)\right)$ it-

erations to reach $\varepsilon$-accuracy. $\qquad\square$

*Remark* F.10. Under conditions of Theorem F.9 Algorithm 2 with $\lambda^t = \frac{1}{\sqrt{L_\infty+\sum\limits_{i=0}^{t-1}\frac{1}{M}\sum\limits_{j=1}^{M}\frac{\left\|g_{j,\xi^{i+1}}^{i+1}-g_{j,\xi^i}^i\right\|_1}{\left\|x^{i+1}-x^i\right\|_\infty}}}$, Option II and mini-batch of the size $t + 1$ at $t$-th iteration

to reach $\varepsilon$-accuracy needs

$$\widetilde{\mathcal{O}}\left(\frac{\Delta^* L_\infty}{\varepsilon^2}+\frac{\|\sigma\|_1^2 L_\infty}{\varepsilon^2}\left(\mathbb{E}\frac{1}{\min\limits_{0\leqslant t\leqslant T-1}L_\infty^{t,\xi^{t+1}}}\right)\right)\quad\text{iterations,}$$

where $\varepsilon\geqslant\frac{1}{T}\sum\limits_{t=0}^{T-1}\left\|\nabla f(x^t)\right\|_1, L_\infty^{t,\xi^{t+1}}=\frac{1}{M}\sum\limits_{j=1}^{M}\frac{\left\|g_{j,\xi^{t+1}}^{t+1}-g_{j,\xi^t}^t\right\|_1}{\left\|x^{t+1}-x^t\right\|_\infty}$.

*Proof.* Proof repeats the proof of Remark 3.10. $\qquad\square$

## F.4 MEMORY-EFFICIENT ALIAS

**Lemma F.11** (Descent lemma). *For Algorithm 2 under Assumptions 3.11, 3.2, 3.3, 3.4, the following estimate is valid:*

$$\sum_{t=0}^{T-1}\gamma^t\left\|\nabla f(x^t)\right\|_1\leqslant\Delta^*+\sum_{t=0}^{T-1}(\gamma^t)^2 d^2 L_1^t,$$

*where $L_1^t=\frac{\left\|\nabla f(x^{t+1})-\nabla f(x^t)\right\|_\infty}{\left\|x^{t+1}-x^t\right\|_1}$.*

*Proof.*

$$
\begin{aligned}
f(x^{t+1}) &\leqslant f(x^t)+\left\langle\nabla f(x^{t+1}),x^{t+1}-x^t\right\rangle=f(x^t)-\gamma^t\left\langle\nabla f(x^{t+1}),\text{sign}\left(\nabla f(x^t)\right)\right\rangle\\
&= f(x^t)-\gamma^t\left\|\nabla f(x^t)\right\|_1-\gamma^t\left\langle\nabla f(x^{t+1})-\nabla f(x^t),\text{sign}\left(\nabla f(x^t)\right)\right\rangle\\
&\overset{Conj}{\leqslant} f(x^t)-\gamma^t\left\|\nabla f(x^t)\right\|_1+\gamma^t\left\|\nabla f(x^{t+1})-\nabla f(x^t)\right\|_\infty\left\|\text{sign}\left(\nabla f(x^t)\right)\right\|_1\\
&\leqslant f(x^t)-\gamma^t\left\|\nabla f(x^t)\right\|_1+\gamma^t d\left\|\nabla f(x^{t+1})-\nabla f(x^t)\right\|_\infty\\
&\overset{(i)}{=} f(x^t)-\gamma^t\left\|\nabla f(x^t)\right\|_1+\gamma^t d\frac{\left\|\nabla f(x^{t+1})-\nabla f(x^t)\right\|_\infty}{\left\|x^{t+1}-x^t\right\|_1}\left\|x^{t+1}-x^t\right\|_1\\
&= f(x^t)-\gamma^t\left\|\nabla f(x^t)\right\|_1+(\gamma^t)^2 d^2\frac{\left\|\nabla f(x^{t+1})-\nabla f(x^t)\right\|_\infty}{\left\|x^{t+1}-x^t\right\|_1},
\end{aligned}
$$

where in (i) we assume $\left\|x^{t+1}-x^t\right\|_1\neq 0$. Indeed, $\left\|x^{t+1}-x^t\right\|_1=0$ follows from the equality $\text{sign}\left(\nabla f(x^t)\right)=0$, which means that we find the optimum and do need to find another point $x^{t+1}$. Now we denote $L_1^t=\frac{\left\|\nabla f(x^{t+1})-\nabla f(x^t)\right\|_\infty}{\left\|x^{t+1}-x^t\right\|_1}$. Summing over all iterations, we obtain

$$\sum_{t=0}^{T-1}\gamma^t\left\|f(x^t)\right\|_1\leqslant\sum_{t=0}^{T-1}\left[f(x^t)-f(x^{t+1})\right]+\sum_{t=0}^{T-1}(\gamma^t)^2 d^2 L_1^t$$

$$= \quad f(x^0) - f(x^*) + \sum_{t=0}^{T-1} (\gamma^t)^2 d^2 L_1^t \leqslant \Delta^* + \sum_{t=0}^{T-1} (\gamma^t)^2 d^2 L_\infty^t,$$

which ends the proof of the lemma. $\qquad\square$

**Theorem F.12** (**Theorem 3.12**). *Suppose Assumptions 3.11, 3.2, 3.3, 3.4 hold. We denote $\varepsilon \geqslant \frac{1}{T} \sum_{t=0}^{T-1} \|\nabla f(x^t)\|_1, L_1^0 = \frac{\|\nabla f(x^1) - \nabla f(x^0)\|_\infty}{\|x^1 - x^0\|_1}$. Then Algorithm 2 with $d^0 < \Delta^*$ and $d \cdot \lambda^t$ as in equation 3 to reach $\varepsilon$-accuracy needs*

$$\widetilde{\mathcal{O}} \left( \frac{(\Delta^*)^2 (L_1)^3 d^2}{d^0 (L_1^0)^2 \varepsilon^2} \right) \quad and \quad \widetilde{\mathcal{O}} \left( \frac{\Delta^* (L_1)^3 d^2}{(L_1^0)^2 \varepsilon^2} \right) \quad iterations \ with \ Options \ I \ and \ II, \ respectively.$$

*Proof.* Let us start with the result of Lemma F.11:

$$\sum_{t=0}^{T-1} \gamma^t \|\nabla f(x^t)\|_1 \leqslant \Delta^* + \sum_{t=0}^{T-1} (\gamma^t)^2 d^2 L_1^t. \tag{62}$$

Now we use our $\gamma^t$ choice. Let us firstly estimate the denominator that is exactly $\lambda^t = \frac{1}{d\sqrt{\sum_{i=0}^{t-1} \frac{\|\nabla f(x^{i+1}) - \nabla f(x^i)\|_\infty}{\|x^{i+1} - x^i\|_1}}} = \frac{1}{d\sqrt{\sum_{i=0}^{t-1} L_1^i}}$ and is the same for both Options I and II. Let us estimate the following term.

$$\sum_{t=0}^{T-1} (\lambda^t)^2 d^2 L_1^t = \sum_{t=0}^{T-1} \frac{L_1^t}{\sum_{i=0}^{t-1} L_1^i}.$$

We mention, that each $L_1^i$ is bounded from the definition of smoothness (see Assumption 3.11), i.e., $L_1^i \leqslant L_1$. We consider the sequence $\{L_1^i\}_{i=0}^{T-1}$. Since each term in this sequence is bounded, there exists $r$ such that $\sum_{i=0}^{r-2} L_1^i \leqslant L_1^{r-1}$ and for each $t \geqslant r - 1$ such that $\sum_{i=0}^{t} L_1^i \geqslant L_1^{t+1}$. In that way, we divide the sum into two parts:

$$\sum_{t=0}^{T-1} \frac{L_1^t}{\sum_{i=0}^{t-1} L_1^i} = \sum_{t=0}^{r-1} \frac{L_1^t}{\sum_{i=0}^{t-1} L_1^i} + \sum_{t=r}^{T-1} \frac{L_1^t}{\sum_{i=0}^{t-1} L_1^i}. \tag{63}$$

Considering the first sum in equation 63, we mention, that we can estimate the denominator as $\sum_{i=0}^{t-1} L_1^i \geqslant L_1^0$. As for the numerator. Thus,

$$\sum_{t=0}^{r-1} \frac{L_1^t}{\sum_{i=0}^{t-1} L_1^i} \leqslant \frac{1}{L_1^0} \left( \sum_{t=0}^{r-2} L_1^t + L_1^{r-1} \right) \leqslant \frac{2 L_1^{r-1}}{L_1^0} \leqslant \frac{2 L_1}{L_1^0}. \tag{64}$$

Considering the second sum in equation 63, we have

$$\sum_{t=r}^{T-1} \frac{L_1^t}{\sum_{i=0}^{t-1} L_1^i} = \sum_{t=r}^{T-1} \frac{L_1^t}{\frac{1}{2} \sum_{i=0}^{t-1} L_1^i + \frac{1}{2} \sum_{i=0}^{t-1} L_1^i}.$$

Estimating any of the sums in the denominator, we claim, that $\sum_{i=0}^{t-1} L_1^i \geqslant L_1^t$, since $t - 1 \geqslant r - 1$. In that way,

$$\sum_{t=r}^{T-1} \frac{L_1^t}{\sum_{i=0}^{t-1} L_1^i} \leqslant \sum_{t=r}^{T-1} \frac{2L_1^t}{\sum_{i=0}^{t} L_1^i} \leqslant 2 \sum_{t=0}^{T-1} \frac{L_1^t}{\sum_{i=0}^{t} L_1^i}. \tag{65}$$

Next we denote $s^t = \sum_{i=0}^{t} L_1^t$ and have

$$L_1^t \frac{1}{\sum_{i=0}^{t} L_1^i} = (s^t - s^{t-1}) \frac{1}{\sum_{i=0}^{t} L_1^i} = \int_{s^{t-1}}^{s^t} \frac{1}{\sum_{i=0}^{t} L_1^i} dx \overset{(i)}{\leqslant} \int_{s^{t-1}}^{s^t} \frac{1}{x} dx, \tag{66}$$

where $(i)$ was done due to $\frac{1}{x}$ is a non-increasing function on $(0, +\infty)$. Summing over $t$, we obtain

$$2 \sum_{t=1}^{T} \frac{L_1^t}{\sum_{i=0}^{t} L_1^i} \leqslant 2 \int_{s^0}^{s^T} \frac{1}{x} dx = 2 \log(s^T) - 2 \log(s^0) = 2 \log \left( \frac{\sum_{t=1}^{T} L_1^t}{L_1^0} \right) \leqslant 2 \log \left( \frac{L_1 T}{L_1^0} \right).$$

Combining this estimate with equation 65,

$$\sum_{t=r}^{T-1} \frac{L_1^t}{\sum_{i=0}^{t-1} L_1^i} \leqslant 2 \sum_{t=1}^{T} \frac{L_1^t}{\sum_{i=0}^{t} L_1^i} + 2 \leqslant 2 \left( \log \left( \frac{L_1 T}{L_1^0} \right) + 1 \right) \leqslant 4 \log \left( \frac{L_1 T}{L_1^0} \right). \tag{67}$$

Substituting equation 64 and equation 67 into equation 63, we obtain

$$\sum_{t=0}^{T-1} (\lambda^t)^2 d^2 L_1^t \leqslant 2 \frac{L_1}{L_1^0} + 4 \log \left( \frac{L_1 T}{L_1^0} \right). \tag{68}$$

We additionally note, that if $r > T - 1$, only first term remains in this estimate, consequently our bound equation 68 is correct.

In this way, utilizing Option I from Algorithm 2, equation 62 together with equation 68 yields

$$\sqrt{d^0} \lambda^{T-1} \sum_{t=0}^{T-1} \|\nabla f(x^t)\|_1 \overset{(i)}{\leqslant} \sum_{t=0}^{T-1} \sqrt{d^t} \lambda^t \|\nabla f(x^t)\|_1 \leqslant \Delta^* + \sum_{t=0}^{T-1} d^t (\lambda^t)^2 d^2 L_1^t$$

$$\overset{\text{Lemma}F.1}{\leqslant} \Delta^* + \Delta^* \sum_{t=0}^{T-1} (\lambda^t)^2 d^2 L_1^t,$$

$$\sum_{t=0}^{T-1} \|\nabla f(x^t)\|_1 \leqslant \frac{\Delta^*}{\sqrt{d^0} \lambda^{T-1}} + \frac{\Delta^*}{\sqrt{d^0} \lambda^{T-1}} \sum_{t=0}^{T-1} (\lambda^t)^2 d^2 L_1^t$$

$$\overset{68}{\leqslant} \frac{\Delta^*}{\sqrt{d^0} \lambda^{T-1}} + 4 \frac{\Delta^*}{\sqrt{d^0} \lambda^{T-1}} \log \left( \frac{L_1 T}{L_1^0} \right) + 2 \frac{\Delta^* L_1}{\sqrt{d^0} \lambda^{T-1} L_1^0}$$

$$\leqslant 7 \frac{\Delta^* L_1}{\sqrt{d^0} \lambda^{T-1} L_1^0} \log \left( \frac{L_1 T}{L_1^0} \right), \tag{69}$$

where $(i)$ was done due to the fact that $d^0$ is minimal from all $\{d^t\}_{t=0}^{T-1}$ (Line 7 from Algorithm 2) and the definition of $\lambda^t$. Utilizing $\frac{1}{\lambda^{T-1}} = d\sqrt{\sum_{t=0}^{T-2} L_1^t} \leqslant d\sqrt{L_1 T}$, we obtain the final estimate:

$$\frac{1}{T} \sum_{t=0}^{T-1} \|\nabla f(x^t)\|_1 \quad \leqslant \quad \frac{7\Delta^* (L_1)^{\frac{3}{2}} d}{\sqrt{d^0 T} L_1^0} \log\left(\frac{L_1 T}{L_1^0}\right).$$

Expressing the number of iterations and using $\varepsilon \geqslant \frac{1}{T} \sum_{t=0}^{T-1} \|\nabla f(x^t)\|_1$ as a criterion, we obtain that

the algorithm needs $\widetilde{\mathcal{O}}\left(\frac{(\Delta^*)^2 (L_1)^3 d^2}{d^0 \left(L_1^0\right)^2 \varepsilon^2}\right)$ iterations to reach $\varepsilon$-accuracy.

Considering Option II from Algorithm 2, we can proceed absolutely analogical, however, using $f(x^0) - \widetilde{f} \geqslant \Delta^*$ instead of Lemma F.1. In that way,

$$\begin{aligned}
\frac{1}{T} \sum_{t=0}^{T-1} \|\nabla f(x^t)\|_1 \quad \leqslant \quad & \frac{\Delta^* \sqrt{L_1} d}{\sqrt{(f(x^0) - \widetilde{f})T}} + \frac{4(f(x^0) - \widetilde{f})\sqrt{L_1} d}{\sqrt{(f(x^0) - \widetilde{f})T}} \log\left(\frac{L_1 T}{L_1^0}\right) \\
& + \frac{2(f(x^0) - \widetilde{f}) (L_1)^{\frac{3}{2}} d}{\sqrt{(f(x^0) - \widetilde{f})T} L_1^0} \\
\leqslant \quad & \frac{7\sqrt{(f(x^0) - \widetilde{f}) (L_1)^{\frac{3}{2}}} d}{\sqrt{T} L_1^0} \log\left(\frac{L_1 T}{L_1^0}\right).
\end{aligned}$$

Expressing the number of iterations, using $\varepsilon \geqslant \frac{1}{T} \sum_{t=0}^{T-1} \|\nabla f(x^t)\|_1$ as a criterion, and utilizing $\widetilde{f}$ is

an approximation of $f(x^*)$, we obtain that the algorithm needs $\widetilde{\mathcal{O}}\left(\frac{\Delta^* (L_1)^3 d^2}{\left(L_1^0\right)^2 \varepsilon^2}\right)$ iterations to reach

$\varepsilon$-accuracy. $\qquad \square$

The proofs under stochastic and distributed settings for the memory-efficient version of ALIAS can be obtained analogously to Theorems F.6, F.9, and F.12.

## G  STEEPEST DESCENT

There is one more approach for sign descent. Classically, we perform the step in the direction of the gradient. However, we do not take into account the length of the gradient in any way in the step. The approach, called steepest descent, is supposed to utilize this information and provide the steps in the direction $\|\nabla f(x^t)\|_1 \text{sign}(\nabla f(x^t))$ at the $t$-th iteration. We provide the formal description of this approach (Algorithm 9).

---

**Algorithm 8** STEEPEST DESCENT

1: **Input:** Initial point $x^0 \in \mathbb{R}^d$, number of iterations $T$
2: **Parameter:** Stepsize $c > 0$
3: **for** $t = 0, \ldots, T-1$ **do**
4: $\qquad x^{t+1} = x^t - c\|\nabla f(x^t)\|_1 \text{sign}(\nabla f(x^t))$
5: **end for**

---

We present the analysis of SOS STEEPEST DESCENT. We start with the descent lemma.

**Lemma G.1** (Descent lemma). *For Algorithm 9 under Assumptions 3.1, 3.2, 3.3, 3.4, the following estimate is valid:*

$$-\Delta^* \leqslant -c_0 \sum_{t=0}^{T-1} \|\nabla f(x^t)\|_1^2 \left(1 - c_0 \widetilde{L}_\infty\right),$$

---

**Algorithm 9** SOS STEEPEST DESCENT

---

1: **Input:** Initial stepsize bound $c_s$, initial bound step $k$, initial point $x^0 \in \mathbb{R}^d$, number of iterations $T$

2: $c_0 = \text{BISECTION}\left(\phi(c), \frac{c_s}{2^{2^k}}, c_s, T\right)$   *// in Algorithm 4 we utilize Algorithm 8 instead of Algorithm 1*

3: $x^T = \text{STEEPEST DESCENT}(x^0, T, c_0)$

---

*where $\widetilde{L}_\infty = \max\limits_{0 \leqslant t \leqslant T-1} L_\infty^t$ and $L_\infty^t = \frac{\left\|\nabla f(x^{t+1}) - \nabla f(x^t)\right\|_1}{\left\|x^{t+1} - x^t\right\|_\infty}$.*

*Proof.* Starting from the convexity of the objective,

$$
\begin{aligned}
f(x^{t+1}) &\leqslant f(x^t) + \left\langle \nabla f(x^{t+1}), x^{t+1} - x^t \right\rangle = f(x^t) - \gamma^t \left\langle \nabla f(x^{t+1}), \text{sign}(\nabla f(x^t)) \right\rangle \\
&= f(x^t) - \gamma^t \left\langle \nabla f(x^t), \text{sign}(\nabla f(x^t)) \right\rangle \\
&\quad - \gamma^t \left\langle \nabla f(x^{t+1}) - \nabla f(x^t), \text{sign}(\nabla f(x^t)) \right\rangle \\
&\overset{Conj}{\leqslant} f(x^t) - \gamma^t \left\|\nabla f(x^t)\right\|_1 + \gamma^t \left\|\nabla f(x^{t+1}) - \nabla f(x^t)\right\|_1 \left\|\text{sign}(\nabla f(x^t))\right\|_\infty \\
&\leqslant f(x^t) - \gamma^t \left\|\nabla f(x^t)\right\|_1 + \gamma^t \left\|\nabla f(x^{t+1}) - \nabla f(x^t)\right\|_1 \\
&\overset{(i)}{=} f(x^t) - \gamma^t \left\|\nabla f(x^t)\right\|_1 + \gamma^t \frac{\left\|\nabla f(x^{t+1}) - \nabla f(x^t)\right\|_1}{\left\|x^{t+1} - x^t\right\|_\infty} \left\|x^{t+1} - x^t\right\|_\infty,
\end{aligned}
$$

where in (*i*) we assume $\left\|x^{t+1} - x^t\right\|_\infty \neq 0$. Indeed, $\left\|x^{t+1} - x^t\right\|_\infty = 0$ follows from $\text{sign}(\nabla f(x^t)) = 0$, which means we find the optimum and do need to search the point $x^{t+1}$. Now we denote $L_\infty^t = \frac{\left\|\nabla f(x^{t+1}) - \nabla f(x^t)\right\|_1}{\left\|x^{t+1} - x^t\right\|_\infty}$. Continue estimate,

$$
\begin{aligned}
f(x^{t+1}) &\leqslant f(x^t) - \gamma^t \left\|\nabla f(x^t)\right\|_1 + (\gamma^t)^2 L_\infty^t \left\|\text{sign}(\nabla f(x^t))\right\|_\infty \\
&\leqslant f(x^t) - \gamma^t \left\|\nabla f(x^t)\right\|_1 + (\gamma^t)^2 L_\infty^t.
\end{aligned}
$$

Now we choose $\gamma^t = c_0 \|\nabla f(x^t)\|_1$, where we find the constant $c_0$ using BISECTION procedure (Algorithm 4). Thus,

$$
\begin{aligned}
f(x^{t+1}) &\leqslant f(x^t) - c_0\|\nabla f(x^t)\|_1^2 + c_0^2 \|\nabla f(x^t)\|_1^2 L_\infty^t \\
&= f(x^t) - c_0 \|\nabla f(x^t)\|_1^2 \left(1 - c_0 L_\infty^t\right).
\end{aligned}
$$

Summing over all iterations and utilizing $\widetilde{L}_\infty = \max\limits_{0 \leqslant t \leqslant T-1} L_\infty^t$ notation, we have

$$
-\Delta^* = f(x^*) - f(x^0) \leqslant f(x^T) - f(x^0) \leqslant -c_0 \sum_{t=0}^{T-1} \|\nabla f(x^t)\|_1^2 \left(1 - c_0 \widetilde{L}_\infty\right),
$$

which ends the proof of the lemma. $\qquad\square$

Now we present the purposes of Algorithm 4. Let us take an arbitrary point $x^{-1} \in \mathbb{R}^d$. We denote $L_\infty^{-1} = \frac{\left\|\nabla f(x^0) - \nabla f(x^{-1})\right\|_1}{\left\|x^0 - x^{-1}\right\|_\infty}$ and $\widetilde{L}_\infty^{-1} = \max\limits_{-1 \leqslant t \leqslant T-1} L_\infty^t$. It is obvious that it implies

$$
\begin{aligned}
L_\infty^{-1} &\leqslant \widetilde{L}_\infty^{-1} \leqslant L_\infty, \\
\widetilde{L}_\infty &\leqslant \widetilde{L}_\infty^{-1}.
\end{aligned}
\tag{70}
$$

Let us put $\phi(c) = \frac{1}{\widetilde{L}_\infty^{-1}(c)}$ in the BISECTION procedure. The following lemma shows guarantees of $\phi(c_{\text{hi}}) \leqslant c_{\text{hi}}$ and $\phi(c_{\text{lo}}) \geqslant c_{\text{lo}}$.

**Lemma G.2** (Bisection entry). *Let $c_{\max} = \frac{1}{L_\infty^{-1}}$. Thus, with the initial $c_{hi} = c_{\max}$, Algorithm 4 always avoids an early infinite termination. Moreover, with the initial $c_{lo} = \frac{1}{2^{2^k}} c_{hi}$, where $k \geqslant \log\log \frac{L_\infty}{L_\infty^{-1}}$, Algorithm 4 always avoids early non-infinite termination.*

*Proof.* Let us start with $c_{\text{hi}}$. The choice of $c_{\max}$ implies

$$c_{\text{hi}} = c_{\max} = \frac{1}{L_\infty^{-1}} \overset{70}{\geqslant} \frac{1}{\widetilde{L}_\infty^{-1}(c_{\text{hi}})} = \phi(c_{\text{hi}}),$$

which means we avoid early infinite termination. As for $c_{\text{lo}}$:

$$c_{\text{lo}} = \frac{1}{2^{2^k}} c_{\text{hi}} \leqslant \frac{1}{\frac{L_\infty}{L_\infty^{-1}}} \cdot \frac{1}{L_\infty^{-1}} = \frac{1}{L_\infty} \overset{70}{\leqslant} \frac{1}{\widetilde{L}_\infty^{-1}(c_{\text{lo}})} = \phi(c_{\text{lo}}),$$

which means we avoid early non-infinite termination. $\qquad\square$

Since we always entry to the BISECTION procedure, we are under the performing of Lemma D.3. Now we are ready to prove the final convergence guarantees for SOS STEEPEST DESCENT.

**Theorem G.3.** *Suppose Assumptions 3.1, 3.2, 3.3, 3.4 hold. Then for Algorithm 9 after obtaining the stepsize $c_0$, the following estimate is valid:*

$$\frac{1}{T} \sum_{t=0}^{T-1} \|\nabla f(x^t)\|_1^2 \leqslant 8 \frac{\Delta^* L_\infty}{T}.$$

*Moreover, taking into account the complexity of Algorithm 4 in relation to the initial stepsize bound $c_s$, to reach $\varepsilon$-accuracy, where $\varepsilon^2 \geqslant \frac{1}{T} \sum_{t=0}^{T-1} \|\nabla f(x^t)\|_1^2$, Algorithm 9 needs*

$$\mathcal{O}\left(\frac{\Delta^* L_\infty}{\varepsilon^2} \log\log \frac{L_\infty}{L_\infty^{-1}}\right) \ \text{iterations.}$$

*Proof.* Firstly, we recall the result of Lemma G.1:

$$-\Delta^* \leqslant -c_0 \sum_{t=0}^{T-1} \|\nabla f(x^t)\|_1^2 \left(1 - c_0 \widetilde{L}_\infty\right).$$

We have already mentioned that we can always avoid early terminations of Algorithm 4, due to Lemma G.2, and thus, $\frac{1}{2\widetilde{L}_\infty^{-1}(c_{\text{hi}}^*)} \leqslant c_0 \leqslant \frac{1}{\widetilde{L}_\infty^{-1}(c_0)}$. Tuning $c_0 = \frac{c_0}{2}$, we obtain

$$
\begin{aligned}
-\Delta^* &\leqslant -c_0 \sum_{t=0}^{T-1} \|\nabla f(x^t)\|_1^2 \left(1 - \frac{1}{2\widetilde{L}_\infty^{-1}(c_0)} \widetilde{L}_\infty(c_0)\right) \\
&\overset{70}{\leqslant} -c_0 \sum_{t=0}^{T-1} \|\nabla f(x^t)\|_1^2 \left(1 - \frac{1}{2}\right).
\end{aligned}
$$

Expressing gradient norms, we obtain

$$\frac{1}{T} \sum_{t=0}^{T-1} \|\nabla f(x^t)\|_1^2 \leqslant \frac{2\Delta^*}{c_0 T} \leqslant \frac{8\Delta^* \widetilde{L}_\infty^{-1}(c_{\text{hi}}^*)}{T} \overset{70}{\leqslant} \frac{8\Delta^* L_\infty}{T}.$$

Assuming $\frac{1}{T} \sum_{t=0}^{T-1} \|\nabla f(x^t)\|_1^2 \leqslant \varepsilon^2$ as a criterion, we easily obtain the estimate on the number of iterations required — $\mathcal{O}\left(\frac{\Delta^* L_\infty}{\varepsilon^2}\right)$. Mention that the total number of iterations (together with the Algorithm 4 performance) – $\mathcal{O}\left(\frac{\Delta^* L_\infty}{\varepsilon^2} \log\log \frac{L_\infty}{L_\infty^{-1}}\right)$. $\qquad\square$

## THE USE OF LARGE LANGUAGE MODELS (LLMS)

In this work, large language models (LLMs) were used exclusively for spelling edits.

