# OpenReview forum: "Sign-SGD via Parameter-Free Optimization"
_ICLR.cc/2026/Conference — ICLR 2026 Poster_

### Official Review · Reviewer_HNri · 2025-10-20

**Soundness:** 3
**Presentation:** 3
**Contribution:** 3
**Rating:** 6
**Confidence:** 3

**Summary:**

The paper proposes a parameter-free variant of SIGN-SGD that automatically determines the learning rate using local gradient information without requiring manual tuning. The method estimates the step size via per-iteration approximations of the smoothness constant and loss gap, extending to stochastic and distributed settings. Theoretical analysis under convex assumptions supports convergence, and experiments on language and vision tasks show comparable performance to well-tuned AdamW and SIGN-SGD baselines, while removing the need for hyperparameter search.

**Strengths:**

The paper tackles a practical and long-standing issue in optimization—manual learning rate tuning—by introducing a parameter-free variant of SIGN-SGD. The proposed formulation is conceptually simple and mathematically grounded, extending classical gradient clipping and step-size estimation ideas in a coherent way. The theoretical analysis is well-presented and the experiments, though limited, are consistent and demonstrate that the approach can achieve competitive performance without hyperparameter tuning.

**Weaknesses:**

Theoretical guarantees are proved under convex smooth assumptions, while the main experiments are on large non-convex models; the gap between theory and practice is not bridged.

Experimental coverage is limited (few architectures/tasks and moderate scales), so generality across modalities and training regimes remains unclear.

The parameter-free step-size depends on noisy quantities (e.g., gradient-norm variation and loss gaps); there is no robustness/sensitivity analysis under high-noise or distributed settings.

**Questions:**

Can the authors clarify how the proposed step-size estimation behaves under high gradient noise or distributed training, where loss variations can be inconsistent?

How sensitive is the performance to the local smoothness and loss-gap approximations used in the parameter-free update rule?

Do the authors plan to extend the theoretical analysis beyond convex assumptions to better align with the non-convex experiments presented?

---

> ### Author Response · Authors · 2025-11-19
>
> Dear Reviewer HNri,
>
> Thanks to the reviewer for taking the time to review our work! Below, we respond to their concerns.
>
> 1. > Non-convex assumption (W1/Q3)
>
> Although neural networks are non-convex, conducting theoretical analysis under convexity assumptions remains meaningful. Recent research indicates that deep neural networks often display locally quasi-convex behavior in certain regions, allowing insights from convex analysis to be informative [1], [2], [3]. Furthermore, convex optimization provides a rigorous theoretical basis for the development of optimization algorithms. For instance, methods with  momentum [4] and AdaGrad [5] were originally proposed and theoretically studied in the context of convex problems. However, we recognize this gap and designate it as future work, while viewing such an analysis as a distinct research direction.
>
> 2. > Limited experiments (W2)
>
> In our work, we provide two distinct series of experiments: LLM pre-training and fine-tuning a ViT architecture for image classification. These are entirely different tasks, and in both cases, we demonstrate the stability of our methods and their ability to achieve strong metrics. However, we agree with the reviewer that including additional domains could further strengthen the experimental section of our work. Therefore, we additionally evaluate it on several tasks from the AlgoPerf benchmark [5]. In particular, we test our method on the MRI reconstruction and molecular property prediction (MPP) tasks. Due to the limited time of the discussion period, we currently include comparison results only on these two tasks, and plan to evaluate our method on the full benchmark in the future. We validate only our ALIAS Adam version, while the results for the remaining methods are taken directly from Table 4 in [5]. For our method, we fix $\gamma^t = 10^{-3}$. The corresponding results are provided in the revised version of the paper on OpenReview in Appendix A.3 (Table 11) and in the table below.
>
> |Algorithm|MRI (SSIM $\uparrow$)|MPP (MAP $\uparrow$)|
> |-|-|-|
> |AdamW|0.723|0.254|
> |DoG|0.714|0.231|
> |D-Adapt (with Adam)|0.722|0.221|
> |MoMo (with Adam)|0.723|0.221|
> |Prodigy|0.723|0.212|
> |ALIAS Adam version|0.724|0.242|
>
> The results demonstrate that our approach outperforms competing parameter-free algorithms on both tasks, and additionally surpasses the tuned AdamW baseline on the MRI reconstruction problem. These experiments further confirm the applicability of our method to tasks of different modalities, as well as its advantage over other parameter-free approaches.
>
> 3. > Robustness to gradient noise (W3/Q1)
>
> We agree with the reviewer that gradient noise has a noticeable impact in stochastic or distributed neural network training. To this end, we conducted the following ablation study. We evaluate our method and Sign-SGD for LLM pre-training with smaller batch sizes $-$ 256, 128, and 64 sequences, thereby increasing the gradient noise. In the original runs, the batch size was 512 sequences. We provide the following comparative table (see also Appendix A.1.2 (Table 6) of the revised version of the paper).
>
> |# Sequences|Algorithm|Loss ($\downarrow$)|
> |-|-|-|
> |512|Sign-SGD|2.980|
> |512|ALIAS|3.006|
> |256|Sign-SGD|2.986|
> |256|ALIAS|3.013|
> |128|Sign-SGD|2.992|
> |128|ALIAS|3.021|
> |64|Sign-SGD|2.999|
> |64|ALIAS|3.029|
>
> The results show that, with the reduced batch size, the performance of both Sign-SGD and ALIAS decreased similarly, indicating that our method is robust to gradient noise.
>
> 4. > Sensitivity of performance on local smoothness and loss-gap approximations (Q2)
>
> If we understand the reviewer’s question correctly, their concern is about how robust our method is with respect to different landscapes of the objective function (therefore, different local smoothness and loss-gap values). Answering this question, we refer to our versatile experimental settings and various architectures (see answer in point 2 for details). As in all settings our method demonstrates consistent results, this directly connects it to the robustness of such approximations.
>
> ---
>
> **References**
>
> [1] Kleinberg et al. An alternative view: When does sgd … **ICML** 2018
>
> [2] Zhou et al.. Sgd converges to global minimum … **ICLR** 2019
>
> [3] Liu et al. Loss landscapes and optimization in over-parameterized … **Applied and Computational Harmonic Analysis** 2022
>
> [4] Nesterov. Lectures on convex optimization. **Springer** 2018
>
> [5] Duchi et al. Adaptive subgradient methods … **JMLR** 2011

---

> ### Author Response · Authors · 2025-11-27
>
> Dear Reviewer HNri,
>
> We would like to gently follow up on our earlier rebuttal. We understand the considerable workload during the review period and sincerely appreciate the time you devote to reviewing submissions.
>
> In addition, we kindly ask you to take a look at the new experimental results included in the revised version $-$ particularly our ablation study on the robustness of our method to gradient noise, as well as the additional validation we performed on other domains. We would be grateful for any feedback or comments you may have regarding these additions.
>
> Thank you very much for your time and consideration.
>
> Best regards,
> Authors

---

> > ### Comment · Reviewer_HNri · 2025-11-27
> >
> > Thank you for the detailed rebuttal and the extra experiments — they have resolved my major concerns.
> > Although I initially gave a 6, I’ve decided to raise my score, as I believe the paper is strong enough for an oral.
> > I’ll leave the final decision to the AC.
> > Good luck!

---

> > > ### Author Response · Authors · 2025-11-28
> > >
> > > We thank Reviewer HNri for the time devoted to our work, for the insightful discussion of our results, and for the valuable suggestions that helped us improve the paper!

---

### Official Review · Reviewer_RrFQ · 2025-10-26

**Soundness:** 3
**Presentation:** 4
**Contribution:** 4
**Rating:** 8
**Confidence:** 3

**Summary:**

This paper revisits the SIGN-SGD optimizer and proposes a novel parameter-free variant that eliminates the need to manually tune the learning rate. The core idea is an algorithm called ALIAS (Automatic Local per-Iteration Approximation of the Step size), which adaptively adjusts the step size at each iteration using the current gradient information. ALIAS estimates problem-specific constants on-the-fly, enabling SIGN-SGD to approach its theoretically optimal step size without needing prior knowledge. The method is developed first for the idealized deterministic setting and then extended to stochastic training and distributed training, addressing a known gap in prior parameter-free methods. The authors further introduce two practical enhancements: 1) a momentum-integrated version (ALIAS ADAM version) that incorporates ADAM-style 1st and 2nd moment accumulations for variance reduction, and (2) a memory-efficient modification that stores only the sign of the previous gradient to preserve the low memory footprint of SIGN-SGD.

Empirical evaluation are conducted on both LLM pre-training and vision model fine-tuning. The results show that ALIAS matches or closely approaches the performance of well-tuned optimizers (like SIGN-SGD with optimally tuned constant or cosine-scheduled LR, and even AdamW) but without any manual tuning. By removing the costly hyperparameter search, ALIAS yields about a 1.5x end-to-end speedup in training for large models. Overall, the paper’s contributions span theoretical guarantees for parameter-free SIGN-SGD and strong practical validation that this approach can simplify and accelerate large-scale training without sacrificing accuracy.

**Strengths:**

1. Novel param-free algorithm for SIGN-SGD: the paper introduces a novel optimizer (ALIAS) that requires no manual step size tuning for SIGN-SGD. This is an original contribution, as prior sign-based methods either fixed a step size or relied on problem-dependent choices. ALIAS uses a clever per-iteration adaptation that accumulates an estimate of the local smoothness and the loss gap $\Delta^*$ to adjust $\gamma_t$ automatically. This approach is param-free in that it doesn’t need line searches or doubling tricks each round, and it doesn’t introduce additional hyperparameter to tune. The authors also propose two extensions: a momentum-incorporated version and a memory-saving version, which further increase the impact and applicability of the method.

2. Theoretical guarantees: The work provides a detailed theoretical analysis in the convex setting, proving convergence rates for the proposed algorithms. The proofs build on and extend the literature (e.g., adapting techniques from adaptive online learning and recent parameter-free methods) and show that ALIAS achieves convergence within a logarithmic factor of the optimal tuned SIGN-SGD. The authors manage to handle the sign operator in analysis: they obtain convergence guarantees in terms of gradient norm, which is appropriate since sign methods inherently target stationarity. They also handle the stochastic case with theoretical guarantees (theorem 3.9 for minibatch SGD) and outline how the method works in distributed training (Appendix F). This adds comprehensiveness to the coverage and shows the method isn’t a brittle special-case solution.

3. Empirical result: The authors pre-train transformer models on C4 dataset and also fine-tune a vision transformer in Appendix, demonstrating generality. ALIAS is shown to be competitive with SOTA optimizers. For instance, validation loss of ALIAS on LLaMA-130M is within 1% of SIGN-SGD with an expertly tuned LR schedule. With momentum, ALIAS even slightly outperforms AdamW. These results substantiate their claim that the new method without tuning can match the accuracy of well-tuned baselines. The results are reinforced by the training curves (Fig.1) which show that the convergence trajectory of ALIAS closely tracks that of tuned existing optimizers throughout training. The authors report metrics like end-to-end speedup (1.5x faster training when tuning is removed) which emphasizes the practical benefit of the method in real work.

**Weaknesses:**

1. Theoretical assumption: the theoretical guarantees are restricted to convex optimization (w/ smoothness assumption) and are expressed in terms of finding stationary points (grad norm). While this is standard for sign-based methods, it means the theory doesn’t directly guarantee improvement on the non-convex training objectives that the experiments address. The authors do discuss why a convex analysis is still informative for sign methods, but the strongest guarantees (Theorems 3.5, 3.9) hold under convexity or particular smoothness bounds. The empirical results suggest the method works for non-convex training but it would be better to close the gap between theory and expeirment.
2. Momentum version is not fully param-free: the ADAM-style version reintroduces a LR parameter and momentum hyperparameter. The momentum version behaves more like AdamW. This is a weakness if one expected the momentum version to also automatically adjust its global scale. Similarly, the authors also had to tune weight decay via validation (0, 0.01, 0.1) for all methods including theirs.

**Questions:**

1. Adaptivity to initial guess: ALIAS requires an initial approximation $d_0$ for $\Delta^* = f(x_0) - f(x^)$ (or alternatively a known lower bound $f_e$ for $f(x^)$ in Option II). How sensitive is the method to this initial guess? If $d_0$ is set much smaller or larger than the true $\Delta^*$, does ALIAS quickly correct the estimate via the $d_t = \max(d_{t-1},\ \cdot)$ update, or could a poor $d_0$ slow down convergence initially?
2. In momentum-integrated ALIAS, there are new hyperparameters: $\beta_1,\beta_2$ and base step size $\gamma$ for the update. In your experiments, it appears you fixed these to standard ADAM values ($\beta_1=0.9,\ \beta_2=0.999$) and chose $\gamma_t=10^{-3}$ without further tuning. To what extent can these be considered default that work across tasks?
3. Just for clarification: in Appendix F3,  ALIAS is extended to the distributed scenario. A question here is how the adaptive step size interacts with data-parallel distributed training: Do all ranks synchronize the value of $\gamma_t$ each iteration, or does each node run its own version of ALIAS based on local gradient estimates?

---

> ### Author Response · Authors · 2025-11-19
>
> Dear Reviewer RrFQ,
>
> Thanks to the reviewer for their time and high evaluation of our work! Below, we respond to their concerns.
>
> 1. > Non-convex assumption (W1)
>
> In our work, we provide a convex analysis of the proposed method and its variants. Extending this analysis to the non-convex setting is challenging within our current proof framework. We recognize this limitation and view a non-convex analysis as a promising direction for future research, albeit one that we believe constitutes a distinct line of investigation.
>
> 2. > Dependence of momentum-based ALIAS on hyperparameters (W2)
>
> We agree with the reviewer that the ALIAS Adam version (Algorithm 3) is not fully parameter-free, as it still requires setting the momentum and weight decay parameters. However, the main goal of our work is to propose a scheme that eliminates the need for manual stepsize tuning. The question of adaptivity to the momentum and weight decay parameters follows a different theoretical intuition and analysis, and, in our view, represents a separate research problem.
>
> 3. > Sensitivity of performance on $d^0$ (Q1)
>
> This is an important ablation study that demonstrates the robustness of our method with respect to the initialization of this parameter. We conducted this analysis and added it to the revised version of our paper in Appendix A.1.2, which we uploaded to OpenReview. A brief summary of the results is provided in the table below.
>
> |$\Delta^*$|$d^0$|Loss ($\downarrow$)|
> |-|-|-|
> |$4$|$10^{-3}$|2.918|
> |$4$|$1$|2.918|
>
> The sequence $d^t$ rapidly reaches its limiting value regardless of its initial initialization and, as a result, does not affect the final convergence of the method.
>
> 4. > Hyperparameters for ALIAS Adam version / AdamW (Q2)
>
> a) In our work, we present experimental results across different tasks $-$ LLM pre-training and ViT fine-tuning for image classification. In both setups, we perform preliminary hyperparameter tuning for baselines, as described in Appendix A.1.1 and Appendix A.2.1 (for more details, we refer the reviewer to point 4a of our response to reviewer uXJ9). We obtain a similar set of optimal hyperparameters across both tasks. Regarding our method, we fix the stepsize $\gamma^t = 10^{-3}$ across all experimental setups. As demonstrated by our experiments on both LLaMA-based pre-training and image classification with ViT architectures, this setting yields strong performance for our approach.
>
> b) Furthermore, we conducted additional evaluations using several setups from the AlgoPerf benchmark [1]. In particular, we validated our method on the MRI reconstruction and molecular property prediction tasks. We adopted the datasets and models from AlgoPerf without modification. For these experiments, we evaluated only our ALIAS Adam version, keeping the stepsize fixed at $10^{-3}$. The corresponding results are provided in the revised version of the paper in Appendix A.3 (Table 11) and summarized in the table below.
>
> |Algorithm|MRI (SSIM $\uparrow$)|MPP (MAP $\uparrow$)|
> |-|-|-|
> |AdamW|0.723|0.254|
> |DoG|0.714|0.231|
> |D-Adapt (with Adam)|0.722|0.221|
> |MoMo (with Adam)|0.723|0.221|
> |Prodigy|0.723|0.212|
> |ALIAS Adam version|0.724|0.242|
>
> As the results show, our method with $\gamma^t = 10^{-3}$ achieves strong performance across fundamentally different tasks. We would like to emphasize that, on both tasks, our approach outperforms the competing parameter-free algorithms, and on the MRI reconstruction task it even surpasses the heavily tuned AdamW. For a detailed discussion of baseline tuning, we kindly refer the reviewer to point 4b of our response to Reviewer uXJ9.
>
> 5. > Distributed algorithm (Q3)
>
> In Appendix F3, we provide the equation for computing the parameter-free stepsize. Indeed, this requires that devices send the norms of the differences of local stochastic gradients to the server, where they are synchronized to perform a global update. The algorithm does not require applying the ALIAS mechanism locally on each device.
>
> ---
>
> **References**
>
> [1] Kasimbeg et al. How far away are truly hyperparameter-free learning algorithms? **TMLR** 2025

---

### Official Review · Reviewer_JFUp · 2025-11-01

**Soundness:** 2
**Presentation:** 2
**Contribution:** 3
**Rating:** 4
**Confidence:** 3

**Summary:**

The authors propose a variant of the sign-GD algorithm. By estimating the Lipschitz and decaying factor during the optimization process, the authors give the algorithm with theoretical analysis on the exact case, the stochastic case, and the memory-efficient case.  The experimental results on the LLaMA 130M model show that the algorithm performance is similar or even slightly better than the tuning version.

**Strengths:**

1. The authors propose a hyper-parameter-free sign-GD algorithm with its stochastic version and memory-efficient version.

2. For all versions, the authors give a convergence analysis.

3. The experimental results of the stochastic version and memory-efficient version work well on 130M model.

**Weaknesses:**

1. It is not clear what algorithm it is in the stochastic version, especially in the calculation of $d_t$.

2. The expression of $\epsilon = xxx$ is confusing. It seems that $\epsilon$ is also related to the $T$ and gradient norm. But according to the proof, it should be $xxx \leq \epsilon$ when $T \leq O(...)$.

3. It is well-known that sign-sgd will not converge. It is unclear how the algorithm overcomes the issue.

**Questions:**

See Weaknesses.

---

> ### Author Response · Authors · 2025-11-13
>
> Dear Reviewer JFUp,
>
> Thanks to the reviewer for the time and evaluation of our work! Below, we respond to their concerns.
>
> 1. > The expression is confusing.
>
> In our theoretical estimates, we meant that the criterion is less than $\varepsilon$. We have corrected this typo, thanks.
>
> 2. > It is not clear what algorithm it is in the stochastic version.
>
> We propose two options regarding the choice of $d^t$ in our base algorithm ALIAS (Algorithm 2): a growing sequence upper-bounded by the initial approximation $f(x^0) - f(x^*)$, and an upper estimate of the initial approximation $f(x^0) - \widetilde{f}$. As for the stochastic version of the algorithm, in practice we use the first option, where the sequence $d^t$ is updated using the corresponding stochastic gradients: $\widetilde{d}^t = \sum\limits_{i=0}^{t-1} \gamma^i \langle g_{\xi^{i+1}}^{i+1}, \text{sign}(g_{\xi^{i+1}}^{i})\rangle$. We use the same stochasticity for both gradients, similar to the smoothness constant approximation, to reduce noise. In theory, however, we focus on the second option, which remains unchanged in the analysis of the stochastic case. Thanks to the reviewer for pointing this out. We have added a formal description of both the practical and theoretical versions of the stochastic algorithm (Algorithm 7) in the revised version of the paper, which we uploaded to OpenReview in Appendix F2. Please, note that we provide an analysis of the stochastic version of our algorithm, in contrast to previous parameter-free works [1], [2], [3], and **do not rely** on the unrealistic stochastic gradient bound assumption used in [4], [5].
>
> 3. > sign-sgd will not converge.
>
> As we mention in Section 2.1, stochastic sign descent does not converge in terms of the variance of stochastic gradients. In the original work [6], convergence to a neighborhood was established, and it was suggested to use mini-batches of increasing size to demonstrate variance convergence. Subsequent works related to Sign-SGD have explored various techniques to ensure convergence, such as introducing error compensation [7]. In our work, we do not focus on these approaches, since our main goal is to propose a parameter-free version of the algorithm. Without using mini-batches, we obtain convergence to a neighborhood; by incorporating them as in [6], the variance converges. Moreover, we would like to note that in the deterministic setting, the method converges, as we demonstrate in Theorem 3.5. We emphasize that our parameter-free technique does not affect the convergence properties of the method and remains consistent with its original convergence behavior. Moreover, in our experiments, we **do not use increasing mini-batches**. It is worth noting that although Sign-SGD does not have promising theoretical guarantees (according to the original paper [6]), it remains a widely used and effective method in various ML applications [8], [9], [10].
>
> We hope that we have fully addressed all of the reviewer’s concerns and remain open to further discussion. If we have adequately clarified their doubts, then in light of this and the positive feedback from other reviewers, we kindly ask the reviewer to reconsider the score.
>
> ---
>
> **References**
>
> [1] Defazio and Mishchenko Learning-rate-free learning by d-adaptation. **ICML** 2023
>
> [2] Khaled et al. Dowg unleashed …**NeurIPS** 2023
>
> [3] Schaipp et al. MoMo: Momentum Models for Adaptive Learning Rates. **ICML** 2024
>
> [4] Carmon et al. Making sgd parameter-free. **COLT** 2022
>
> [5] Ivgi et al. Dog is sgd’s best friend … **ICML** 2023
>
> [6] Bernstein et al. signsgd: Compressed optimisation … **ICML** 2018
>
> [7] Karimireddy et al. Error feedback fixes signsgd … **ICML** 2019
>
> [8] Kunstner et al. Noise is not the main factor behind the gap … **ICLR** 2023
>
> [9] Zhao et al. Deconstructing what makes a good optimizer for language models. **OPT** 2024
>
> [10] Zmushko et al. FRUGAL: Memory-Efficient Optimization … **ICML** 2025

---

> ### Author Response · Authors · 2025-11-19
>
> We kindly ask the reviewer to take note of the new experimental results that we have provided in our responses to the other reviewers, as well as in the revised version of the paper in Appendix A.1.2, A.1.3, and A.3.

---

> ### Author Response · Authors · 2025-11-27
>
> Dear Reviewer  JFUp,
>
> We would like to gently follow up on our previous rebuttal. We understand the substantial reviewing load and sincerely appreciate the time you dedicate to the evaluation process.
>
> We also kindly ask you to take a look at the new experimental results we have added in the revised version (as also discussed in our responses to the other reviewers). If possible, we would greatly appreciate any feedback or comments you may have.
>
> Thank you very much for your time and consideration.
>
> Best regards,
> Authors

---

> > ### Comment · Reviewer_JFUp · 2025-11-28
> >
> > Thanks for the authors' detailed response. I have no more questions, and I will increase the score to 6.

---

> > > ### Author Response · Authors · 2025-11-28
> > >
> > > We thank Reviewer JFUp for the time and effort dedicated to evaluating our work, as well as for the important comments that helped us improve the paper! We kindly look forward to the score update on OpenReview.

---

### Official Review · Reviewer_uXJ9 · 2025-11-09

**Soundness:** 3
**Presentation:** 2
**Contribution:** 3
**Rating:** 6
**Confidence:** 3

**Summary:**

This paper presents a variant of SignSGD which does not require manual step size selection for a given optimization task and the proposed ALIAS method provides stepsize prescription at each iteration guided by theory. Experiments on llama (upto 350M) and swin-transformer (28M) tasks show that ALIAS is competitive against tuned setups with learning rate schedules.

**Strengths:**

* The paper tackles an important problem of mitigating hyperparameter tuning of optimizers which is quite relevant in the current era of LLMs.
* The proposed algorithm, guided by theory, is simple to implement and has practical benefits such as lower costs in single-node/distributed settings and no need for LR tuning. Also, kudos to authors for providing a memory-efficient version of the algorithm that is already efficient relative to SOTA optimizers such as AdamW.

**Weaknesses:**

* There are a bunch of hyperparameter-free methods that also eliminate the need for manually setting step sizes [1] but no such baselines are presented in this work, why?
* Appendix A.1.1 mentions that only main model params are optimized by ALIAS and LM head params are optimized by AdamW — this detail is quite critical and must be mentioned in the main paper. How is AdamW learning rate selected for LM head params? Is it tuned through grid-search? If yes, then isn’t the point of param-free / saving tuning effort redundant since one can simply tune main model params while tuning LM head params? Moreover, how does ALIAS perform if it is also used for LM head params?
* It looks ALIAS SignSGD performs a bit worse compared to tuned LR setup with cosine schedule (Table 1) and ALIAS Adam version also benefits from LR schedules (Figure 1). Can authors please confirm if this work eliminates the need for LR tuning but the problem of tuning schedules still remains to achieve the optimal performance?
* There is a scope in improving the experimental setup. AdamW b2=0.999 as used in pretraining experiments isn’t standard, b2=0.95 is generally used and ideally should be tuned as per the setup. Also, it’s unclear if the method generalizes to different tasks such as those in AlgoPerf [1].

[1] Kasimbeg et al. “How far away are truly hyperparameter-free learning algorithms?” TMLR 2025. https://arxiv.org/abs/2505.24005

**Questions:**

See weaknesses above.

Additional question: can authors expand on LR and other hyperparameters (e.g. weight decay, schedule hyperparams) search space used for tuning the baselines? How many hyperparameter trials were performed for each baseline and the proposed approach?

---

> ### Author Response · Authors · 2025-11-19
>
> Dear Reviewer uXJ9,
>
> Thanks to the reviewer for taking time to review our work! Below, we respond to their concerns.
>
> 1. > Additional parameter-free baselines (W1)
>
> We note that in our experimental setup we compare our method with Prodigy and witness a noticeable improvement. As shown in [1] Prodigy, outperforms other baselines, specifically, DoG [2] and D-Adaptation [3]. However, for additional justification, we have included some baselines that do not require manual hyperparameter tuning. Specifically, we selected DoG [2], D-Adaptation [3], and MoMo [4] as they were among the best methods for transformer architecture according to the results in [5]. We evaluated them on the LLaMA-based architecture pre-training task, fully preserving the setup described in Section A.1.1 of the paper. The results are presented in the revised version of the paper in Section A.1.3 (Table 7). For the reviewer’s convenience, we provide a summary of these results in the table below.
>
> | Algorithm|Loss ($\downarrow$)|
> |-|-|
> | DoG| 2.939|
> | D-Adapt (with Adam)| 2.927|
> | MoMo (with Adam)| 2.925|
> | ALIAS Adam version (ours)| 2.918|
>
> The results confirm that our approach outperforms these baselines.
>
> 2. > LM head parameters (W2)
>
> a) First and foremost, we emphasize that replacing the optimizer with AdamW for the LM head parameters is not specific to our method, ALIAS. In a recent work [6], the authors progressively applied optimization using sign-based methods, including Sign-SGD with momentum, to layers of the model. This approach demonstrated strong performance up to the point where the LM head parameters were optimized using sign descent. By optimizing the LM head parameters with AdamW and using Sign-SGD for the remaining parameters, we achieve better metric values.
>
> b) It is important to note that we modify the optimizer **only** for these parameters in the final layer. To give concrete numbers for the LLaMA-based architecture with 130M parameters, 110M parameters are optimized using our optimizer, while only 20M are optimized with AdamW. Note that the relative number of such parameters decreases as the model size grows. For instance, for the 350M-parameter model, on which we also validate our methods in the paper, their number amounts to 30M.
>
> c) Moreover, to ensure a fair comparison with baselines, we consistently used AdamW for optimizing the LM head in the final layer. Regarding the stepsize tuning for these parameters, for fairness, we did not perform any tuning and used the default value of $10^{-3}$ across all experiments. As shown by our experiments on different tasks (LLM pretraining and ViT fine-tuning for image classification), this configuration yields strong final metrics.
>
> 3. > Our method with/without scheduling (W3)
>
> a) Indeed, referring to the validation results of the methods on the LLM pre-training task (Table 2), one can observe that our approach with cosine scheduling outperforms AdamW with scheduling, while our method without scheduling performs worse.
> First, note that competing parameter-free optimizers, such as Prodigy [1], show inferior metrics compared to AdamW both with and without scheduling (see Table 2). Moreover, in [1], Algorithm 3 specifies the default stepsize configuration uses cosine annealing scheduling.
>
> b) As for our method, we would like to emphasize that the performance **without cosine scheduling**, and hence **without any step size tuning at all**, is only slightly worse than that of the tuned AdamW (see Tables 2, 10 for details). At the same time, the initial goal $-$ to lower runtime $-$ fulfilled, ALIAS is approximately **$1.5 \times$ faster**.
>
> Regarding Weakness 4, we kindly ask the reviewer to follow our next Official Comment, in which we provide a detailed response.

---

> ### Author Response · Authors · 2025-11-19
>
> 4. > Hyperparameters tuning for baselines (W4)
>
> a) Let us start with the question regarding the non-standard value of $\beta_2$. We performed a separate tuning of the parameters $\beta_1$ and $\beta_2$ on the LLM pre-training task. In our experiments, the setting $\beta_2 = 0.999$ resulted in better performance compared to $\beta_2 = 0.95$. Regarding the tuning of the weight decay parameter, we conducted a grid search over the values [0, 0.01, 0.1]. For AdamW, we obtained the following results: loss = 2.93425 for wd $= 0$, loss = 2.92839 for wd $= 0.01$, and loss = 2.93548 for wd $= 0.1$. Moreover, in our setup, these results were almost identical to those obtained for sign-based methods. For this reason, we fixed wd $= 0.01$ for all baselines in our experiments. As for the tuning of the schedule parameters, we used cosine annealing and varied the number of iterations used for warm-up. With 2% and 10% of the total number of iterations, the final metric values remained almost the same, so we chose the more standard setting of 10%. We also reduced the step size to 10% of its value after warm-up, and we did not change this parameter further. The hyperparameters tuning procedure is described in detail in Appendix A.1.1 and Appendix A.2.1 of our paper.
>
> b) In our view, we performed sufficiently thorough tuning of the baselines to allow them to reach strong metrics. Our tuned parameters are nearly identical to those used in a similar experimental setup in [1]. We also note that our parameter search ranges match those reported in [7]. We acknowledge that there is still room for further tuning, and with more extensive tuning, the baselines could potentially outperform our parameter-free methods. However, our goal is not to claim that perfectly tuned baselines cannot outperform parameter-free approaches. Rather, we aim to demonstrate how **efficient in terms of time** such methods can be while still maintaining **competitive performance** across different tasks. With our level of baseline tuning, our method accelerates the overall training process by approximately $1.5 \times$. If one were to improve the tuning further, for example, through more exhaustive exploration of scheduling hyperparameters, AdamW might eventually outperform ALIAS, but the tuning time, which ALIAS does not require, would increase dramatically. For example, in [5], the authors sampled 200 hyperparameter points, with each configuration running up to 50% of the full number of iterations $-$ a process that requires significantly more time than running our method. However, since a reviewer asked about the performance on the AlgoPerf benchmark, we evaluated our method on several tasks from this benchmark (see the next point).
>
> c) We evaluate ALIAS Adam version (Algorithm 3) on several tasks from the AlgoPerf benchmark [5]. In particular, we test our method on the MRI reconstruction and molecular property prediction (MPP) tasks. We validate only our ALIAS Adam version, while the results for the remaining methods are taken directly from Table 4 in [5]. For our method, we fix $\gamma^t = 10^{-3}$. The corresponding results are provided in the revised version of the paper in Appendix A.3 (Table 11) and in the table below.
>
> |Algorithm|MRI (SSIM $\uparrow$)|MPP (MAP $\uparrow$)|
> |-|-|-|
> |AdamW|0.723|0.254|
> DoG|0.714|0.231|
> D-Adapt (with Adam)|0.722|0.221|
> MoMo (with Adam)|0.723|0.221|
> Prodigy|0.723|0.212|
> ALIAS Adam version|0.724|0.242|
>
> The results demonstrate that our approach outperforms competing parameter-free algorithms on both tasks, and additionally surpasses the tuned AdamW baseline on the MRI reconstruction problem. These experiments further confirm the applicability of our method to tasks across different modalities, as well as its advantage over other parameter-free approaches.
>
> ---
>
> **References**
>
> [1] Mishchenko et al. Prodigy: An expeditiously adaptive … **ICML** 2024
>
> [2] Ivgi et al. Dog is sgd’s best friend … **ICML** 2023
>
> [3] Defazio and Mishchenko Learning-rate-free learning by d-adaptation. **ICML** 2023
>
> [4] Schaipp et al. MoMo: Momentum Models for Adaptive Learning Rates. **ICML** 2024
>
> [5] Kasimbeg et al. How far away are truly hyperparameter-free learning algorithms? **TMLR** 2025
>
> [6] Deconstructing what makes a good optimizer for language models. **OPT** 2024
>
> [7] Medapati et al. Training neural networks faster … **arXiv** 2025

---

> ### Author Response · Authors · 2025-11-27
>
> Dear Reviewer uXJ9,
>
> We would like to kindly follow up on our earlier response. We fully understand the heavy reviewing load and sincerely appreciate the effort you invest in evaluating submissions.
>
> If you have the opportunity, we would be grateful for any additional remarks or clarifications, particularly regarding our newly added experimental results. Your feedback on these updates is highly valuable to us.
>
> Thank you again for your time and consideration.
>
> Best regards,
> Authors

---

### Author Response · Authors · 2025-12-03

Dear Area Chair,

In this comment, we would like to summarize the discussion process.

The primary concerns raised by the reviewers focused on the performance of our algorithms across tasks of different modalities, the comparison against other parameter-free approaches, and the behavior of our procedure under high gradient noise. We provided the reviewers with these experimental results in our rebuttals, and we have also incorporated them into the paper in Appendix A.3, A.1.3, and A.1.2, respectively.

The reviewers additionally raised questions regarding the configuration of the experimental setup (Reviewers uXJ9 and RrFQ), convergence of our method in the stochastic setting (Reviewer JFUp), and the non-convex case (Reviewers RrFQ and HNri). We addressed these points in detail in our rebuttals.

As a result of the discussion, Reviewers HNri and JFUp increased their scores from 6 to 8 and from 4 to 6, respectively. Due to circumstances, we were not able to fully complete the discussion with Reviewers uXJ9 and RrFQ.

We truly appreciate the Area Chair’s efforts and the additional workload they have been required to take on under these circumstances.

Kind regards,

The Authors

---

### Meta-Review · Area_Chair_KLqq · 2025-12-30

**Summary:**

The paper is generally seen as a useful and timely contribution to the community, a parameter-free step-size prescription for SignSGD (ALIAS), with extensions to stochastic/distributed settings and practical variants (momentum / memory-efficient), plus experiments on LLM pre-training and vision fine-tuning. The concerns were not about whether the method works at all, but about generality, fairness of comparisons, and robustness. Given the discussion outcomes reported by the authors (HNri 6→8, JFUp 4→6), the rebuttal appears to have meaningfully reduced the major risks, pushing the paper toward a clear accept.

**Reviewer Concerns:**

**Reviewer uXJ9 (6)**

Addressed by rebuttal:

Missing parameter-free baselines: authors added DoG / D-Adaptation / MoMo, with results showing ALIAS Adam version outperforming these on LLaMA pre-training.

LM head being optimized by AdamW: authors clarified this design choice, argued it’s standard in related sign-based work.

Baseline tuning transparency: authors described search spaces for β₂, weight decay, warm-up, and schedules; they also contextualized tuning costs and cited AlgoPerf-style extensive tuning.

Generality to other tasks (AlgoPerf): authors added results on MRI reconstruction and molecular property prediction (Appendix A.3 / Table 11), showing competitiveness and even surpassing tuned AdamW on MRI.

Standardness of some baseline choices (e.g., β₂=0.999 vs 0.95): authors claim they tuned and found 0.999 better in their setup.

Still outstanding / partially outstanding:

Schedule tuning vs LR tuning: authors argue scheduling is standard even for other “parameter-free” methods, but the reviewer’s concern that “schedules still matter” is not eliminated.




**Reviewer JFUp (4)**

Addressed by rebuttal:

Stochastic algorithm clarity: authors separated practical vs theoretical variants and added a formal algorithm description.

SignSGD convergence concern: authors clarified convergence-to-neighborhood behavior, relationship to classic SignSGD theory, and what their contribution changes.





**Reviewer RrFQ (8)**

Addressed by rebuttal:

Sensitivity to initial guess: added ablation showing rapid stabilization and unchanged final loss (Appendix A.1.2).

Distributed setting detail: clarified synchronization of quantities and that ALIAS is updated globally (not locally per device).

Hyperparameter-free baselines and modality breadth: additional baselines + AlgoPerf tasks strengthen breadth.



Still outstanding (minor):

Non-convex theory gap: authors acknowledge as future work (not solved).

Adam-style variant not fully parameter-free: acknowledged; authors treat as outside scope.





**Reviewer HNri (6)**

Addressed by rebuttal:

Limited experiments / modality generality: authors added MRI + MPP AlgoPerf tasks and cross-modal evidence.

Robustness to gradient noise: ablation across batch sizes (512→64) showing similar degradation vs SignSGD, supporting robustness claim.

Distributed/high-noise behavior concerns: addressed via ablations + discussion.



Still outstanding (minor):

Non-convex theoretical guarantees remain open (but now positioned as future work rather than a blocker).

**Reviewer Scores:**

uXJ9 (initial 6) → 6 (likely unchanged):
the main concerns were directly addressed with new experiments and clearer tuning details.

JFUp (initial 4) → 6 (already happened): rebuttal addressed the main concerns.

RrFQ (initial 8) → 8 (likely unchanged):
already strongly positive; rebuttal answers sensitivity/distributed details.

HNri (initial 6) → 8 (already happened): reviewer explicitly says “strong enough for an oral” and raised score.

---

### Decision · Program_Chairs · 2026-01-26

Accept (Poster)